# Exceptional Multi Stage Mineralization of Secondary Minerals in Cavities of Flood Basalts from the Deccan Volcanic Province, India

**Berthold Ottens** [1,*]**, Jens Götze** [2]**, Ralf Schuster** [3]**, Kurt Krenn** [4]**, Christoph Hauzenberger** [4]**, Benkó Zsolt** [5] **and Torsten Vennemann** [6]

1    Kalkofenstraße 15, D-96194 Walsdorf, Germany
2    Technische Universität Bergakademie Freiberg, Institute of Mineralogy, Brennhausgasse 14, D-09599 Freiberg, Germany; jens.goetze@mineral.tu-freiberg.de
3    Geologische Bundesanstalt, Neulinggasse 38, A-1030 Wien, Austria; Ralf.Schuster@geologie.ac.at
4    Institut für Erdwissenschaften, Universität Graz, Universitätsplatz 2, A-8010 Graz, Austria; kurt.krenn@uni-graz.at (K.K.); christoph.hauzenberger@uni-graz.at (C.H.)
5    Institute for Nuclear Research, Hungarian Academy of Sciences, Bem tér 18/C, H-4026 Debrecen, Hungary; benko.zsolt@atomki.mta.hu
6    Institute of Earth Surface Dynamics, University of Lausanne, CH-1015 Lausanne, Switzerland; torsten.vennemann@unil.ch
*    Correspondence: ottens-mineralien@t-online.de; Tel.: +49-9549-980412

**Abstract:** Flood basalts of the Deccan Volcanic Province erupted between about 67.5 to 60.5 Ma ago and reached a thickness of up to 3500 m. The main part consists of compound and simple lava flows with a tholeiitic composition erupted within 500,000 years at about 65 Ma. Within the compound lava flows, vesicles and cavities are frequent. They are filled by secondary minerals partly of well development and large size. This study presents data on the secondary mineralization including detailed field descriptions, optical, cathodoluminescence and SEM microscopy, X-ray diffractometry, fluid inclusions, C and O isotope analyses, and Rb-Sr and K-Ar geochronology. The investigations indicate a multistage precipitation sequence with three main stages. During stage I clay minerals and subsurface filamentous fabrics (SFFs), of probably biogenic origin, formed after the lava flows cooled down near to the Earth's surface. In stage II, first an assemblage of calcite (I) and zeolite (I) (including mordenite, heulandite, and stilbite) as well as plagioclase was overgrown by chalcedony, and finally a second calcite (II) and zeolite (II) generation developed by burial metamorphism by subsequent lava flows. Stage III is characterized by precipitation of a third calcite (III) generation together with powellite and apophyllite from late hydrothermal fluids. Rb-Sr and K-Ar ages of apophyllite indicate a large time span for stage III. Apophyllite formed within different time intervals from the Paleogene to the early Miocene even within individual lava flows at certain localities. From the Savda/Jalgaon quarry complex, ages cluster at 44–48 Ma and 25–28 Ma, whereas those from the Nashik area are 55–58 Ma and 21–23 Ma, respectively.

**Keywords:** Deccan Volcanic Province; zeolites; subsurface filamentous fabrics; biosignatures; apophyllite; geochronology; multistage mineralization

---

## 1. Introduction

The Deccan Volcanic Province (DVP) is one of the largest igneous provinces on Earth. Therefore, the flood basalts of the DVP have been extensively studied concerning their genesis, composition, stratigraphy, and timing of eruption during the last decades [1–6]. One conspicuous feature of

the Deccan basalts is the frequent occurrence of secondary minerals (e.g., zeolites, carbonates, and silica minerals) in amygdales and cavities of the rocks, often forming large, euhedral crystals [7]. The precipitation of these secondary minerals in the basaltic rocks is commonly explained either by low-grade metamorphic reactions in different P/T zones during burial [8–11] or in the context of a hydrothermal origin [12,13]. Early investigations on secondary minerals of the DVP favored the hypothesis of a formation in different burial zones [8,14]. However, late investigations by Jeffery [15] disclosed the hypothesis of burial zoning based on the distribution, stratigraphy, and depth variation of the secondary minerals. Further results of fluid inclusion studies of lately formed secondary minerals from the DVP [16], and investigations concerning the spatial distribution of specific mineral species [17] provided strong arguments for a late hydrothermal mineralization.

Since 1975, well-developed minerals of DVP were increasingly observed in new occurrences and have updated the knowledge about the distribution of the secondary minerals. Besides the booming trading process, new sites were discovered not only in the western DVP but also in the central area, and their minerals have been reported [18]. In the present study, the mineralization sequence of secondary minerals in open cavities of a huge quarry complex (20°59'N, 75°27'E, 230 m asl) at Savda village close to the city of Jalgaon (Maharashtra, India) as well as neighboring occurrences in the State of Maharashtra, India, were investigated to reveal the mineralization processes of the secondary minerals in the DVP (Figure 1).

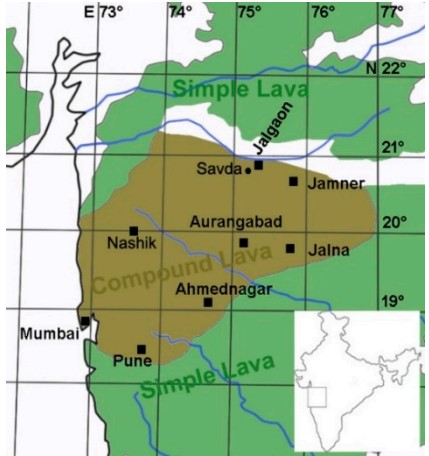

**Figure 1.** Map of the western and central part of the Deccan Volcanic Province (DVP), India.

The quarry complex of Savda is characterized by an exceptional mineralization of large SFF encrusted by chalcedony, unusual twinned calcite crystals, and lately formed zeolites and apophyllite in the same cavities. This complex sequence of secondary minerals provides information about the timing and physico-chemical conditions of the mineralization process.

Alteration and secondary mineralization have been extensively studied in several other basalt provinces. Most basic work was undertaken for the basalts of Iceland [19]. Later studies on the basalts of Iceland [11], Greenland [20,21], Faroe Islands [22], Isle of Skye, Scotland [23], and Northern Ireland [24] completed the understanding about the formation processes of zeolites occurring in volcanic provinces.

Zeolitization was also investigated in other volcanic rocks such as in Italy [13], Iran [25,26], and Kerguelen Islands [27]. Of interest are the studies of the basalts of the Columbia River Province with similarities to the DVP [28] and to the southeastern Parana Basin, Brazil [29,30]. The comprehensive data considering the formation of zeolites and associated secondary minerals can partially compensate and supplement missing data and information of the DVP. General aspects of basalt alteration are widely accepted according to the basic work of Kristmannsdottir [31] about the alteration of basaltic rocks by hydrothermal activity; therefore, they were commonly not investigated and not described

in detail in papers about the secondary minerals, predominantly zeolites in specific basalt provinces. Considering the petrographic composition and texture of the tholeiitic rocks in Savda and the main part of the DVP, a detailed investigation of the alteration in the lava flow core line was not considered.

Most work about the zeolitization of specific basalt provinces is based on the samples found in small vesicles, amygdales, and veins [9,11,22–27]. The size of these cavities is not big enough to contain in each case the whole mineral sequences precipitated during all stages of mineralization. According to the variation of porosity and permeability during alteration and mineralization processes over a long time span, the fillings of the vesicles, amygdales, and veins can widely differ, even in neighboring ones [32]. In contrast, the big cavities in the lava flow core zone in Savda often contain, because of their size, a full spectrum of minerals precipitated over a long period, which allows an unusual insight into the mineralization sequence.

Another specific condition of the big cavities in Savda is the occurrence of apophyllite together with different calcite generations, which enables age dating and investigations of fluid inclusions. Commonly, secondary mineralization of clay minerals and zeolites in the basalts is explained as result of hydrothermal alteration or low-grade metamorphism during the burial stage. The fluid inclusion data of calcite in Savda allow a comparison with the estimated formation temperatures of zeolites. Age dating of most zeolites is not possible because of too low contents of usable radiogenic isotopes. Mineralization processes are commonly assumed to happen shortly after crystallization of the basalt and subsequent burial, or they are associated with later hydrothermal events but without chronological data. The age data of apophyllite from Savda for the first time provide evidence for late geological events in the DVP.

## 2. Geological Setting and Secondary Mineralization

### 2.1. Geological Setting

The flood basalts of the DVP erupted through fissures between about 67.5 to 60.5 Ma ago [2,33,34], and three main volcanic pulses can be distinguished, starting with phase 1 around 67 Ma ago [35]. After 2 Ma of quiescence, large volumes of predominantly tholeiitic lava erupted during the main phase at about 65 Ma. During this second phase approximately 80% of the total lava pile was formed in a time span of 500,000 years or less [1,36]. The third phase was prominent in the southern part of the DVP and lasted from 64 to 60 Ma. The central part of the DVP stratigraphically belongs to the Sahyadri Group, which is subdivided in order of age into a Lower and Upper Ratangarh, Ajanta and Shikhli Formation [37]. The lava flows can be distinguished into compound and simple flows based on their typical surface morphology [38,39]. The units of a compound flow are vesicular with pipe amygdales at the base and highly vesicular at the top with spherical vesicles and a rather dense core zone. Individual flows consist of a number of sheet-like emplaced cooling units and ellipsoidal toes.

Chemical analyses of the Deccan Trap rocks show that the lavas are mainly of tholeiitic type [40–46]. Most common is fine-grained, aphyric, dense, dark grey basalt in the plateau subprovince. In numerous occurrences, a porphyritic texture with phenocrysts of plagioclase, pyroxene, and olivine with a varying content and size is observed. The surrounding groundmass of the phenocrysts consists of finer grains of plagioclase, clinopyroxene, olivine, and glass [40].

The basalts in the area of the Savda quarries belong to the Ajanta Formation of the Sahyadri Group consisting of 16 basaltic lava flows with alternating simple and compound characters [37]. The basaltic flows of this group are nonporphyritic to sparsely porphyritic. The compound flows comprise a few to several units and exhibit the typical features like pipe amygdales, ropy structures, squeeze ups, and vesicular tops with spheroidal vesicles.

The maximum depth of burial of the basalt in the Savda area is unknown. It is believed that the maximum thickness of the DVP had been approximately 3500 m near Nashik with an escarpment to the western margin [36]. Based on a rough calculation by a linear decreasing paleo-surface, the total thickness of overlaying basalt and burial at Savda after eruption can be assumed to be 1500 m.

The paleo geothermal gradient during the period of eruption of the DVP in the Savda region is unknown. Neuhoff et al. [9] estimated for basaltic lavas at Teigarhorn, Eastern Iceland, a paleo geothermal gradient of 53 ± 7 °C/km, Weisenberger and Selbekk [11] estimated for Hvalfjördur area, Iceland, a paleo geothermal gradient of 133 ± 10 °C/km, and Jørgensen [18] estimated for specific areas of the Faroe Islands a paleo geothermal gradient between 56 ± 7 and 63 ± 8 °C/km. It can be speculated that the geothermal gradient in Savda was between 50 and 130 °C/km. Assuming similar values for a maximum burial of 1500 m at Savda, a temperature of 150 °C can be estimated. It is assumed that the geothermal gradient decreased significantly after the end of the volcanic activity of the DVP.

### 2.2. Distribution and Mineralization Sequence of Secondary Minerals

Walker [8] investigated the frequency of secondary amygdale minerals from the DVP regarding their distribution and postulated three zones (laumontite, scolecite, and heulandite) as a result of low-grade regional metamorphism. Sukheswala et al. [14] studied the distribution of the zeolites and associated minerals, and they pointed out that their occurrence was restricted to certain localities around Bombay (Mumbai), Baroda (Vadodara), Poona (Pune), and Nasik (Nashik) and could not be found generally in the Western Deccan Traps. It was believed that laumontite, scolecite, and heulandite zones in ascending order were indicative of burial metamorphism, and the crystallization sequence in the cavities appeared mainly temperature controlled. Jeffery et al. [15] studied the distribution of the secondary minerals in the three proposed zones of the Western Deccan Traps and plotted them against the recent altitude and individual formations with defined geochemical signatures. The more common species occurred in all formations and levels providing no evidence for the zonation scheme proposed by Walker [8] and Sukheswala et al. [14]. Table 1 represents the distribution of secondary minerals for a few significant localities of the DVP and is based on long-term observations [18,47]. The reported distribution of the secondary minerals is based on sampling from large vesicles and cavities. It has to be pointed out that several species, such as levyne, phillipsite, gismondine, and offretite, which are common as secondary minerals in other basalt provinces [9,10,13], do not occur in the main part of the DVP. The zeolites from the DVP are characteristic for tholeiitic host rocks and are predominantly Ca rich. The altitude is given as an indicator for different stratigraphic units.

Sabale and Vishwakarma [7] explained the mineralization based on a two-stage model. In the first stage, the zeolites formed because of the interaction of the entrapped volatiles with the host rock, which explained the formation of minute amygdales. In the second stage, minerals formed at high temperatures during the emplacement of dykes and diastrophism (part of geotectonics) [7]. The authors believed that the absence of a vertical zonation of zeolites in the Western DVP disclosed the possibility of burial metamorphism. In addition, fluid inclusion studies on secondary cavity minerals like amethyst, powellite, apophyllite, and scolecite from the DVP carried out by Srikantappa and Mookherjee [16] showed the presence of five types of fluid inclusions and high homogenization temperatures of 250 to 280 °C, which could not be explained by a low-grade burial metamorphism in the investigated area.

More recent studies concerning the general mineralization sequence at specific outcrops such as in the Mumbai spilite were carried out by Ottens [18,47]. It was concluded that, in addition to a general sequence of mineralization at different locations in the DVP, several specific sequences exist (Figure 2a–c), which need further detailed investigations to establish a thorough genetic model.

**Table 1.** Frequency of secondary minerals in cavities of different occurrences within the DVP: Nashik 1 (19°56′N, 73°43′E) and Nashik 2 (19°54′N, 73°57′E)—several quarries. Pune—former quarries at Pashan hill (18°31′N, 73°48′E) and several quarries at Wagholi (18°36′N, 74°E). Savda—quarry complex (20°59′N, 75°27′E). Malad (19°11′N, 72°52′E)—former quarry complex. *Ca-Fe minerals are ilvaite, babingtonite, pumpellyite, julgoldite, hematite, and pyrite. Vr—very rare; r—rare; o—occasionally; c—common; and f—frequent.

| Altitude asl [m] | Nashik 1 720-650 | Nashik 2 650-600 | Pune 680-590 | Savda 230-220 | Malad 40-30 |
|---|---|---|---|---|---|
| apophyllite | f | f | f | f | c |
| calcite | c | c | o | f | f |
| Ca-Fe minerals* | | | | | c |
| cavansite | | | c | | |
| chabazite | | | | | r |
| clay minerals | f | f | f | f | f |
| epistilbite | r | r | | | |
| fluorite | | r | | | |
| goosecreekite | | vr | | | |
| gyrolite-okenite | | | | | f |
| heulandite | f | f | f | f | r |
| laumontite | | c | | | c |
| mesolite | | | c | | |
| mordenite | | | | r | |
| natrolite | | | | | r |
| powellite | vr | | | vr | |
| prehnite | | | | | c |
| quartz-chealcedony | f | f | | f | c |
| scolecite | | c | | | r |
| stilbite-stellerite | f | f | f | f | r |
| yugawaralite | | | | | vr |

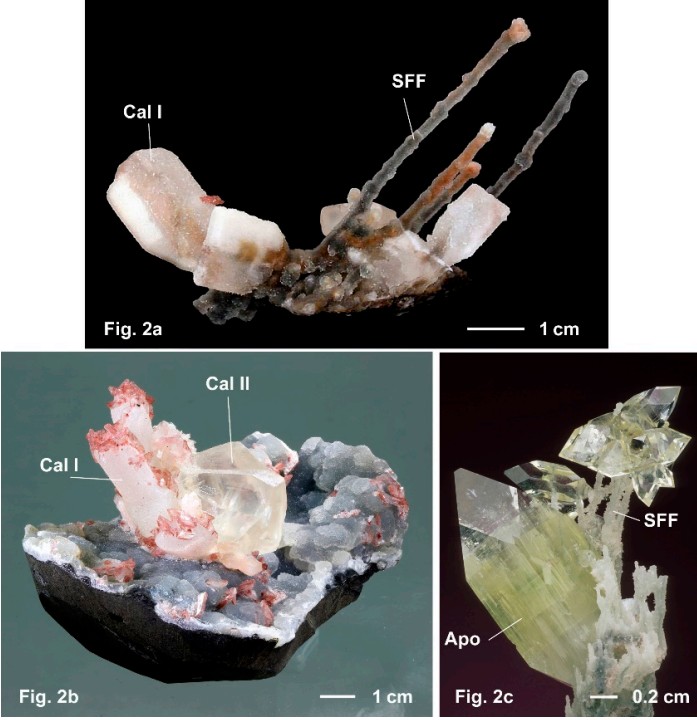

**Figure 2.** Hand specimens illustrating the precipitation sequence of secondary minerals from the DVP at Savda. (**a**) Twinned calcite I (Cal I) crystals and subsurface filamentous fabrics (SFFs) overgrown by chalcedony (© Superb Minerals Pvt. Ltd, Nashik, India). (**b**) Twinned calcite I (Cal I) crystals on basalt overgrown by chalcedony and later on by yellowish calcite II (Cal II) crystals and red heulandite. (**c**) Double-terminated apophyllite crystals on chalcedony subsurface filamentous fabrics (SFFs) (© J. Scovil, Phoenix, AZ, United States).

The investigated lava flow in Savda contains different kinds of cavities. Small ones up to approximately 5 cm in size occur in the upper and bottom line of the flow and are usually described as vesicles or as amygdales filled with secondary minerals, but they are not the subject of this paper. In the core line of the flow, the irregularly shaped large cavities are more or less partly filled with secondary minerals, commonly with a free space in the center. They reach up to approximately 50 cm (maximum is approximately 100 cm wide) in size, and are usually not covered by the term vesicle or amygdule. We follow other authors [9,25] in using the term vesicle for small open cavities, amygdales for small filled cavities, and the term cavity for large filled cavities. The shape of the cavities varies widely, even for neighboring cavities (Figure 3). Rather round cavities are common, and sometimes they are horizontally or vertically elongated. Cavities with a flat bottom and an upward convex, skirted shape occur frequently. Tunnels or breccia were not observed. Big cavities indicate in several cases an interconnection of smaller ones. Photographs of six cavities in Figure 3 represent a variation of shape and fillings.

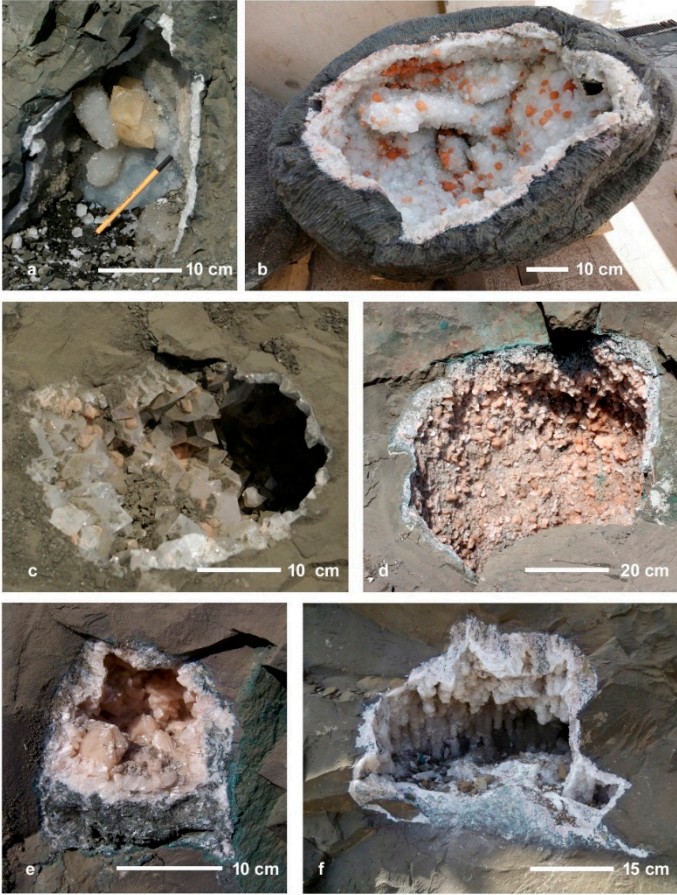

**Figure 3.** Cavities from the core line in Savda representing different shapes and sizes. The secondary minerals are chalcedony, calcite, stilbite, apophyllite, and minor heulandite. (**a**) arched roof, (**b**) big cavity with flat bottom, (**c**) round shape, (**d**) upward convex, skirted shape, (**e**) flat bottom, and (**f**) flat bottom, upward convex skirted shape.

## 3. Materials and Methods

### 3.1. Sample Material

The sample material mainly included secondary minerals from the big, freshly opened cavities within the flow core line in the quarry complex of Savda near Jalgaon (Maharashtra, India) and was collected by two authors during a field trip in February 2013. Additional samples for comparison were selected from other outcrops in the surrounding DVP complex in the Nashik area (19° 53′N, 73° 56′E,

670 m asl) and Jamner (20° 47'N, 75° 44'E, 266 m asl). Numerous field observations (between 1996 and 2016) and the careful evaluation of dozens of hand specimens provided information concerning the mineralization sequence of the secondary minerals. Representative samples for all mineralization steps were selected for further analytical studies (Tables 2 and 3). Therefore, thin sections and polished sections were prepared, and pieces of selected minerals were isolated for isotopic investigations.

**Table 2.** Localities of the investigated samples from the Deccan volcanic province (India). Abbreviations: Apo = apophyllite, Cal = calcite, Cha = chalcedony, Heu = heulandite, Qtz = quartz, Sap = saponite, and Stb = stilbite.

| Sample Number | Sample Material | Locality | Coordinates (WGS84) | |
|---|---|---|---|---|
| | | | N | E |
| **JAL-1** | Cal III, Stb and Apo crystals from single cavity | Jalgaon, Savda | 20° 59' 12.1" | 75°26'26.7" |
| JAL-2 | Cal III, Stb and Apo crystals from single cavity | Jalgaon, Savda | 20° 59' 12.2" | 75°26'56.8" |
| JAL-05 | complex filament, handspecimen | Jalgaon, Savda | 20° 59' | 75°27' |
| JAL-11 | single filament | Jalgaon, Savda | 20° 59' | 75°27' |
| JAL-12A | Apo crystals from single filament | Jalgaon, Savda | 20° 59' 05.7" | 75°27'04.1" |
| JAL-13A | basalt with clay mineral, handspecimen | Jalgaon, Savda | 20° 59' 05.5" | 75°27'01.2" |
| JAL-13B | basalt, handspecimen | Jalgaon, Savda | 20° 59' 05.5" | 75°27'01.2" |
| JAL-13C | Apo crystal, 4 × 4 × 1 cm | Jalgaon, Savda | 20° 59' 05.5" | 75°27'01.2" |
| JAL-13D | Apo crystal, 4 × 4 × 1 cm | Jalgaon, Savda | 20° 59' 05.5" | 75°27'01.2" |
| JAL-16B | Qtz, Apo 1, Cal III, Apo 2 on handspecimen | Jalgaon, Savda | 20° 59' 13.5" | 75°27'01.2" |
| JAL-24 | clay mineral on handspecimen | Jalgaon, Savda | 20° 59' | 75°27' |
| JAL-27 | clay mineral on handspecimen | Jalgaon, Savda | 20° 59' | 75°27' |
| JAL-31 | Cal I overgrown by Cha, 5 × 2 × 2 cm | Jalgaon, Savda | 20° 59' | 75°27' |
| JAL-32 | Cha, Cal I, Heu, Cal II on basalt, handspecimen | Jalgaon, Savda | 20° 59' | 75°27' |
| JAL-33 | Cha, Cal II, Apo on handspecimen | Jalgaon, Savda | 20° 59' | 75°27' |
| JAL-CHO | wallrock with clay mineral, Cha and Mor | Jalgaon, Savda | 20° 59' | 75°27' |
| JAL-SM-01 | filament without Cha overgrown | Jalgaon, Savda | 20° 59' | 75°27' |
| JAM-01 | section of single filament | Jamner | 20° 47' | 75°44' |
| JAM-10 | section of a complex filament | Jamner | 20° 47' | 75°44' |
| JLN-01 | section of a complex filament | Jalna | 19° 49.3' | 75°51.1' |
| NAS-6 | Stb and Apo on handspecimen | Nashik, Panduleni | 19° 56' 24" | 73°43'13" |
| NAS-9 | Stb and Apo on handspecimen | Nashik, Dindori | 20° 08' 39" | 73°49'25" |
| NAS-10 | Heu, Stb and Apo on handspecimen | Nashik, Mohu | 19° 54' 24.8" | 73°56'36.9" |
| NAS-11 | Qtz, Stb and Apo on handspecimen | Nashik, Mahodari | 19° 52' 51.2" | 73°56'26.0" |

**Table 3.** Investigated material of the present study and applied analytical methods (Apo—apophyllite, Cal—calcite, Cel— celadonite, Qtz—quartz, Stb—stilbite, and Zeo—zeolite).

| Sample | XRD, SEM/EDX (Table 4) | Fluid Incl. (Table 5) | C and O Isotope (Table 6) | Mineral Chemistry (Tables 7 and 8) | Rb-Sr Dating (Table 9) | K-Ar Dating (Table 10) |
|---|---|---|---|---|---|---|
| JAL-1 | | | | | Cal III, Stb, Apo | |
| JAL-2 | | Cal III | | | Stb, Apo | |
| JAL-05 | Clay min. | | | | | |
| JAL-11 | Zeo, Clay min. | | | | | |
| JAL-12A | Clay min. | | | | Apo | Apo |
| JAL-13A | Clay min. | | | | Cel | |
| JAL-13B | | | | Apo | whole rock | |
| JAL-13C | | | | | Apo-core/rim | Apo-core/rim |
| JAL-13D | | Apo | | Apo | | |
| JAL-16B | Clay min. | Qtz, Cal III | | | Cal III, Apo 1. Apo 2 | Apo |
| JAL-24 | Clay min. | | | | | |
| JAL-27 | Clay min. | | | | | |
| JAL-31 | | Cal I | Cal I | | | |
| JAL-32 | | Cha, Cal I + II | Cal I +II | | | |
| JAL-33 | | Cal II | Cal II | | | |
| JAL-CHO | Clay min. | | | | | |
| JAL-SM-01 | Clay min. | | | | | |
| JAM-01 | Clay min. | | | | | |
| JAM-10 | Zeo, Clay min. | | | | | |
| JLN-01 | Clay min. | | | | | |
| NAS-6 | | | | | Stb, Apo | Apo |
| NAS-9 | | | | | Stb, Apo | |
| NAS-10 | | | | | Stb, Apo | Apo |
| NAS-11 | | | | | Stb, Apo | Apo |

*3.2. Optical Microscopy, Scanning Electron Microscopy (SEM) and Electron Microprobe Analysis (EMP)*

Polarizing microscopy in transmitted light on polished thin sections was performed on a ZEISS Axio Imager A1m microscope (ZEISS, Thornwood, NY, USA). Documentation of microtextures in linear and crossed polarized light was realized using a digital camera Axiocam MRc5 and the software Axiovision (Version 4.6, ZEISS microscopy, Jena, Germany). SEM measurements (SE, BSE) were performed on carbon-coated thin sections and sample pieces using a JEOL JSM-7001F (20 kV, 2.64 nA) (JEOL, Tokyo, Japan) with a BRUKER EDX system. In addition, sample pieces without carbon coating were analyzed using a ZEISS EVO 10 SEM coupled with a BRUKER Quantax EDS system. (Table 4).

Quantitative chemical analyses of minerals were carried out at the NAWI Graz Geocenter Institute of Earth Sciences, Karl-Franzens University, Graz, Austria, with a JEOL 6310 SEM equipped with a LINK ISIS energy dispersive system (counting time 100 s) and a MICROSPEC wavelength dispersive system (counting time on peak and total background: 20 s). Standard analytical conditions were set to an accelerating voltage of 15 kV and 6 nA sample current. Matrix corrections were made using the phi-rho-z procedure. Natural minerals were used as standards.

*3.3. Cathodoluminescence (CL)*

Cathodoluminescence (CL) measurements were performed on carbon-coated, polished thin sections using a "hot cathode" CL microscope HC1-LM (LUMIC, Bochum, Germany). [48]. The system was operated at 14 kV accelerating voltage and a current of 0.2 mA (current density of about 10 μA/mm$^2$). Luminescence images were captured "on-line" during CL operations using a peltier cooled digital video camera (OLYMPUS DP72, OLYMPUS Deutschland GmbH, Hamburg, Germany). CL spectra in the wavelength range 380 to 1000 nm were recorded with an Acton Research SP-2356 digital triple-grating spectrograph with a Princeton Spec-10 CCD detector (OLYMPUS Deutschland GmbH, Hamburg, Germany) that was attached to the CL microscope by a silica-glass fiber guide. CL spectra were measured under standardized conditions (wavelength calibration by an Hg-halogen lamp, spot width 30 μm, and measuring time 1 s). For characterization of the transient CL behavior of hydrothermal quartz, CL spectra were taken after distinct times of electron irradiation.

*3.4. X-Ray Diffractometry*

The mineralogical compositions of separated and prepared (<20 μm) fractions were analyzed by X-ray diffraction using a URD 6 (Seifert/Freiberger Präzisionsmechanik) (Seifert/Freiberger Präzisionsmechanik, Freiberg, Germany) with Co Kα radiation in the range 5–80° (2θ). Analytical conditions included a detector slit of 0.25 mm, 0.03° step width, and 5 s measuring time. Data were evaluated qualitatively with Analyse RayfleX v.2.352 software (Version 2.352, International Union of Crystallography, Chester, England) (Table 3).

*3.5. Fluid Inclusions*

Fluid inclusions (FIs) were investigated in doubly polished sections with a thickness of ~0.15 mm using a LINKAM THSMG600 heating and freezing stage with an operating range from –196 °C to +600 °C at the Institute of Earth Sciences, University of Graz, Austria. The Synthetic Fluid Inclusion Reference Set (Bubbles Inc. Blacksburg, VA, USA 24062-0146) was used for stage calibration. T-measurements were reproducible to within 0.2 °C at a heating rate of 0.1 °C/min. Fluid densities and salinities were calculated with the appropriate equations of state (EoS) by using the program bulk in the software package FLUIDS 1 [49]. EoS after Oakes et al. [50] was taken to calculate FI properties of the aqueous system. All FIs were initially cooled to below –190 °C and subsequently heated for phase identification. Optical examinations were used to determine the temperatures of phase transitions. Depending on the compositional system for any given FI, the following values were documented (L = liquid; V = vapor; S = solid that melts at eutectic): ($T_e$) eutectic temperature or apparent eutectic temperature (e.g., SIceV → IceLV); $T_e$ means the minimum temperature of liquid stability in a specified

system and was used to identify the saline aqueous fluid system after Davis et al. [51] and Goldstein and Reynolds [52]; $T_m$(Ice) final melting temperature of ice (IceLV → LV); $T_m$(Ice) was taken to calculate salinities of aqueous fluid inclusions using freezing point depression as well as equations after Bodnar [53]. Homogenization temperatures $T_h$(LV → L or V) for aqueous inclusions were measured to avoid leakage or decrepitation and to get minimum conditions for formation of homogeneously trapped FIs. All microthermometry data are given in Table 5.

*3.6. C and O Isotope Analyses*

The C and O isotope analyses of calcite were performed in the stable isotope laboratory of the University of Lausanne (Lausanne, Switzerland) following the method of acid digestion at 70 °C [54]. Extracted $CO_2$ was cleaned by cryogenic distillation and analyzed on a Finnigan MAT 252 mass spectrometer. $\delta^{13}C$ and $\delta^{18}O_C$ values were calculated relative to the Pee Dee Formation belemnites (VPDB) and VSMOW, respectively (Table 6).

*3.7. Geochronology*

Mechanical sample preparations for Rb-Sr and K-Ar analyses were performed at the Geological Survey of Austria in Vienna. Crystals (1 cm in size) of apophyllite and calcite, stilbite aggregates, and fine-grained celadonite masses were selected in the field and broken off from hand specimens. To select the core and rim of apophyllite (4 cm in diameter) from sample JAL-13C, the central part of the crystal was cut by a diamond saw. Apophyllite, stilbite, and calcite were crushed in a hand mortar and sieved to select a grain size of 0.2–0.3 mm.

Chemical sample preparations for Rb and Sr isotope analyses were performed at the Geological Survey of Austria in Vienna. Weights of samples used for dissolution were about 200 mg. Apophyllite, stilbite, and celadonite were dissolved in a mixture of $HF:HNO_3 = 4:1$, whereas calcite was dissolved in HCl. Chemical sample preparation followed the procedure described by Sölva et al. [55]. Isotopic measurements were done at the Department of Lithospheric Research at the University of Vienna. Rb and Sr concentrations were determined by isotope dilution using mixed Rb-Sr spikes. Rb ratios were measured with Finnigan® MAT 262 MC-TIMS, whereas Sr ratios were analyzed with a ThermoFinnigan®Triton MC-TIMS (Thermo Fisher Sientific, Waltham, Massachusetts, USA). Sr was run from Re double filaments, whereas Rb was evaporated from a Ta single filament. Total procedural blanks were <1 ng for Rb and Sr. During periods of measurement, the NBS987 standard yielded $^{86}Sr/^{87}Sr$ = 0.710248 ± 4 (*n* = 17) on the Triton TI. Ages were calculated with the software ISOPLOT/Ex [56] assuming an error of 1% on the $^{87}Rb/^{86}Sr$ ratio (Table 8).

The separated size fractions of apophyllite were analyzed by the K-Ar method following the procedure of Balogh [57] in the K-Ar laboratory of the Institute for Nuclear Physics, Hungarian Academy of Sciences, Debrecen. Potassium content was measured on 50 mg sample aliquots, after dissolution by HF and $HNO_3$, with a Sherwood-400-type flame spectra-photometer with accuracy better than ±1.5%. Separated mineral sample splits were subjected to heating at 100 °C for 24 h under vacuum to remove atmospheric Ar contamination that was adsorbed on the surface of the mineral particles during sample preparation. Argon was extracted from the minerals by fusing the samples by high-frequency induction heating at 1300 °C. Released gases were cleaned in two steps in a low-blank vacuum system by St-700 and Ti-getters. Isotopic composition of the spiked Ar was measured by a Nier-type mass spectrometer and corrected for the atmospheric $^{40}Ar/^{36}Ar$ ratios [58]. The accuracy and reproducibility of isotope ratio measurements were periodically controlled by the Rodina 2/65 internal standard. Decay constants recommended by Steiger and Jäger [59] were used for age calculation with an overall error of ±2% (Table 10).

## 4. Results

### 4.1. Observations in the Outcrops and Host Rock Characteristics

A typical structure of a basalt flow in the quarries of Savda (about 7 m high) consisting of different units is shown in Figure 4. The photo represents an individual flow at the quarry wall with a slightly convex surface and numerous large cavities in the core line (3).

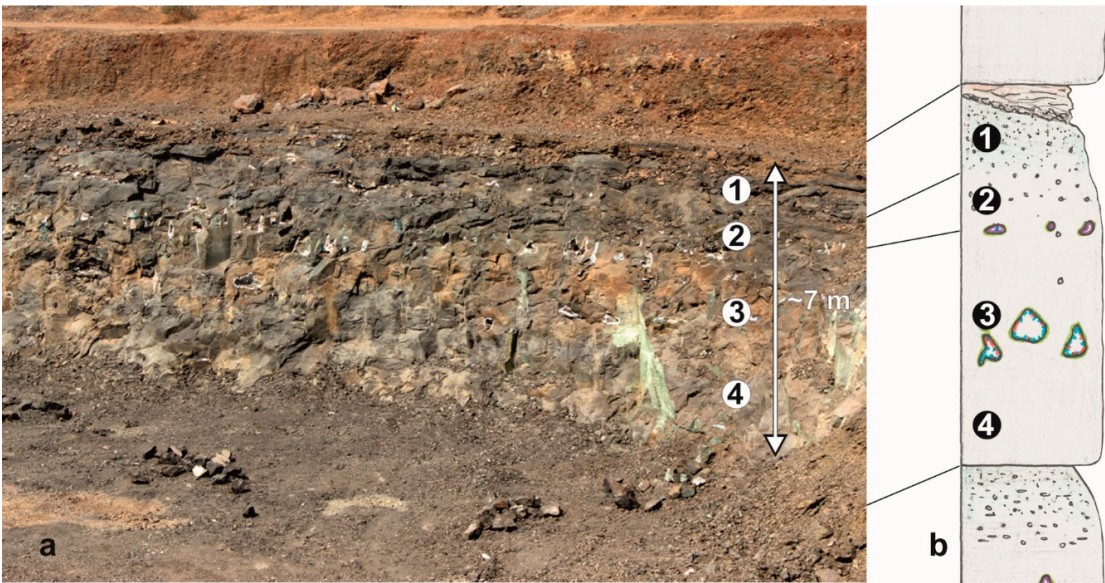

**Figure 4.** Structure of a basalt flow in the quarries of Savda. (**a**) Rock wall in the quarry (about 7 m high). (**b**) Schematic column of the flow. The individual flows (Figure 4a,b) are subhorizontal and mostly about 7 m in thickness. (1) The top is formed by an erosional surface or by a layer consisting of weathered basaltic material. (2) The upper part of the flow is rich in vesicles that are millimeters to a few centimeters large and are mostly completely filled up by secondary minerals such as celadonite, smectite, heulandite, epistilbite, and/or chalcedony. (3) In the core line of the flow, consisting of a rather dense basalt, large (several decimeters and up to more than one meter), and often irregularly shaped cavities with a rim of celadonite or smectite, well-developed often centimeter-sized euhedral crystals of heulandite, stilbite, calcite, apophyllite, and chalcedony SFF occur. (4) The lower part of the flow contains millimeter to a few centimeter-sized vesicles, filled more or less with clay minerals, chalcedony, zeolites, and calcite. The massive basalt is fine-grained with a few plagioclase phenocrysts. Olivine is often replaced by secondary minerals. The basalt is fractured, and the big cavities in the core line are connected by fissures with the upper zones.

The host rocks of the investigated outcrops mainly represented tholeiitic basalts. Data from literature showed a variable chemical composition with 49.93–51.96 wt. % $SiO_2$, 13.28–14.76 wt. % $Al_2O_3$, 12.73–14.00 wt. % $Fe_2O_3$ ($Fe^{2+}$ and $Fe^{3+}$ given as $Fe_2O_3$), 9.19–10.55 wt. % CaO, 4.43–6.85 wt. % MgO, 2.35–2.95 wt. % $Na_2O$, 0.48–0.54 wt. % $K_2O$, and 1.86–3.13 wt. % $TiO_2$ (41). The basalts in the flow core line in Savda were variably fresh and showed the usual fine-grained, intersertal texture with some porphyric plagioclase crystals up to 2 cm in length and altered olivine (Figure 5a). In the fine-grained groundmass, lath-shaped plagioclase microlithes and clino-pyroxene (augite according to XRD analyses) were freshly preserved, whereas former glass and probably olivine were replaced by brown, greenish-brown, or more seldom green clay minerals and opaque ore minerals (Fe-oxides/hydroxides). In the upper part of the flow, alteration was more intense, and Fe-oxides/hydroxides were present everywhere along the grain boundaries.

The porphyric plagioclase contains two types of inclusions and intergrowths, respectively. Those formed by hematite were derived from olivine as indicated by their shape. Roundish inclusions

of green clay minerals may have formed from glass. The pseudomorphs after olivine exhibited mash textures and consisted of greenish clay minerals and Fe oxides/hydroxides. Small vesicles were completely filled by clay minerals. Bigger vesicles up to 2 cm contained additional secondary minerals such as stilbite, heulandite, epistilbite, calcite, and chalcedony.

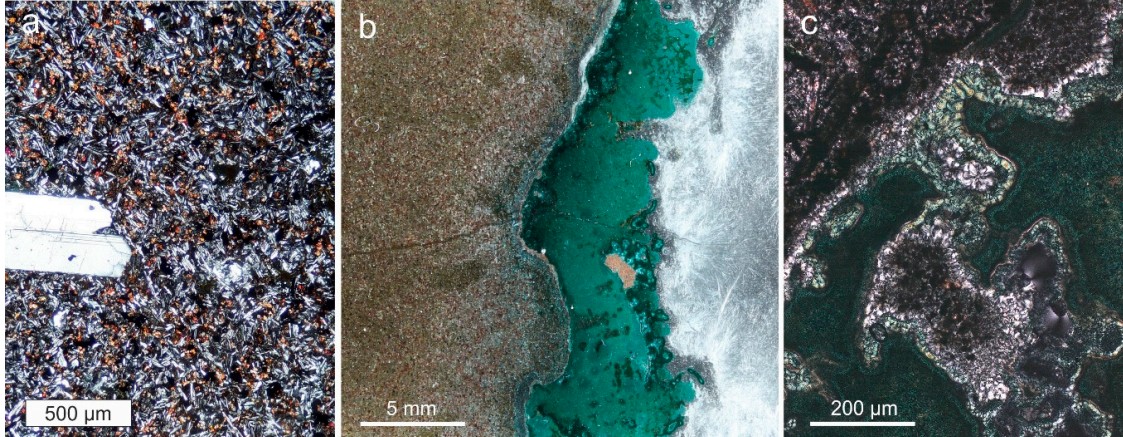

**Figure 5.** Micrographs showing the characteristics of the basalt in the core line of Savda and its interface with secondary mineralization. (**a**) Transmitted light micrograph (crossed polars) showing fine-grained basalt groundmass of lath-shaped plagioclase microlithes and clino-pyroxene with large plagioclase phenocryst. (**b**) Cross-section of the interface between basalt and secondary mineralization in a cavity consisting of green celadonite and needle-like zeolite crystals (mordenite) in chalcedony matrix. (**c**) Transmitted light micrograph (crossed polars) of collomorph celadonite with intercalated zeolites (stilbite/heulandite).

In Savda, most cavities in the core line were filled with secondary minerals. Figure 5 shows the interface between the basaltic host rock and the cavity mineralization. Visual observations suggested the following general mineralization sequence in the big cavities of the flow core line: phyllosilicates/clay minerals (wall lining and as SFF)–calcite I–chalcedony–calcite II–heulandite–stilbite–calcite III–apophyllite. Mineralization in the big cavities, even in neighboring cavities, varied concerning the number of species, their frequencies, and associations. Visual observations of the cavities in the quarry indicated that the large cavities were completely filled by fluids during the time of precipitation because all minerals crystallized all over the cavity walls without any specific crystallization direction.

*4.2. Secondary Minerals (According to Their Precipitation Sequence)*

### 4.2.1. Clay Minerals

Iron and magnesium containing phyllosilicates (clay minerals such as celadonite or smectites), which always represent the first generation of minerals in the sequential filling of vesicles [8,11], covered the inner walls of cavities in a concentric layer (Figure 6) and formed complex filaments. Celadonite and saponite were analyzed by XRD as wall-lining clay minerals. Both were extremely Fe-rich and had iron contents up to 19.7 wt. % (saponite) and 26.6 wt. % (celadonite), respectively (compare Table 4). Celadonite (according to master list of the International Mineralogical Association —IMA), the composition is covered by the isostructural minerals aluminoceladonite, celadonite, chromceladonite, ferroaluminoceladonite, and ferroceladonite [60]) and smectites are low-temperature minerals formed at temperatures below 100 °C in hot springs between 50–100 °C [61,62]. Both minerals did not occur in the same cavities but in different cavities of the same flow core zone.

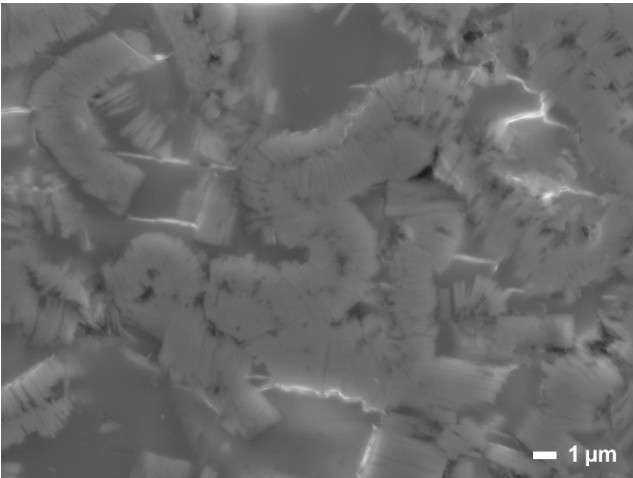

**Figure 6.** SEM micrograph showing the heterogeneous microtexture of Fe-rich celadonite (17.69 wt. % Fe according to EDX analysis) with agglutinated platelets forming worm-like structures.

4.2.2. Subsurface Filamentous Fabrics (SFFs)

Hofmann [63,64] defined the term subsurface filamentous fabrics (SFFs) as microscopic to macroscopic mineral fabrics that resulted from the precipitation of minerals on a substrate of filamentous (thread-like) geometric units in subterraneous environments. The mineralization sequence started with the precipitation of clay minerals and the formation of SFF.

SFFs occurred in the open space of large cavities. Easily recognizable were forms, commonly described as pseudo stalactites. SFFs showed a wide variety of macroscopic morphologies (Figure 7) and sizes from several centimeters up to 10 cm (extreme cases up to 100 cm in the Jamner area). SFF, to several cm in length according to Figure 5a–d, were easy to observe and not very frequent. In contrast, short SFFs or embedded SFFs in a silica layer, as shown in Figure 7d–f, were very common.

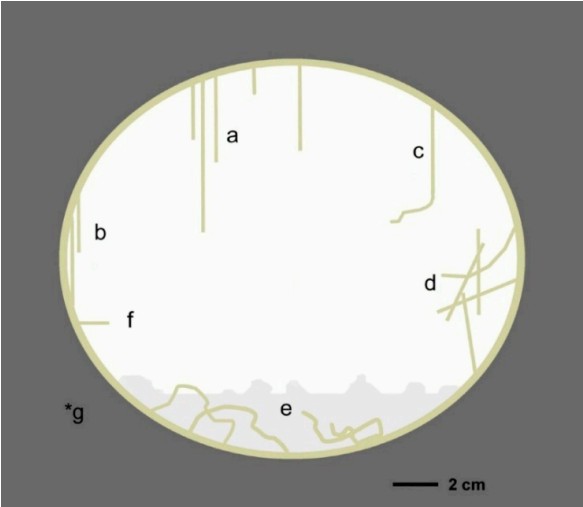

**Figure 7.** Schematic sketch of a cavity in the lava flow core line with a wide variety of macroscopic morphology of SFF: (**a**) Gravity-controlled linear SFF growing downwards from the cavity ceiling into the free space. (**b**) Gravity-controlled linear SFF growing downwards and connected with the side walls. (**c**) SFF starting with gravity-controlled growth and continuing as curved shapes. (**d**) Intergrowths of curved SFF without any specific orientation forming stable frameworks. (**e**) Zone of intergrowths of helical SFF at the cavity bottom, commonly completely embedded in chalcedony of irregular surface. (**f**) Horizontal SFF with lengths of 1 to 3 cm. Free, upwardly developed SFF have not been observed. *g unaltered whole rock in the lava flow core line.

The outer shape was characterized by a round cross-section or an irregular form as a result of several SFFs grown together or as ribbon-like mats and strands. It was remarkable that the external diameter of the SFF in general did not change over the total length, which varied from a few millimeters up to more than 100 cm. According to the texture, the SFF could be subdivided into two general groups. SFF of the first group developed a core, which was embedded directly in chalcedony and formed predominantly ribbon-like mats (Figure 8a,b).

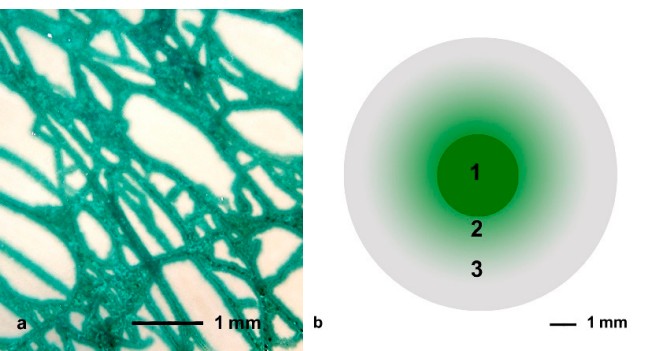

**Figure 8.** SFF mat directly embedded in chalcedony: (**a**) Micrograph in transmitted light. (**b**) Schematic cross section of an SFF: 1 = core SFF, 2 = chalcedony with inclusions of clay minerals, and 3 = pure $SiO_2$.

The second group was characterized by SFF (Figure 9a,b) with three different major zones: (a) central SFF, (b) surrounding zone of an empty tube, occasionally with fibrous mordenite, and (c) an external zone consisting of a thick $SiO_2$ layer (0.5 to 20 mm) and occasionally embedded mordenite. The central SFF consisted of an innermost core SFF with diameters between 5 and 50 μm and a surrounding zone of different clay minerals. The color varied between intense green, olive-green, and brown to nearly black. The surface texture was characterized by joint spherical aggregates with a common uniform surface. If mordenite was precipitated between the central SFF and the $SiO_2$ crust, the surface of the central SFF was partly covered by adherent fibrous mordenite crystals. SFFs of the second group formed predominantly linear to curved fabrics. The cross-section of the core SFF showed a radiating arrangement of curved, flake-like minerals without any characteristic crystal shape, even at high magnification.

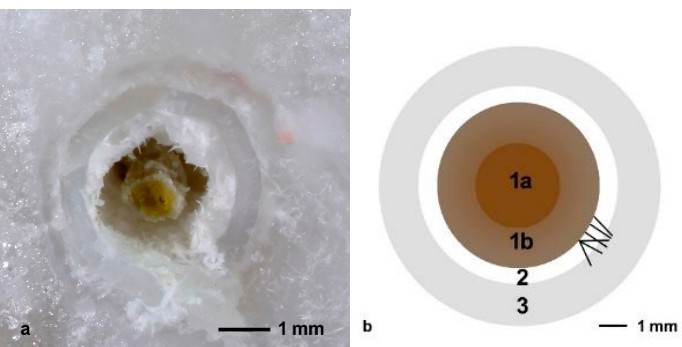

**Figure 9.** Central SFF with surrounding zone of an empty tube and an external zone consisting of a 0.5 to 20 mm-thick chalcedony layer: (**a**) Micrograph of an SFF cross section. (**b**) Schematic cross section of an SFF: 1a = core SFF (Fe rich clay minerals), 1b = outer zone of the central SFF (Fe containing clay minerals), 2 = surrounding zone of an empty tube (occasionally containing fibrous mordenite), and 3 = external zone consisting of a 0.5 to 20 mm-thick $SiO_2$ layer.

Based on the results of local chemical analyses (Table 4), two mineral groups could be distinguished in the central SFF. Green core SFF consisted of celadonite and glauconite, whereas dark brown to black core SFF consisted of smectite (e.g., Fe-rich saponite). The compositions of the wall-lining clay minerals

were identical to that of the SFF in the same cavity. It was significant that Fe-rich members dominated in the SFF.

**Table 4.** Detected clay minerals in SFF and related Fe contents (SEM-EDX analyses) and results of XRD analyses for wall-lining clay minerals in different samples.

| Sample | Occurrence | Color | Method | Mineral | Fe [wt%] |
|---|---|---|---|---|---|
| JAL-SM-01 | filament | brown | SEM-EDX | saponite | 4.9–7.4 |
| JAL-05 | filament | black | SEM-EDX | saponite | 17.1–19.7 |
| JAL-11 | filament | brown | SEM-EDX | saponite | 2.5–5.6 |
| JAM-01 | filament | olive | SEM-EDX | glauconite | 12.7–16.7 |
| JAM-10 | filament | green | SEM-EDX | celadonite | 24.2–26.6 |
| JLN-01 | filament | green | SEM-EDX | celadonite | 16.6–22.5 |
| JAL-12A | wall lining | green | XRD | celadonite | - |
| JAL-13A | wall lining | green | XRD | celadonite | - |
| JAL-24 | wall lining | brown | XRD | saponite | - |
| JAL-27 | wall lining | brown | XRD | saponite | - |
| JAL-CHO | wall lining | green | XRD | celadonite | - |

A zone of an empty tube could frequently be observed between the central SFF and the external zone (Figure 9a). Mordenite crystallized occasionally as fibrous crystals in the empty tube of the central SFF. In other cases, mordenite was directly overgrown and embedded in silica. The external silica zone consisted of a sequence of amorphous opal, crypto-crystalline chalcedony, and macrocrystalline quartz. The silica zone commonly enclosed several SFFs and developed a more or less round cross-section with proceeding precipitation.

### 4.2.3. Calcite

Three generations of calcite occurred in the big cavities in Savda. The first-generation calcite I crystallized directly on the clay mineral assemblage. It formed unusual twinned and distorted crystals, commonly up to 15 cm in length, and were always overgrown by a 0.5 to 20 mm thick chalcedony layer. Twinned crystals often showed a multiphase growth history resulting in scepter shapes. Calcite I twins occurred without any specific orientation and grew over the entire cavity wall. Since all calcite twins were overgrown with chalcedony, it was necessary to remove the crust at a few specimens to study the morphology. The crystals already showed an elongated, bladed morphology in an early precipitation stage with a pseudoprismatic habit (Figure 10a). The primary habit consisted of the distorted ditrigonal scalenohedron $\{21\bar{3}1\}$. The crystals were twinned along the twinning plane $(0\bar{1}12)$. The variable, elongated, pseudoprismatic habit was the result of distortion of the planes $(3\bar{1}21)$ and $(\bar{3}2\bar{1}1)$.

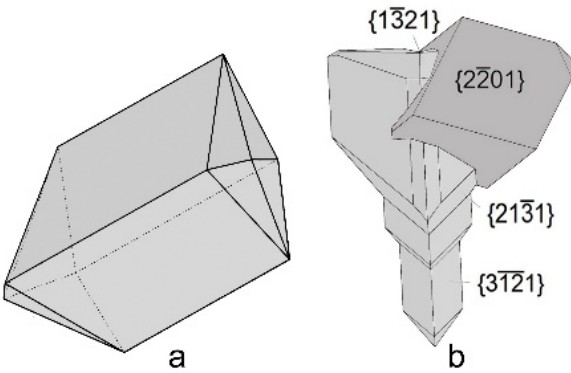

**Figure 10.** Morphology of calcite. (**a**) Distorted elongated calcite I twin. (**b**) Multiphase calcite I scepter twin with late rhombohedra.

The habit could be influenced by additional forms such as different rhombohedra and scalenohedra (Figure 10b). Since the calcite twins formed in a solution, it was assumed that they were growth twins. Changing physico-chemical conditions during mineralization resulted in a multiple scepter habit. Calcite scepters occurred commonly in the same cavities like the filaments and were overgrown by the same chalcedony layer. Calcite II of the second generation formed colorless to light yellowish rhombohedra, commonly as intergrown individuals. Calcite III of the third generation from Savda developed golden-yellow, transparent crystals up to 10 cm in size. The habit was composed of different rhombohedral forms, predominantly $\{30\bar{3}5\}$ and $\{80\bar{8}7\}$. Characteristically, intergrowth resulted in aggregates associated with chalcedony, pink stilbite, and reddish heulandite as well as colorless to green apophyllite.

Based on eutectic melting of ice $T_e$(ice) between –40 and –27 °C, fluid inclusions (FIs) in all host minerals (calcite I, calcite II, calcite III, chalcedony, and apophyllite) consisted of $H_2O$ + NaCl ± KCl ± $MgCl_2$ chemistry. $T_e$(ice) was not always easy to observe, and rough estimates are given in Table 5. Final melting temperatures of ice $T_m$(ice) from –1.5 to 0 °C indicated low salinity content. Resulted large ranges in homogenization temperatures (i.e., $T_h$: minimum entrapment temperature conditions) came from postentrapment physical changes in shape (necking-down). This meant that original fluid-filled cleavage cracks modified into individual isolated inclusions close to their initial primary entrapment temperature conditions [65]. Those necked FIs were additionally characterized by high variations in liquid/vapor ratios, which occurred frequently in the investigated samples.

FIs in calcite I and II entrapped in JAL-31/JAL-32 and JAL-32/JAL-33, respectively, were tube-like inclusions or were arranged along planes with elongated and irregular shape characteristics (Figure 11a,b). They consisted of a 2-phase at room temperature (RT) with high variations in liquid/vapor ratios. In many cases, FIs showed bubble nucleation after the first cool run. Homogenization temperatures ($T_h$) were scattered between 101.2 and 157.6 °C in calcite I and between 94.1 and 173.7 °C in calcite II. Large, tube-shaped FIs in calcite II (JAL-33) yielded $T_h$ from 130.1 to 173.7 °C adjacent to primary single FIs with lower $T_h$ between 116.7 and 123.3 °C. As result of necking, calculated densities scattered between 0.96 and 0.92 g/cm$^3$ in calcite I and between 0.96 and 0.89 g/cm$^3$ in calcite II.

FIs in calcite sample JAL-33 defined large-sized (≤100 μm) 2-phase inclusions surrounded by small satellites, suggesting local decrepitation (Figure 11c). However, FIs in JAL-2 calcite III indicated a primary origin and homogenized to liquid between 167.0 and 254.2 °C, indicative of densities from 0.91 to 0.79 g/cm$^3$, respectively.

FIs in sample JAL-16B were investigated for calcite and quartz aggregates. FIs in calcite III appeared mostly as necked tube inclusions parallel to twinning, suggesting a primary growth character like tube FIs in calcite II from sample JAl-33 (Figure 11d). Homogenization temperatures ranged from 144.2 to 246.4 °C, corresponding to a density range from 0.93 to 0.80 g/cm$^3$, respectively.

FIs in quartz showed $T_m$(ice) at –0.5 °C and precise homogenization temperatures to liquid between 140.0 and 142.5 °C. This yielded densities around 0.93 g/cm$^3$. Thus, homogenization to liquid at 140 °C was almost close to the high-density FIs in calcite II.

**Table 5.** Summary of fluid inclusions properties. $T_e$(Ice) = temperature of eutectic melting of ice; $T_m$(Ice) = temperature of final melting of ice; $T_h$ (L) = temperature and homogenization into liquid; n = number of fluid inclusions measured; n.o. not observed; and n.r. not reproducible.

| Sample | Host | n | System | Phase | $T_e$(Ice) [°C] | $T_m$(Ice) [°C] | $T_h$ (L) [°C] | p [g/cm³] |
|--------|------|---|--------|-------|-----------------|-----------------|----------------|-----------|
| JAL-16B | Qtz | 15 | $H_2O$-NaCl ± KCl ± $MgCl_2$ | L + V | ~ −34 | −0.5 to 0.0 | 140.0 to 142.5 | 0.93 |
| JAL-31 | Cal I | 15 | $H_2O$-NaCl ± KCl ± $MgCl_2$ | L + V | −37.9 to −40 | −1.5 to −0.5 | 101.2 to 157.6 | 0.96 to 0.92 |
| JAL-32 | Cal I | 15 | $H_2O$-NaCl ± KCl ± $MgCl_2$ | L + V | ~ −35 | −1.0 to 0.0 | 105.0 to 134.3 | 0.96 to 0.94 |
| JAL-32 | Cha | 5 | $H_2O$-NaCl ± KCl ± $MgCl_2$ | L + V | n.o. | −0.5 to 0.0 | n.r. | —— |
| JAL-2 | Cal III | 12 | $H_2O$-NaCl ± KCl ± $MgCl_2$ | L + V | ~ −27 | −0.6 to −0.5 | 167.0 to 254.2 | 0.9 to 0.7 |
| JAL-32 | Cal II | 12 | $H_2O$-NaCl ± KCl ± $MgCl_2$ | L + V | −32.0 to −39.0 | −0.4 to 0.0 | 94.1 to 149.0 | 0.96 to 0.92 |
| JAL-33 | Cal II | 18 | $H_2O$-NaCl ± KCl ± $MgCl_2$ | L + V | ~ −30 | −0.1 to 0.0 | 116.7 to 173.7 | 0.95 to 0.89 |
| JAL-16B | Cal III | 25 | $H_2O$-NaCl ± KCl ± $MgCl_2$ | L + V | −35.0 to −40.0 | −0.8 to −0.2 | 144.2 to 246.4 | 0.93 to 0.80 |
| JAL-13D | Apo | 20 | $H_2O$-NaCl ± KCl ± $MgCl_2$ | L + V | −29.0 to −31.0 | −0.1 to 0.0 | 141.1 to 222.2 | 0.94 to 0.85 |

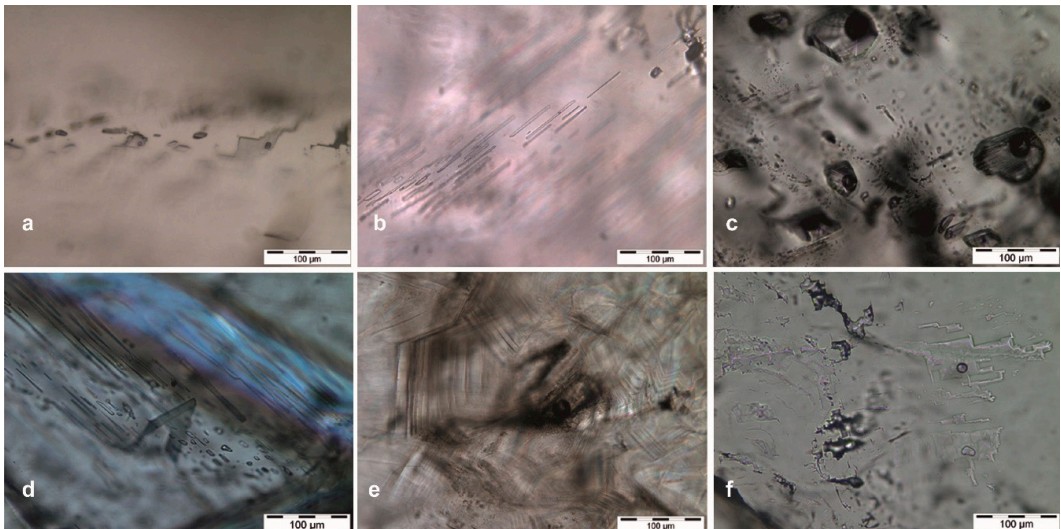

**Figure 11.** Fluid inclusion textures from investigated samples. (**a**) 2-phase FIs with irregular shape in calcite I (sample JAL-31). (**b**) Tube-like FIs parallel to twinning (calcite II, sample JAL-33). (**c**) Large 2-phase single FIs in calcite II (sample JAL-33). (**d**) Tube-like FIs parallel to twinning in calcite III (sample JAL-16B). (**e**) 2-phase inclusion entrapped between grain boundaries in chalcedony (sample JAL-32). (**f**) Irregular shaped 2-phase FIs in apophyllite (sample JAL-13D).

Differences were also noted for isotope compositions of calcite generations (Table 6). Calcite I twins (Cal I) had rather homogeneous isotope compositions, which could be explained by growth in a uniform temperature and fluid milieu. The low $\delta^{13}C$ values of –10.0 to –10.9 ‰ suggested that the carbon source for the formation of calcite was dissolved inorganic carbon (DIC) typically derived from mixed organic/inorganic carbon sources [66]. Although the $\delta^{13}C$ values of calcite II (Cal II) were still relatively constant between –13.6 and –15.1 ‰, in contrast to the scattering $\delta^{18}O$ values, they were much lower than those for the Cal I generation. Clearly, this showed the dominantly organic carbon origin for the DIC. The $\delta^{18}O$ values scattered widely between +19.6 to +27.9 ‰, requiring higher $\delta^{18}O$ values for the mineral-forming fluids of the Cal II generation at comparable temperatures.

**Table 6.** Stable isotope values (in ‰) of the two calcite generations (Cal I) and (Cal II) from different samples at Savda.

| Sample | Mineral | $\delta^{13}C$ VPDB | $\delta^{13}C$ VPDB | $\delta^{18}O$ SMOW |
|---|---|---|---|---|
| JAL-31 | Cal I | −10.00 | −15.30 | +15.13 |
| JAL-31 | Cal I | −10,28 | −15.32 | +14.91 |
| JAL-32 | Cal I | −10,95 | −15.55 | +14.88 |
| JAL-32 | Cal I | −10,94 | −15.39 | +15.05 |
| JAL-32 | Cal II | −13.61 | −6.84 | +23.86 |
| JAL-32 | Cal II | −13.68 | −8.06 | +22.60 |
| JAL-33 | Cal II | −15.11 | −10.96 | +19.61 |
| JAL-33 | Cal II | −13.81 | −2.93 | +27.89 |

Investigations by CL microscopy and spectroscopy revealed that both homogeneously appearing calcite generations exhibited orange CL (activated by $Mn^{2+}$ at $Ca^{2+}$ lattice position) and distinct growth zoning (Figure 12a,b,g,h). Remarkable changes in the physico-chemical conditions during crystallization were not detectable. However, there was a contrast in the luminescence intensity of calcite generations I and II. Calcite II was almost nonluminescent (i.e., it had a very low concentration of defects and trace elements indicating different growth conditions).

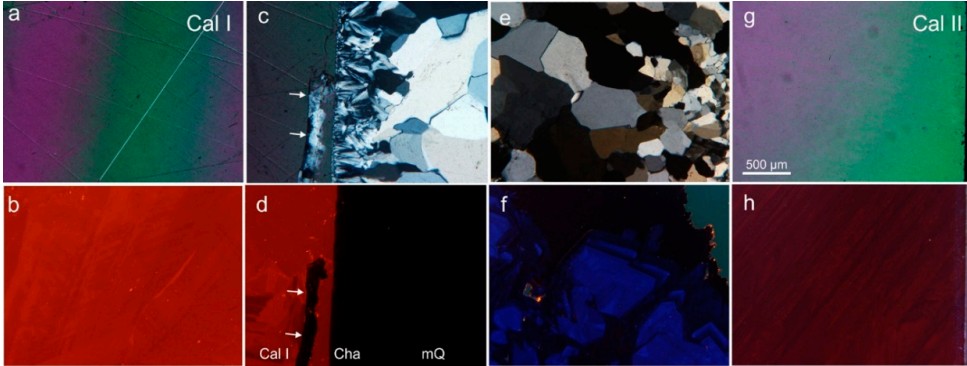

**Figure 12.** Micrographs of secondary minerals in polarized light (upper) and cathodoluminescence (lower). (**a**,**b**) Calcite of the first-generation Cal I showing growth zoning under CL—the orange CL is caused by $Mn^{2+}$ activation. (**c**,**d**) Transition zone Cal I-$SiO_2$ mineralization starting with spherulitic chalcedony (Cha)—the arrows point to corrosion of Cal I by $SiO_2$. (**e**,**f**) Macrocrystalline quartz (mQ) showing remarkable growth zoning under CL—the characteristic transient blue CL points to hydrothermal origin. (**g**,**h**) Homogeneous calcite of the second-generation Cal II with regular growth zoning under CL—the $Mn^{2+}$-activated CL intensity is much lower than in Cal I (note that exposure time for b,d is 1 s and for f,h 20 s.

The transition between calcite I and the following silica precipitation is documented in Figure 12c,d. It started with the crystallization of spherulitic chalcedony accompanied by dissolution and replacement of carbonate. Macrocrystalline quartz developed with ongoing crystallization of $SiO_2$. The quartz showed short-lived blue CL with distinct growth zoning (Figure 12e,f), which was characteristic for hydrothermal conditions and confirmed the results of fluid inclusion studies.

### 4.2.4. Silica Phases

Chalcedony occurred as overgrowth on all previously formed clay minerals, zeolite I, and calcite I. It was assumed that chalcedony formation ran through several structural states of $SiO_2$ with amorphous silica as the first solid phase [67]. In several samples, multiple crystallization sequences were detectable that developed from microcrystalline silica to macrocrystalline quartz (Figure 13a). Sometimes, precipitation of silica had encrusted crystals of mordenite or clay minerals (Figure 13b). The thickness of the crust varied between approximately 1 and 20 mm. The filaments described above were generally completely protected by a chalcedony crust. Filaments without chalcedony crust were very rare.

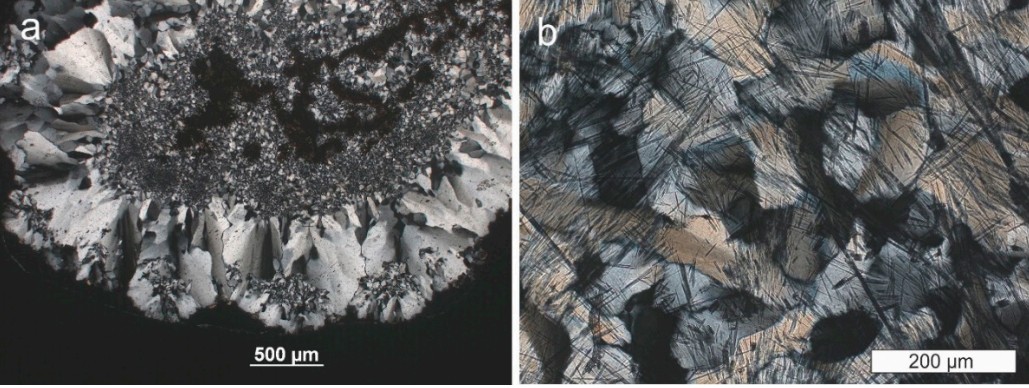

**Figure 13.** Transmitted light (crossed polars) micrographs showing typical appearance of $SiO_2$ phases. (**a**) Filament core encrusted by different silica layers showing grain coarsening from micro-crystalline chalcedony (center) to macro-crystalline quartz (margin). (**b**) Matrix of coarse-grained quartz with needle-like inclusions of mordenite.

Fluid inclusions in chalcedony were very rare and were only observed in sample JAL-2. Inclusions occurred at grain boundaries (Figure 11e) and showed high variations in liquid/vapor proportions. $T_e$(ice) was not observed, and $T_m$(ice) was comparable to calcite generations from –0.5 to 0 °C. Homogenization to liquid was observed near 120 °C and to vapor between 145 and 165 °C. However, homogenization temperatures ($T_h$) were not reproducible, suggesting leakage of the inclusions during heating. Therefore, fluid inclusions were not calculated but listed in Table 5 for consideration.

### 4.2.5. Feldspar, Zeolites, and Powellite

Stilbite-Ca, heulandite/clinoptilolite-Ca, and plagioclase were discovered by SEM/EDX (Figure 14, Table 7) as well-developed idiomorphic crystals 0.5 mm in size (maximum) and were overgrown by chalcedony in a zone of the filaments, originally at a distance of 1 to 4 cm to the host rock. Secondary feldspar was only detected in sample JAM-10, so that general conclusions concerning the conditions of formation could not be drawn. However, the position of the feldspar crystals and their morphology clearly indicated a formation mechanism by precipitation from a solution rather than as alteration products of primary feldspar in the host rock matrix. Local chemical analysis of the subhedral feldspar crystals resulted in a composition of 3.62 Ca, 0.66 Na, and 0.22 K (at. %) indicating plagioclase.

In addition, mordenite was detected in several samples (Figure 14b). The fibrous crystals of mordenite, the zeolite with the highest $SiO_2$ content in Savda, grew on the clay minerals of the wall lining layer and central filaments consisting of clay minerals, and they were overgrown by chalcedony/quartz (Figure 13). Accordingly, mordenite was precipitated on clay minerals after other minerals such as plagioclase (Figure 14d) and heulandite/clinoptilolite (Figure 14c) or stilbite (Figure 14a) were formed. In sample Jam 10, the following mineralization sequence was established by visual observations: celadonite filamentous fabric–plagioclase–heulandite/clinoptilolite–stilbite–unknown mineral as spherical aggregate–mordenite–dissolution of the spherical mineral–chalcedony (Figure 14c). Zeolites I of the first generation were characterized by a low K content. EDX analysis did not allow to clarify whether the characteristically shaped crystals were heulandite or clinoptilolite.

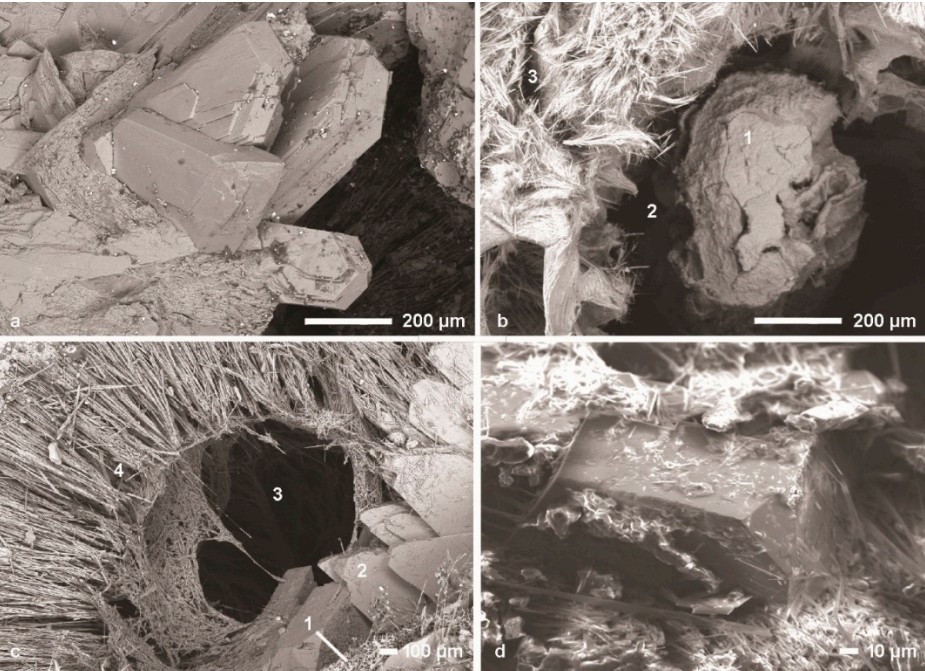

**Figure 14.** SEM micrographs showing the distribution of zeolites and feldspar: (**a**) Sample JAM-10: euhedral stilbite-Ca crystals. (**b**) Sample JAL-SM 01: 1-central SFF, 2-empty tube, and 3-mordenite. (**c**) Sample JAM-10: 1-filamentous fabric, 2-euhedral heulandite/clinoptilolite, 3-empty spherical space, and 4-fibrous mordenite. (**d**) Sample JAM-10: subhedral plagioclase with mordenite.

**Table 7.** EDX data of stage II zeolites and plagioclase of sample JAM 10—stilbite-Ca, mordenite, heulandite-Ca, plagioclase; the slight deviations from the theoretical stoichiometric compositions are due to intimate intergrowths with other mineral phases (e.g., mordenite in the $SiO_2$ matrix).

| Mineral | Stilbite-Ca | Mordenite | Heulandite-Ca | Plagioclase |
|---|---|---|---|---|
| | at [%] | at [%] | at [%] | at [%] |
| O | 73,94 | 50,06 | 71,16 | 61,58 |
| Si | 16,63 | 40,09 | 19,39 | 26,93 |
| Al | 5,09 | 6,91 | 6,15 | 6,99 |
| Ca | 2,99 | 1,61 | 2,42 | 3,62 |
| Na | 1,34 | 1,24 | 0,68 | 0,66 |
| K | 0,01 | 0,08 | 0,20 | 0,22 |
| Total | 100,00 | 100,00 | 100,00 | 100,00 |

A second generation of zeolites was precipitated on chalcedony or calcite II. XRD analyses confirmed earlier investigations [18,47] concerning the presence of heulandite-Ca and stilbite-Ca. The color of heulandite-Ca varied from rarely white to reddish and occasionally green. The green color was probably caused by inclusions of celadonite. In contrast to heulandite occurring in the vesicles and amygdales of the upper flow zone as millimeter-sized wall linings, the mineral formed well-developed, individual, thick-tabular crystals of up to 10 cm in size within the cavities of the flow core zone in Savda. The usual habit displayed the dominant form {010} and the less dominant forms {100}, {011}, and {110} [47]. Heulandite in Savda typically formed intergrowths of several individual crystals to fan-shaped groups. The large heulandite (stage III) crystals from Savda were generally deposited on chalcedony/quartz. Comparable large heulandite crystals were not observed from other outcrops in the DVP.

Salmon-pink stilbite-Ca crystals, identified by XRD, were also very common and reached sizes of up to 5 cm. As in numerous other outcrops in the DVP, stilbite in Savda formed single crystals deposited on chalcedony or calcite. The forms {010}, {001}, and {111} predominated, although {111} may be partly supplemented or totally replaced by {001} (47). Stilbite rarely developed ideally shaped individual crystal—they tended to form curved, plate-like intergrowths. Although stellerite, the Ca-rich endmember of the stilbite subgroup, belonged to the orthorhombic system and stilbite to the monoclinic, crystals of the two species nevertheless showed the same shapes since stilbite formed pseudo-orthorhombic crystals. While stellerite was determined in samples of several outcrops in the DVP, crystals from Savda were confirmed as stilbite-Ca. Other zeolites that commonly occurred in the DVP, such as laumontite, mesolite, scolecite, and chabazite, did not occur in Savda.

Moreover, powellite ($CaMo_4$), the rare calcium molybdate, was found in Savda showing euhedral, tetragonal dipyramidal crystals up to 3 cm in size. The nonuniform color ranged from almost colorless to honey-yellow. Chemical analyses of powellite from Nashik [68,69] showed a tungsten content of only 0.5–2.5 wt. %. Higher tungsten content was found in the darker samples. The samples from Savda showed a pale-yellow color, indicating a tungsten content of 1–1.5 wt. %. Powellite from Savda showed {123}, {311}, and {111} as the dominant forms. The striations on the faces were characteristic. Actually, the form {123} represented only an apparent plane surface; it was approximated by alternations of the forms {112}, {235}, {156}, and {011}, which were responsible for the striations (47). The powellite crystals formed in Savda during or after the late stage of the stilbite deposition, indicated by the observation that they were partly embedded in stilbite. Powellite was visually determined by the characteristic habit of the dipyramidal crystals and fluorescence under short-wave UV light excitation.

4.2.6. Apophyllite

Typical SEM-EDX analyses of apophyllite from samples JAL-13D and JAL-16B are given in Table 8. The individual crystals were more or less homogeneous and closely approached the fluor-apophyllite end-member, even when small amounts of $Al_2O_3$ (<0.1–0.39 wt. %) and Na (0.05–025 wt. %) were

present. In XRD analyses, a double peak was present at the beginning of the spectra, indicating a hydroxyl-apophyllite component. The measured molar K/Ca ratios were in the range of 0.238–0.259 and fit to those of ideal fluor-apophyllite (0.244) [70].

**Table 8.** Chemical composition (SEM-EDX analyses in wt. %) of apophyllite from samples JAL-13D and JAL-16B. Formula is calculated on the basis Si = 8. Theoretical $H_2O$ content was calculated based on stoichiometry of the formula with 8 $H_2O$. The F-bearing position is completely filled by F for most analysis, only some contain minor amounts of OH, which was neglected.

| Label | JAL-13D-8 | JAL-13D-9 | JAL-13D-10 | JAL-13D-11 | JAL-13D-12 | JAL-13D-13 | JAL-16B-6 | JAL-16B-7 | JAL-16B-8 | JAL-16B-13 | JAL-16B-14 | JAL-16B-16 | JAL-16B-17 |
|---|---|---|---|---|---|---|---|---|---|---|---|---|---|
| $SiO_2$ | 52,76 | 52,26 | 52,78 | 52,85 | 52,46 | 52,08 | 52,52 | 52,78 | 52,75 | 52,88 | 52,80 | 52,72 | 52,25 |
| $Al_2O_3$ | 0,18 | 0,33 | 0,38 | 0,34 | 0,32 | 0,28 | 0,23 | <0.1 | 0,19 | 0,25 | <0.1 | 0,24 | 0,39 |
| CaO | 23,95 | 23,70 | 23,83 | 23,75 | 23,78 | 23,80 | 23,82 | 23,76 | 23,96 | 24,05 | 23,90 | 23,81 | 23,73 |
| $K_2O$ | 4,88 | 4,99 | 4,89 | 4,82 | 4,76 | 4,77 | 5,19 | 4,99 | 5,09 | 4,97 | 5,15 | 4,96 | 4,97 |
| $Na_2O$ | 0,18 | 0,21 | 0,19 | 0,25 | 0,21 | 0,19 | 0,09 | 0,09 | 0,05 | 0,09 | 0,14 | 0,09 | 0,24 |
| F | 2,13 | 2,07 | 2,12 | 2,08 | 2,03 | 2,10 | 2,04 | 2,09 | 1,94 | 1,93 | 2,13 | 1,99 | 2,18 |
| $H_2O^*$ | 15,81 | 15,66 | 15,81 | 15,83 | 15,72 | 15,60 | 15,73 | 15,81 | 15,80 | 15,84 | 15,82 | 15,79 | 15,65 |
| Sum | 99,88 | 99,21 | 100,00 | 99,92 | 99,27 | 98,82 | 99,62 | 99,52 | 99,77 | 100,00 | 99,94 | 99,60 | 99,42 |
| | | | | | | $H_2O^*$ = calculated | | | | | | | | |
| Si | 8,000 | 8,000 | 8,000 | 8,000 | 8,000 | 8,000 | 8,000 | 8,000 | 8,000 | 8,000 | 8,000 | 8,000 | 8,000 |
| Al | 0,032 | 0,059 | 0,068 | 0,060 | 0,057 | 0,051 | 0,042 | - | 0,034 | 0,044 | - | 0,043 | 0,071 |
| Ca | 3,891 | 3,886 | 3,869 | 3,851 | 3,885 | 3,918 | 3,887 | 3,859 | 3,894 | 3,898 | 3,879 | 3,872 | 3,893 |
| K | 0,944 | 0,975 | 0,945 | 0,931 | 0,926 | 0,935 | 1,008 | 0,964 | 0,984 | 0,959 | 0,996 | 0,959 | 0,970 |
| Na | 0,053 | 0,063 | 0,055 | 0,073 | 0,063 | 0,056 | 0,027 | 0,025 | 0,015 | 0,026 | 0,042 | 0,028 | 0,071 |
| F | 1,020 | 1,001 | 1,018 | 0,997 | 0,977 | 1,019 | 0,984 | 1,000 | 0,929 | 0,923 | 1,021 | 0,954 | 1,057 |
| Sum | 13,94 | 13,98 | 13,96 | 13,91 | 13,91 | 13,98 | 13,95 | 13,85 | 13,86 | 13,85 | 13,94 | 13,86 | 14,06 |
| K/Ca | 0,243 | 0,251 | 0,244 | 0,242 | 0,238 | 0,239 | 0,259 | 0,250 | 0,253 | 0,246 | 0,257 | 0,248 | 0,249 |

Apophyllite represented the youngest mineral generation in the cavities of the Savda quarries and was commonly deposited on the chalcedony crust of the filaments or wall linings and other earlier crystallized minerals. The mostly colorless and rarely greenish crystals with the forms {110} and {101} were up to 6 cm in length. The green color of apophyllite was caused by a high vanadium (V) content of up to 3000 ppm [18,71]. Similar high V contents of apophyllite occurred at other localities in the DVP too and reflected an intensive dissolution of vanadium-bearing minerals and high V concentrations in the mineral-forming fluids. The V content of basalts in the DVP was in general in the range of 200–400 ppm (24).

Fluid inclusions in apophyllite were investigated in sample JAL-13D (Table 5). They appeared irregular in shape, flat, necked, and characterized two phase inclusions at room temperature (Figure 11f). $T_e$(ice) at about –30 °C and $T_m$(ice) between –1.0 and 0 °C characterized comparable fluid chemistries (i.e., $H_2O$ +NaCl ± KCl ± $MgCl_2$) like FIs of the investigated host minerals calcite, quartz, and chalcedony. Homogenization temperatures ranged from 141.1 to 222.2 °C, which were indicative of densities between 0.94 and 0.85 g/cm$^3$, respectively.

Rb-Sr and K-Ar Geochronology

Geochronological data were aimed at determining the timing of the formation of the secondary mineralization. However, the only suitable mineral for age dating found in the cavities was apophyllite. Apophyllite ((K,Na)Ca$_4$Si$_8$O$_{20}$(F,OH)·8H$_2$O) belonged to a group of phyllosilicates with about 5 wt. % K and high Rb/Sr ratios suitable for geochronological measurements using the K-Ar and Rb-Sr methods, respectively. For the latter method, at least one more datapoint with a low Rb/Sr ratio was necessary for age calculations. Besides the basaltic host rock, stilbite (NaCa$_2$Al$_5$Si$_{13}$O$_{36}$·14H$_2$O) and calcite (CaCO$_3$) occurring frequently within the cavities fulfilled this criterion. Both minerals were rich in Ca, which was partly substituted by Sr, and they were characterized by a low or absent K content.

Nine samples from Savda and Nashik were investigated by the Rb-Sr method (Figures 15 and 16, Table 9) and partly also by the K-Ar method (Table 10).

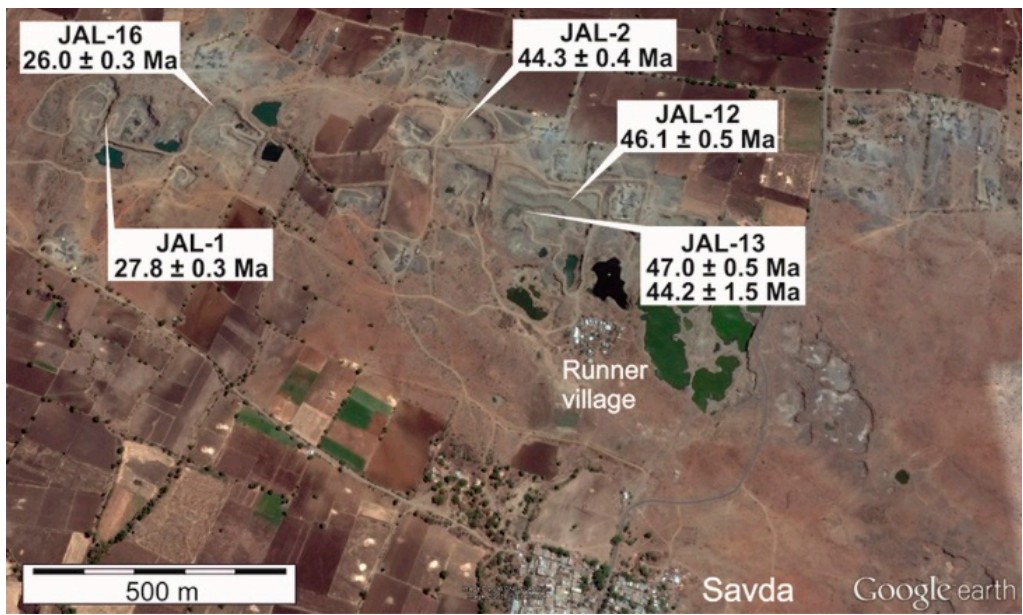

**Figure 15.** Locations of samples in the Savda quarries and determined Rb-Sr ages of apophyllite; samples JAL-1 and JAL-16B were collected in the western part of the quarries and yield ages of about 27 Ma, whereas JAL-2, JAL-12, and JAL-13 from the eastern part provide an age of about 45 Ma.

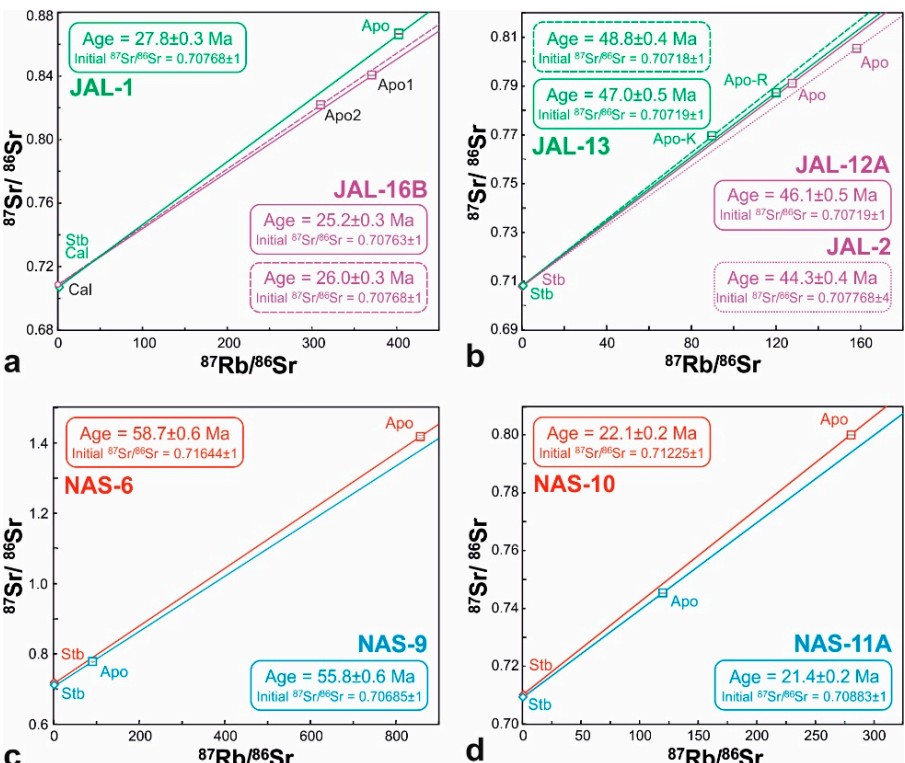

**Figure 16.** Rb-Sr age data of apophyllites sampled at different localities. Data from the western (**a**) and eastern (**b**) part of the Savda quarry complex; Paleogene (**c**) and early Miocene (**d**) ages from the Nashik area.

**Table 9.** Rb-Sr isotopic data on whole rock (WR), calcite (Cal), stilbite (Stb), apophyllite (Apo), and celadonite (Cel) from Savda and Nashik (DVP, India). Rb-Sr ages are calculated from apophyllite and corresponding calcite, stilbite, or whole rock, assuming an error of ±1% on the $^{87}$Rb/$^{86}$Sr ratio.

| Sample | Material | Rb | Sr | $^{87}$Rb/$^{86}$Sr | $^{87}$Sr/$^{86}$Sr | +/-2s$_m$ | Age | Calculated | $^{87}$Sr/$^{86}$Sr |
|---|---|---|---|---|---|---|---|---|---|
| | | [ppm] | [ppm] | | | | [Ma] (±2s) | with | Initial |
| JAL-1 | Cal III | 0,002 | 11,04 | 0,0005 | 0,706549 | 0,000004 | 28.0 ± 0.3 | Cal-App | 0.706549 ± 0.000004 |
| JAL-1 | Stb | 0,206 | 4,490 | 0,1331 | 0,707733 | 0,000005 | 27.8 ± 0.3 | Stb-App | 0.707680 ± 0.000004 |
| JAL-1 | Apo | 99,87 | 0,732 | 401,21 | 0,866312 | 0,000144 | 27.9 ± 2.8 | Cal-Stb-App | 0.7071 ± 0.0072 |
| JAL-2 | Stb | 0,059 | 5,291 | 0,0323 | 0,707788 | 0,000004 | | | |
| JAL-2 | Apo | 102,5 | 1,892 | 158,18 | 0,805454 | 0,000017 | 44.3 ± 0.4 | Stb-App | 0.707768 ± 0.000004 |
| JAL-12A | Apo | 106,8 | 2,429 | 128,31 | 0,791158 | 0,000017 | 46.1 ± 0.5 | WR-App | 0.707193 ± 0.000005 |
| JAL-13A | Cel | 293,4 | 1525 | 0,5568 | 0,708432 | 0,000004 | | | |
| JAL-13B | WR | 17,65 | 259,0 | 0,1971 | 0,707322 | 0,000004 | | | |
| JAL-13C | Apo-core | 97,26 | 3,157 | 89,688 | 0,769401 | 0,000008 | 48.8 ± 0.4 | WR-App | 0.707185 ± 0.000002 |
| JAL-13C | Apo-rim | 102,7 | 2,500 | 119,84 | 0,787254 | 0,000023 | 47.0 ± 0.6 | WR-App | 0.707190 ± 0.000002 |
| JAL-16B | Cal III | 0,202 | 10,611 | 0,0550 | 0,707653 | 0,000007 | | | |
| JAL-16B | Apo1 | 95,84 | 0,759 | 370,07 | 0,840116 | 0,000026 | 25.2 ± 0.3 | WR-App1 | 0.707633 ± 0.000007 |
| JAL-16B | Apo2 | 95,88 | 0,907 | 309,20 | 0,821699 | 0,000015 | 26.0 ± 0.3 | WR-App2 | 0.707632 ± 0.000007 |
| NAS-6 | Stb | 0,065 | 3,112 | 0,0601 | 0,716487 | 0,000005 | | | |
| NAS-6 | Apo | 161,6 | 0,584 | 857,07 | 1,417889 | 0,002596 | 58.7 ± 0.6 | Stb-App | 0.716437 ± 0.000005 |
| NAS-9 | Stb | 0,178 | 26,55 | 0,0194 | 0,710789 | 0,000005 | | | |
| NAS-9 | Apo | 113,8 | 3,373 | 98,371 | 0,783354 | 0,000014 | 55.8 ± 0.6 | Stb-App | 0.710775 ± 0.000005 |
| NAS-10 | Stb | 0,070 | 12,69 | 0,0160 | 0,712251 | 0,000008 | | | |
| NAS-10 | Apo | 175,5 | 1,826 | 280,66 | 0,800312 | 0,000010 | 22.1 ± 0.2 | Stb-App | 0.712246 ± 0.000008 |
| NAS-11A | Stb | 0,042 | 20,75 | 0,0058 | 0,708837 | 0,000003 | | | |
| NAS-11A | Apo | 93,21 | 2,198 | 123,19 | 0,746279 | 0,000014 | 21.4 ± 0.2 | Stb-App | 0.708835 ± 0.000004 |

**Table 10.** K-Ar isotopic compositions of apophyllite (Apo) from the DVP (India) from Savda/Jalgaon and Nashik. In comparison, the corresponding Rb-Sr ages are also given.

| Sample | Material | Labrat. | K | K$_2$0 | $^{40}$Ar rad | $^{40}$Ar rad | K-Ar Age | Rb-Sr Age | Difference |
|---|---|---|---|---|---|---|---|---|---|
| | | Number | [wt%] | [wt%] | [ccSTP/g] | [%] | [Ma] (± 1s) | [Ma] (± 2s) | K-Ar/Rb-Sr [%] |
| JAL-12A | Apo | 8780 | 4,036 | 4,862 | 5,6084E-06 | 35,3 | 35.4 ± 1.4 | 46.1 ± 0.5 | 23,2 |
| JAL-13C | Apo-core | 8737/2 | 3,854 | 4,643 | 6,8065E-06 | 42,5 | 44.9 ± 1.6 | 48.8 ± 0.4 | 8,0 |
| JAL-13C | Apo-rim | 8737/1 | 3,829 | 4,612 | 6,6535E-06 | 52,1 | 44.2 ± 1.4 | 47.0 ± 0.6 | 6,0 |
| JAL-16B | Apo | 8778 | 4,088 | 4,924 | 1,6500E+01 | 16,5 | 22.2 ± 1.8 | 25.2 ± 0.3 | 34,5 |
| NAS-6 | Apo | 8781 | 4,053 | 4,882 | 6,3391E-06 | 23,1 | 39.8 ± 2.4 | 58.7 ± 0.6 | 32,2 |
| NAS-10 | Apo | 8779 | 3,965 | 4,776 | 3,1475E-06 | 36,7 | 20.3 ± 0.8 | 22.1 ± 0.2 | 8,1 |
| NAS-11 | Apo | 8738 | 4,016 | 4,838 | 3,1597E-06 | 18,4 | 20.1 ± 1.5 | 21.4 ± 0.2 | 6,1 |

Samples JAL-1 and JAL-16B were collected in the western part of the Savda quarries, whereas JAL-2, JAL-12B, and JAL-13 came from the eastern part (Figure 15). Samples from the Nashik area derived from four different quarries in the Nashik area.

Rb-Sr data

In general, all apophyllites were characterized by distinct Rb and Sr contents of 95 to 175 ppm and 0.8 to 3.0 ppm, respectively. The $^{87}Rb/^{86}Sr$ ratios were in the range of 40–860. All other investigated materials showed much lower $^{87}Rb/^{86}Sr$ ratios of <0.6. Calcite of sample JAL-1 was nearly free of Rb and contained 11 ppm Sr, resulting in a very low $^{87}Rb/^{86}Sr$ ratio of 0.0005. Stilbites contained 0.04–0.2 ppm Rb and 3.1–26.6 ppm Sr, and they were characterized by $^{87}Rb/^{86}Sr$ ratios of 0.006–0.13. The investigated celadonite concentrate contained about 290 ppm Rb and more than 1500 ppm Sr, resulting in an $^{87}Rb/^{86}Sr$ ratio of 0.56. As Ca and Sr were not compatible in the structure of celadonite, the high Sr content indicated a contamination by a Ca- and Sr-rich mineral. According to X-ray diffraction of the sample, there was contamination by heulandite. Ages of the individual apophyllites were calculated with the corresponding stilbite or calcite. For some apophyllites no corresponding mineral was available. In this case, the whole rock from the direct surrounding was used knowing that it might be not in an ideal equilibrium. To grasp possible errors resulting from this uncertainty, ages of apophyllite together with celadonite from sample JAL-13a and calcite from sample JAL-1 were calculated. These data points were choose because they caused the lowest and highest age values. The deviation of the age was about ±0.5 Ma, which was more or less in the range of the given error. In summary, all determined ages scattered between 21 and 59 Ma (Table 9). An interpretation of these data is given in the discussion.

K-Ar Data

The investigated apophyllite concentrates contained 3.8–4.1 wt. % K (corresponding to 4.6–4.9 wt. % $K_2O$). The amount of radiogenic $^{40}Ar$ was in the range of 16% to 52%. The ages determined scattered between 20 and 45 Ma (Table 10). An explanation is given in the discussion.

4.2.7. Mineralization Sequence

The long-term observations in the Savda quarries and accompanied studies of the samples led to a general mineralization sequence. Clay minerals precipitated in a first stage (stage I). This stage included the early formation of the core filamentous fabrics consisting of clay minerals. Stage II was represented by a sequence of early twinned calcite I followed by crystals of less than 1 mm in size of zeolites I (heulandite, stilbite, and mordenite) and feldspar, a complete overgrowth of previous minerals by chalcedony, calcite II, and finally zeolites II with up to several centimeter-sized crystals of heulandite and stilbite. Calcite III, powellite, and as the last mineral apophyllite formed in stage III.

In fact, mineralization varied widely in different cavities, even in neighboring ones. All mineral species occurring in Savda did not completely coexist in separate cavities, indicating individual mineral assemblages. In addition, the volume/weight of precipitated mineral species was very different. Sometimes, minerals such as chalcedony, stilbite, or apophyllite formed just a thin wall lining. In other cases, stilbite or apophyllite filled the complete large cavity. SFF embedded in the chalcedony wall lining occurred frequently. The number of cavities containing large parallel and gravity-controlled SFF together with calcite generations I, II, III, and apophyllite was limited, although it was observed over all the years of quarrying. Based on the evaluation of numerous specimens, a general mineralization sequence is given in Figure 17.

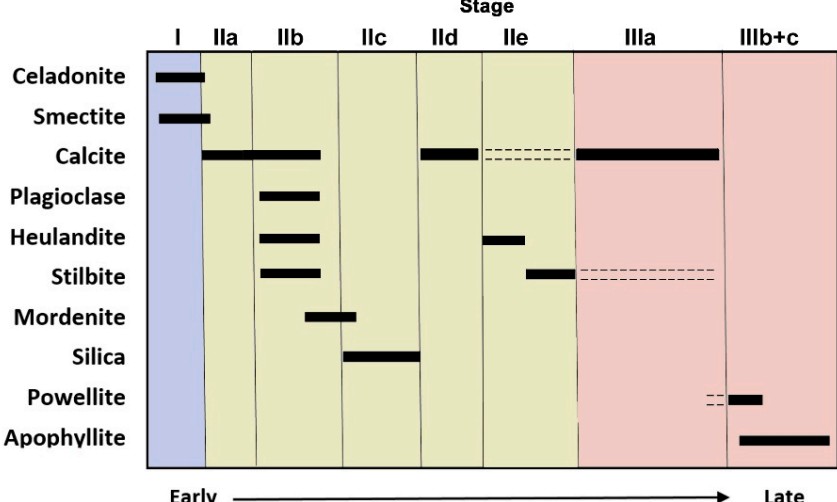

**Figure 17.** General mineralization sequence of the secondary minerals in the large cavities of the flow core zone in Savda.

## 5. Discussion

### 5.1. Alteration of Primary Minerals

The geological conditions in the central part of the DVP [41] as well as the chemical composition of secondary minerals indicated that the fluids necessary for precipitation of the secondary minerals in the vesicles, amygdales, and cavities contain dissolved elements released during the water–rock interaction. The basalt in the studied area is described as fine porphyritic tholeiitic basalt with phenocrysts of plagioclase and clinopyroxene and less abundant olivine with microcrystalline groundmass of variable size. The structure of an individual subhorizontal compound lava flow in Savda is characterized by three zones: the upper zone of the flow, which is rich in millimeter to a few centimeter large vesicles and amygdales; the central zone, which consists of a rather dense basalt containing large cavities; and the bottom zone of dense basalt with a small number of centimeter-sized vesicles. It is commonly accepted that during cooling of a fresh erupted flow in the upper zone, intensive interactions of rock with predominately meteoric and magmatic water lead to a first alteration of more reactive glass and olivine [26,27], which supply the main components for the clay minerals. The later formed zeolites are considered the result of alteration by burial metamorphism or hydrothermal events [9,11,25]. During this alteration stage, plagioclase plays an important role. The tholeiitic basalt of the studied area is characterized by a high-Ca content, based on the plagioclase composition [17,43], resulting in Ca-rich fluids forming zeolites such as heulandite-Ca, clinoptilolite-Ca, stilbite-Ca, or scolecite. Similar results were reported from other basalt provinces, although former research papers on zeolites from the DVP [8,9,14,15,17] did not explain the alteration of the primary minerals.

Neuhoff et al. [9] reported different stages of alteration and formation of secondary minerals in basaltic lavas at Taigarhorn, Northern Iceland. Low-grade alteration of the basaltic host rocks prior to deep burial resulted in the formation of celadonite and silica in the first stage. During burial (stage II), hydrolysis of olivine and basaltic glass first led to the formation of chlorite/smectite mixed-layer clays and, with increasing temperature, led to the crystallization of zeolites (scolecite/heulandite, stilbite, mordenite, and epistilbite) and replacement of plagioclase by zeolites and albite. Late hydrothermal alteration (stage III) finally overprinted the first two stages and resulted in the crystallization of large crystals of quartz, calcite, and zeolites.

Kousehlar et al. [25] studied zeolite-bearing basalts in the volcanic rocks of Kahrizak, Iran, and also found variable degrees of alteration resulting from low-grade metamorphism and subsequent hydrothermal activity. They noted that volcanic glass and olivine were replaced by clays and iron oxides. Plagioclase phenocrysts in the whole rock are affected by alteration and are partly or totally replaced

by zeolites, mafic phyllosilicates, albite, and calcite. Similar observations were made by Weisenberger and Selbeck [11] in Cenozoic flood basalts from Hvalfjördur, Iceland. Near-surface alteration with celadonite and silica precipitation in primary pores was followed by hydrolysis of olivine and basaltic glass during low-grade zeolite facies metamorphism and formation of chlorite/smectite clays and zeolites with increasing burial.

In the studied area of Savda, the whole rock in the central flow zone does not show features of strong alteration. It is assumed that the mineralizing fluids originate from the upper zone percolating through fissures into the cavities of the central zone. Alteration and mineralization of the upper zone had not been the subject of detailed investigations. However, observations of the highly altered upper zone of an individual lava flow in Savda show a high porosity and permeability, which are an important factor for low-temperature metamorphism [34] and an intensive alteration of the upper zone. It can be assumed that most components forming the secondary minerals in the big cavities in the central zone during stage I and during burial stage II are generated from the upper zone. The fluids for minerals formed later in stage III containing vanadium (green color of apophyllite) or molybdenum (formation of powellite) are believed to be generated by hydrothermal alteration from large basalt masses and supplied over longer distances.

## 5.2. Clay Minerals

Investigations of natural volcanic rocks and laboratory experiments showed that clay minerals (sheet silicates) were quantitatively the most significant alteration minerals [72–76]. The fluids percolating the volcanic rocks contain all released elements of the host rocks such as Si, Mg, Al, and Fe, which are necessary for clay mineral formation. It is argued that celadonite and smectites are formed under neutral to alkaline pH conditions, but they can also be formed in a weakly acidic environment [75]. Celadonite can form flakes of tens of nanometers in thickness and some 100 to 1000 nm in length, and it is chemically heterogeneous with varying Fe–Al–Mg–Si ratios and K content. Minerals of the smectite group, such as saponite, commonly form crystals several hundred nanometers in size. They are also chemically heterogeneous and can be subdivided into two subgroups of Fe-poor and Fe-rich smectites [76]. Iron-bearing saponite (magnesioferrosaponite) is a typical mineral representative of low-temperature alteration processes in basaltic rocks [75,76] and is experimentally formed from volcanic glass and diabase at 150 °C in the laboratory [61]. It is noteworthy that, after the formation of the clay minerals, no other secondary minerals with important concentrations of iron or magnesium have been found in the cavities of the Savda basalts [17,18].

## 5.3. Origin of Subsurface Filamentous Fabrics (SFFs)

SFFs occur in cavities, which do not meet the terminological requirements for caves [77,78]. SFFs are not stalactites, which are formed by a solution dropping from the cave ceiling [79]. SFFs formed in cavities are completely filled with fluids. While stalactites have a monomineral composition, the investigated SFF from the DVP consisted of a multiphase mineral assemblage. In contrast to stalactites, the SFF showed versatile morphologies. Moreover, the cross-section of the SFF does not change, whereas the diameter of stalactites decreases with their length. These environmental and morphological features prove that the filaments do not represent stalactites or speleothems. Amethyst-bearing geodes occurring in the flood basalts of the Arapey formation at Artigas (Uruguay) contain fabrics, explained as stalactites, consisting of agate or an agate core [80]. According to the authors, formation of these vertical and parallel stalactites cannot be explained by ripening processes. Precipitation of amethyst and calcite in the geodes at Artigas was caused by geothermal fluids by an artesian stratified aquifer and not by burial or late hydrothermal fluids with elevated temperatures. This process differs significantly from the cavities in Savda and cannot give evidence for a similar formation.

One possible process for SFF formation could be a self-organized assemblage [81]. However, processes similar to the development of a chemical garden cannot be considered because the high Na and Si content in the fluid required for such a process [82,83] is very unlikely. Solid metal salts,

which initiate the formation of a "chemical garden" when added to the mineralizing solution, are not available during the alteration of the tholeiitic basalt host rock.

Inorganic reactions of silica, metal carbonates, and metal hydroxides can promote self-organization to form complex, biomorphic structures imitating the texture and morphology of bio-minerals and primitive organisms [84]. Kellermeier et al. [85] argued that pH plays a decisive role in the morphological evolution of biomorphs. They stated that the growth of silica biomorphs can occur in a pH range of 9.3–9.8 in gels and 10.2–11.1 in solutions, while other parameters such as temperature or different cations can affect morphology. Accordingly, high pH values are necessary for successful experiments of such self-assembled biomorphs. However, the petrographic composition of the basalt at Savda and the first precipitated minerals in the big cavities (like celadonite) point to a neutral to slightly acidic pH of the early mineral-forming solutions. Based on these pH considerations, the formation of SFF by a process of self-assembling should be excluded. In addition, the complex texture of the innermost SFF with curved, flake-like minerals lacking any characteristic crystal shape does not favor a hypothesis of self-assembling.

Biomineralization, which is an accepted process in geological environments today, has to be taken into consideration for the initial stages of secondary mineralization processes in the DVP [64,86,87]. Filamentous biogenetic fabrics can occur in subsurface units of volcanic rocks [64]. Because of the general composition of the tholeiitic basalts, alteration fluids could release all necessary components from the host rocks and may have been supplied by juvenile and meteoric water and air. During cooling of a lava flow over a longer period, and before burial and cover by new lava flows, temperatures decrease by an open system at atmospheric pressure below 100 °C, which is necessary for the living conditions [64,88] of thermophile iron-oxidizing chemolithotrophic bacteria such as the helically twisted Gallionella ferruginea and the sheathed Leptothrix ochracea. Such bacteria form thread strands, with a diameter of up to 50 µm and lengths of more than 100 cm, consisting of a larger number of single accumulated strands with a characteristic morphology [63,64]. The formation of biogenic filamentous fabrics is described in three mineralization stages: (1) initial precipitation of tiny filaments (ca. 1 µm in diameter) consisting of Fe minerals and celadonite. (2) deposition of later minerals on filaments (increasing diameter up to 20–50 µm), and (3) final cementation and formation of aggregates of up to several centimeters in diameter and several tens of centimeters in length.

Detection of excess carbon and typical phases of biomineralization processes (Fe-rich celadonite and Fe-rich saponite) in the innermost parts of the SFF from Savda indicate microbial activities during early filament formation. During the process of biogenesis by iron-oxidizing bacteria, soluble $Fe^{2+}$ is oxidized to $Fe^{3+}$ as unstable ferrihydrite $((Fe^{3+})_2O_3 \cdot 0.5H_2O)$. This ferrihydrite is either transformed to iron oxides/hydroxides, or the iron compounds can be incorporated into Fe-rich minerals such as clay minerals [64,89]. In fact, the iron-containing fabrics of the innermost SFF cores do not show characteristic shapes of crystals but the morphology of bacteria filaments (Figure 18). SFFs in the large cavities in the Savda quarry represent most features of a biogenic signature [64].

Remaining constituents of organic filaments are unlikely to be detected. The SFFs from Savda have an exceptional length and diameter that are not observed elsewhere in other basalt provinces, allowing the possibility of further successful investigations of micromorphology and chemical composition. The present investigations of SFF from the DVP are predominately based on the macromorphology and first results of the texture. In consideration of Figure 7, SFFs occur in all parts of the cavities with a wide range of morphologies. Horizontal and upward orientations are limited in length or are excluded. SFFs are always directly or indirectly connected with the cavity walls. It is estimated that the weight of the first-developed bacterial filaments does not differ significantly from the density of the fluid, resulting in floating in the fluid. Because of subsequent overgrowth by ferrihydrite and the following transition to Fe, the weight of phyllosilicates such as celadonite or smectite increased. As a consequence of the heavy weight, the SFFs sank to the cavity bottom, as can be observed commonly, or connected to the cavity walls. Only gravity-controlled SFFs could be preserved as long, linear, individual SFFs. Most SFFs were formed as helical intergrowths. SFFs from the DVP are unique in

consideration of their length up to 100 cm. The genesis of bacteria filaments and the transition to Fe phyllosilicates is suggested to occur after cooling of the lava near to the surface. The occasional overgrowths caused by mordenite and the final preservation by silica occurred during burial over a long time span of several to thousands of years. It is assumed that an increasing viscosity of the solution in the cavity can support the stability of long SFFs.

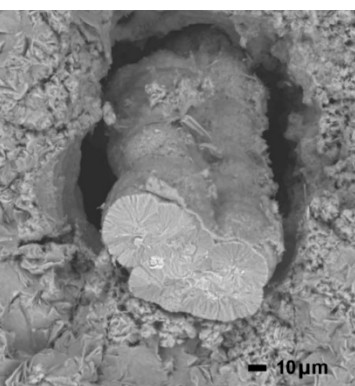

**Figure 18.** Morphology of the innermost SFF (JAL-SM-01) formed by connected spheres with a common surface; the negative contour is visible on the surrounding clay minerals, interrupted by a free space.

The micromorphology is still under investigation, but it is difficult to study considering the size and mineral composition. Longitudinal sections through the core filaments consisting of the innermost filaments and overgrowths by clay minerals could give more information. The textures of cross-sections (Figures 8 and 9) have shown a variable zoning from the center to the margin, including the empty tube. The empty tube seems to be the result of a contraction of the clay minerals forming the core filament (Figure 18). The cause for a variable contraction can probably be influenced by the clay mineral specie. Another reason could be estimated in a different protection of the core filament. In mats, as demonstrated in Figure 8, the core filament is directly embedded in silica with a thin zone of a clay–silica mixture in between. The section in Figure 9 shows, between the core filament and silica, an additional surrounding mineralization by clay minerals and mordenite. The details of such a process are still under investigation.

*5.4. Calcite*

Numerous early precipitated big calcite I crystals confirm the presence of $CO_2$ during this stage of mineralization. It is noteworthy that calcite occurs in the Savda quarries as crystals of the first generation as unusual, elongated twins only. Calcite twins are distributed randomly over the cavity walls without any specific orientation. This indicates precipitation in a cavity completely filled with fluids. The euhedral crystals grow directly on earlier deposited clay minerals.

It was assumed that the first calcite generation formed contemporaneously with or directly after the SFF. Therefore, fluid inclusion and isotope data of this calcite generation can provide general information about the physico-chemical conditions during this mineralization stage.

A morphological change during crystallization from bladed, distorted scalenohedra to rhombohedra and scepter morphology can be detected in many samples. The largest observed calcite scepter from Savda measured ca. 50 cm in length. Similar scepter forms of untwinned calcite crystals were reported from the volcanic tuffs in the Yucca mountains (Nevada, United States) [90]. According to these authors, a drastic change in the anisotropy of the growth rate is required to produce a change in the growth pattern from predominantly bladed to blocky scepters. Development of scepters seems to be a common process while oversaturation takes place and environmental conditions (e.g., pH, temperature, and chemical composition) change significantly [82]. The morphological transition of calcite I crystals from Savda quarries suggests growth over several steps, indicating a slower change of the mineralization conditions.

Figure 19 presents a comparison of oxygen and carbon isotope data of the investigated calcite I and calcite II generations from Savda with calcites from secondary mineralization in volcanic rocks of other locations as well as hydrothermal vein mineralization. In general, the isotope data of Savda calcites were similar to those of hydrothermally formed calcite from other deposits. The low $\delta^{13}$C values of Savda calcites suggested that the carbon source for the formation of calcite was dissolved inorganic carbon (DIC), typically derived from mixed organic/inorganic carbon sources. The diagram emphasizes the extreme isotopic homogeneity of calcite I generation. The scatter of the $\delta^{18}$O values of the calcite II generation, however, may point to fluctuations either in the isotopic composition of the mineral-forming fluid and/or the temperature.

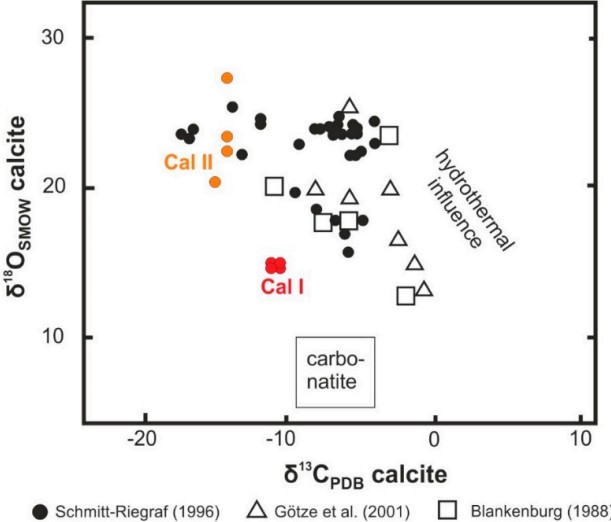

**Figure 19.** Stable isotope data of the two calcite generations Cal I and Cal II from Savda (JAL-31, JAL-32) in the $\delta^{18}$O/$\delta^{13}$C diagram compared to data from literature of calcite from secondary mineralization in altered volcanic rocks of different occurrences (Schmitt-Riegraf [91], Götze et al. [92], and Blankenburg [93]). The carbonatite field is defined according to Keller and Hoefs [94].

More information about the mineralization conditions of the two calcite generations is available by additionally considering the results of fluid inclusion studies (compare Tables 5 and 6). Figure 20 shows a plot of the isotope and fluid inclusion data of calcite I and II in relation to the fractionation curves of $\delta^{18}$O$_{PDB}$ in calcite as a function of temperature for oxygen isotope equilibration with different fluids.

Using the homogenization temperatures of fluid inclusions (Table 5) and the corresponding $\delta^{18}$O data (Table 6), the field of calcite I plots close to the fractionation curve of oxygen from marine water. However, there is no indication from the geological background of the DVP concerning marine influence during formation of the secondary mineralization. Therefore, it was most likely that the fluid for the formation of calcite I was of magmatic origin, which was probably mixed with a meteoric component.

The homogenization temperatures of fluid inclusions of the calcite I twins from Savda scattered between $T_h$ 101–157 °C (Table 5). The calcite scepter twins were probably deposited at the beginning of burial and before the silica precipitation. Based on the observation that clay minerals occurred in the early crystallized zones of the calcite twins, precipitation of the calcite twins must have started before the clay mineral assemblage ended. This indicated that a possible biomineralization of the Fe-bearing minerals had also been finished, and the $CO_2$ used for the genesis of the bacteria was returned to the fluids. Such a crystallization sequence near to the surface could explain the participation of a meteoric fluid component.

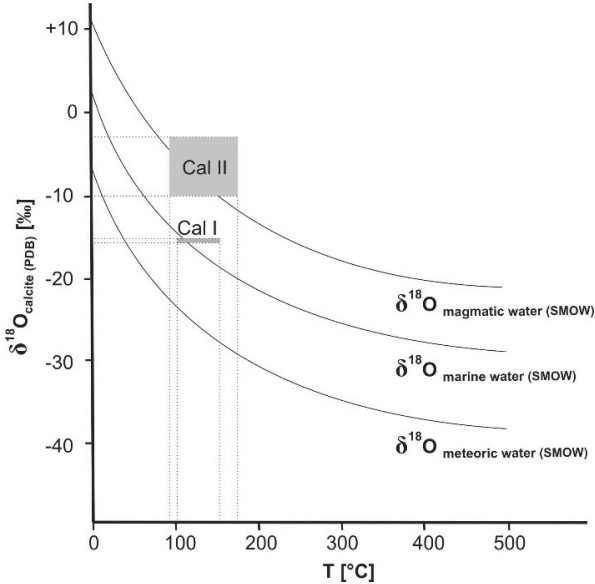

**Figure 20.** Fractionation curves of $\delta^{18}O_{PDB}$ calcite as a function of temperature for oxygen isotope equilibration with meteoric water ($\delta^{18}O = -10‰$), marine water ($\delta^{18}O = 0‰$), and magmatic water ($\delta^{18}O = +8‰$) calculated according to the data of O'Neil et al. [95]. The grey fields show the positions of the two calcite generations Cal I and Cal II from Savda (JAL-31, JAL-32).

In contrast, homogenization temperatures of stage II calcite ($T_h$ 94–174 °C) associated with relatively high $\delta^{18}O$ values require considerably higher $\delta^{18}O$ values for the mineralizing fluid. The plot in Figure 20 shows that the calcite II generation seems to be crystallized from a dominantly magmatic fluid. The dominance of a magmatic component can be explained by an increasing burial. The analyzed elevated fluid inclusion temperatures of calcite II indicate its hydrothermal origin. The values are similar to hydrothermally crystallized calcite in cavities of Permian basic volcanics of the Saar-Nahe region in Germany [91].

The strongly scattering homogenization temperatures (i.e., densities from 0.79 to 0.96 g/cm$^3$) of FIs in calcite II can, beside postentrapment modifications like necking-down and/or re-equilibration, also provide indications of fluctuating temperatures during crystallization. Both crystallization during burial and uplift can change the temperature regime and also influence the isotope fractionation processes. This would also explain the scattering $\delta^{18}O$ values in calcite II (Table 6). Evidence for a correlation between temperature/depth and $\delta^{18}O$ values was found for hydrothermal calcite in fissures and fractures of different locations [96–98]. These variations in the temperature and isotope regime also indicate a long-lasting mineralization process under slightly changing crystallization conditions.

### 5.5. Feldspar, Zeolites, and Powellite

Former reports on zeolites from the DVP did not take in account different mineralization stages [7,9,14–16,99]. Sampling, identification, and description were mainly based on the abundant occurrence of finely developed, centimeter-sized crystals. Zeolites as alteration replacements of plagioclase in the whole rock, as described from other basalt provinces [9,11,25], were not reported. Countless centimeter-sized zeolite crystals, such as heulandite, stilbite, or scolecite, were deposited on mordenite or chalcedony [18,47]. Zeolites overgrown by chalcedony were not reported and probably not observed and investigated before. The precipitation of chalcedony during the mineralization of secondary minerals in basalts is commonly assumed during the burial stage [31].

The first detection of zeolites overgrown by chalcedony in the Savda quarry plays a significant role for the developed mineralization model. These stage I zeolites have been indubitably formed by burial metamorphism and not in a late stage by new hydrothermal events. Another new finding was the detection of subhedral plagioclase that was precipitated together with heulandite, stilbite,

mordenite, and overgrown by chalcedony. While plagioclase as a primary mineral is a main component in whole rock, the occurrence as a secondary mineral formed from a fluid was surprising and has not been reported before, even from other basalt provinces. The only occurrence of secondary feldspar in sample JAM-10 does not allow a general conclusion concerning the crystallization conditions. Morphological criteria and the common occurrence with zeolites, however, clearly point to a formation from mineralizing fluids.

Jørgensen [22] estimated formation temperatures for heulandite, stilbite, and mordenite of 70 to 200 °C with a main range for heulandite and stilbite between 110–130 °C and for mordenite between 70–150 °C. Accordingly, the subhedral crystals of plagioclase and first generation of zeolites I were formed between the assemblage of clay minerals and chalcedony/quartz. This stage happened probably contemporaneously or after the crystallization of calcite I during burial by subsequent lava flows.

Barth-Wirsching and Holler [100] and Wirsching [101] studied the experimental hydrothermal formation of zeolites at varying conditions and temperatures between 50–250 °C in open and closed systems. They concluded the influence of the starting material (basaltic or rhyolitic glass or plagioclase) and temperature during alteration as important factors for formation of distinct zeolite species. The formation of plagioclase and anorthite from basaltic glass as source material in reaction with a calcium solution was observed in an open system. Anorthite formed as an alteration product from earlier wairakite from a basaltic glass by reaction with a calcium solution indicating formation in an early stage of alteration [101]. Albitization of plagioclase in the matrix of the host rock, as reported in other basalt provinces [9], was not observed in the central flow zone in Savda. The host rock containing large cavities in the core line of the flow in Savda presents a dense texture without significant alteration. Therefore, detection of plagioclase as subhedral crystals in SFFs between clay minerals and overgrown silica was surprising.

The crystallization sequence of the minerals formed after chalcedony and calcite II was established by visual observations of numerous samples: heulandite-Ca–stilbite-Ca–calcite III–powellite–apophyllite (mineral names considering the IMA nomenclature for zeolite minerals [102]). Heulandite-Ca crystals are deposited on chalcedony or calcite II and do not form intergrowths with stilbite. Stilbite-Ca is deposited after heulandite-Ca. It can be stated that heulandite-Ca and stilbite-Ca in Savda crystallized earlier than apophyllite or in the same stage close together. James and Walsh [17] studied the chemical composition of zeolites from the Deccan basalts. Their analyses confirmed that the zeolites from the Deccan basalts were very pure and contained only Ba and Sr as substantial trace elements.

A conspicuous feature of the secondary mineralization in several localities of the central DVP is the occurrence of powellite [68,69,103]. It is rarely present as a few small crystals in the cavities, very rarely up to 5 cm in size and grown on stilbite-Ca and overgrown by apophyllite. Although specific data concerning the Mo content for the DVP are not available, results of Liang et al. [104] show that Mo is present only at a low ppm level (<5 ppm) in basalts. Therefore, the formation of big powellite crystals in the cavities of Mo-poor basalts is surprising. In general, powellite is only common in the oxidation zone of molybdenum deposits related to more felsic rocks. In the investigated localities, the occurrence of powellite could probably be related to the release of Mo from the basalts during alteration of large basalt masses by fluids and its supply and accumulation by the hydrothermal fluids in the cavities. Powellite precipitation is noteworthy, since molybdenum must have been present in the same fluids as the elements necessary for the formation of stilbite. This point needs further investigation. The presence of powellite at least points to oxidizing conditions during crystallization.

## 5.6. Apophyllite

For an exact dating of the secondary mineralization in relation to the general geological development of the DVP, only apophyllite is suitable, whereas the zeolites heulandite and stilbite cannot be used for direct dating. Analyses of different apophyllite samples from Savda provided an age of 25–49 Ma. Such scattering of the absolute age was not expected. A possible explanation could be

a complex geological process running at several time and space scales [20]. This could probably also explain the strongly varying values of the homogenization temperatures of fluid inclusions. The high $T_h$ values of 250–280 °C for apophyllite from Savda (exact position in the quarry and age is unknown) determined by Srikantappa and Mookherjee [16] and Mookherjee [105] indicate a genesis at boiling conditions. The range in homogenization temperatures ($T_h$) between 141–222 °C for sample JAL-13D (44–47 Ma) is lower than determined by Srikantappa and Mookherjee [16] and Mookherjee [105], but it also favors a hydrothermal genesis from a more dense fluid. If density increase results from volume loss of the fluid, it can be assumed that the scattering values are conditioned by different precipitation ages of apophyllite.

The high homogenization temperatures and the apophyllite age indicate an origin by late hydrothermal fluids. Well-developed, large zeolite crystals are believed to have in general a hydrothermal origin [106]. Considering the results of geochronology, it was obvious that a geological event generating hydrothermal fluids did not happen during or directly at the end of the burial. Apophyllite precipitation in Savda occurred much later than the formation of the DVP at 65 Ma. In addition, during apophyllite formation at 25–49 Ma, much of the overlaying basalt was already eroded, and an explanation of the observed temperatures by late burial metamorphism is very unlikely.

Interpretation of the geochronological data is not straightforward, as the data from both sampling areas exhibit distinct age groups, and all K-Ar ages are younger than the corresponding Rb-Sr ages. Since apophyllite, as well as corresponding stilbite and calcite separates used for dating, were prepared from centimeter-sized, idiomorphic, limpid, and chemically homogeneous crystals, effects of secondary alteration of the investigated material can be ruled out.

With respect to the Rb-Sr method, it has to be mentioned that even if the Rb content is low in stilbite, extremely low in calcite, and the Sr content in some of the apopyllite is less than 1 ppm, the results are reliable from an analytical point of view. Two arguments support this assumption: (i) apophyllite and stilbite show characteristic concentrations for Rb and Sr in a limited range, which fits to data from the literature [70,107]; and (ii) repeated analyses of one apophyllite concentrate (JAL-16B) and two separates representing the core and rim of a centimeter-sized apophyllite crystal (JAL-13C) yield consistent analytical results. The $^{87}Rb/^{86}Sr$ ratios (40–860) fit to those reported in Fleming [70] and are in the range of muscovite or biotite. The second data points used for age calculation are characterized by low $^{87}Rb/^{86}Sr$ ratios (<1), and, therefore, the measured $^{87}Sr/^{86}Sr$ ratios are close to the initial values of the individual age regression lines. Accepting a geological significance of the Rb-Sr ages in both investigated areas, regional clusters are observed even over small distances of a few hundred meters to kilometers. Three ages from the eastern part of the Savda quarry complex are in the range of 44–48 Ma, whereas two samples from the western part show ages of 25–28 Ma. From the Nashik area, one cluster is 55–58 Ma and the other is 21–23 Ma.

K contents of the apophyllite concentrate used for K-Ar age determination were measured by atomic absorption spectrometry (AAS). They were in the range of 3.82–4.09% and, therefore, were similar to the in situ SEM measurements on sample JAL-13D and JAL-16B (Tables 8 and 10). The measured $^{40}Ar_{rad}$ was relatively low and ranged between 18.4–52.1%. The K-Ar ages corresponded to the Rb-Sr ages measured on the same sample material, but they were systematically younger by 6–35%.

Combined Rb-Sr and K-Ar studies on apophyllite are reported by Chukrov et al. [108], Fleming et al. [70], and Molzahn et al. [107]. Both latter studies used apophyllite from Antarctica and additionally applied the Ar-Ar method. According to Fleming et al. [70] the Ar-Ar age spectra showed a rapid loss of Ar in a narrow temperature interval (750–800 °C), and the incremental heating spectra did not contribute much more information than the total fusion ages. In addition, the K-Ar and Ar-Ar ages given in Mohlzahn et al. [107] were more or less identical. However, the results from all studies fit to those reported here: The K-Ar and Ar-Ar ages of the apophyllites are always much younger than those of the hosting basalts. K-Ar and Ar-Ar ages are generally younger than the Rb-Sr ages, and the age data of individual areas cover a wide age range. Therefore, the ages are interpreted to

reflect apopyllite growth during hydrothermal processes not directly related to the magmatic event. Chukhrov [108] was more convinced by the Rb-Sr data. In contrast, Mohlzahn et al. [107] believed in some of their Ar-Ar ages. However, their geologically meaningless Rb-Sr data were due to very large errors on the Sr isotopic ratios, which were most probably a result of contamination by Rb. Mohlzahn et al. [107] and Fleming et al. [70] discussed several reasons for the scattering ages including Ar-loss, alkali (K, Rb) and/or Sr mobility, or an extended period of mineral precipitation.

For the well-preserved crystals of the DVP investigated in this study, it is hard to believe in chemical changes of the apophyllite in particular, as it is the latest mineral in the mineralization sequence and there is no later overprint. Further, it is difficult to explain why K, Rb, or Sr should have been mobile after formation, whereas Ar with a smaller ionic radius and a different electric charge stays stable. This contradicts the observations on other mica minerals showing higher blocking temperatures for the Rb-Sr than for the K-Ar isotopic system [109,110]. Therefore, we followed Chukhrov et al. [108] who believed that Ar-loss might be an important factor. However, with the available data it is not possible to give an accurate explanation for the Ar-loss.

The Rb-Sr ages indicated that apophyllite formed at different times within the cavities of individual lava flows but also within different parts of one and the same lava flow over distances of a few hundred meters. The formation seemed to be controlled by pathways of meteoric water, which was generated over several million years at different places. In the area around Nashik, crystallization took place shortly after deposition of the basalts in the late Paleogene to early Eocene and in the Miocene, whereas at Savda apophyllite formed in the middle Eocene and the Oligocene.

### 5.7. Multistage Mineralization Model of Secondary Minerals in Savda

Early formation of SFFs and associated minerals, unusual calcite twins, and the late-deposited zeolites with their paragenetic minerals such as powellite and apophyllite lead to a multistage mineralization model (Figure 21).

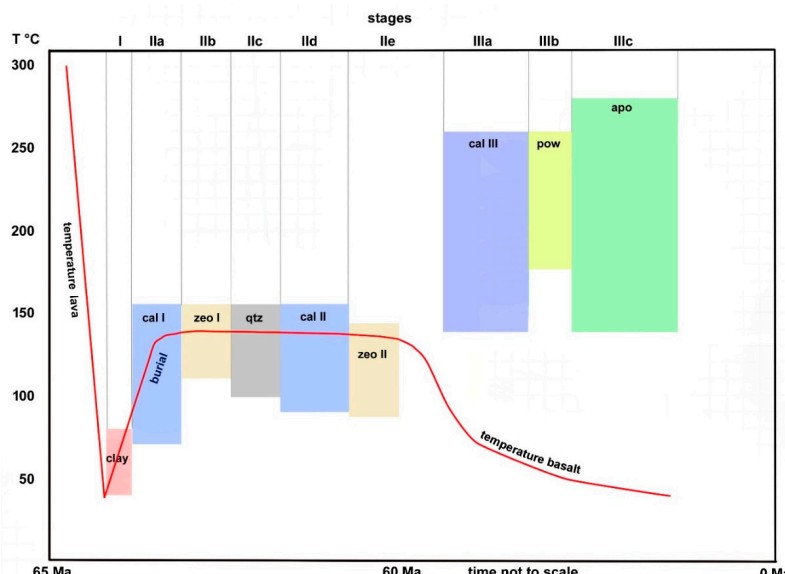

**Figure 21.** Multistage mineralization model for the big cavities in the core zone of the flow in Savda. (clay)—clay minerals; (cal)—calcite; (zeo I)—heulandite; stilbite, mordenite, and plagioclase; (qtz)—chalcedony/quartz; (zeo II)—heulandite and stilbite; (pow)—powellite; and (apo)—apophyllite. The red line represents the estimated temperature of the core zone of the flow, influenced by burial, thermal gradient, and erosion. The time near to the Earth's surface (stage I) can be calculated with several years. The time for burial (stage II) is assumed for a period of approx. 5 Ma, up to the end of the volcanic activity of the DVP, starting with calcite I and ending with zeolite II. The remaining time for stage III amounts approx. 60 Ma.

The effusion of a single lava flow was followed by weathering and alteration at high temperatures. Interaction of largely meteoric water with glass and other easily soluble mineral components of the cooling lava, such as olivine and plagioclase, led to a fluid percolating through fissures/joints and into the large cavities of the core line in the lobe. The fluid filled the big cavities completely, and its temperature continuously decreased. It was assumed that this natural filling of the cavities happened in a rapid process near the surface.

Stage I—The clay minerals and core filamentous fabrics as first subsurface mineralization products occur on all positions of the cavities, and no zonal development of the mineralization within the cavities is detectable. The fluids in the large cavities contained all early released elements of the host rock, such as Si, Mg, Al, and $Fe^{2+}$, and components supplied by the meteoric water such as C, P, and N. Oxidation of $Fe^{2+}$ to $Fe^{3+}$ started producing energy, which was necessary for bacterial life. The bacteria developed filaments, encrusted by unstable ferrihydroxide. Ferrihydroxide was converted to Fe-rich Fe-Mg clay minerals, such as celadonite–glauconite and smectite, which formed the SFF and the cavity wall layers. Stage I happened near to the Earth's surface before burial.

Stage II—Temperature, pH, pressure, and chemical composition of the fluids changed after beginning of burial, resulting in the precipitation of calcite I and/or plagioclase, zeolites I (heulandite, stilbite, and mordenite), chalcedony, calcite II, and finally zeolite II. The precipitation of calcite I (stage IIa), which is influenced by meteoric water, started before the clay mineral precipitation ended. Stage IIa+b mineralization included the precipitation of zeolite I generation and ended with the formation of a thin first $SiO_2$ layer. In several samples, a second mordenite generation could be observed that was deposited between a thin first $SiO_2$ layer and the final thick chalcedony layer. In conclusion, the conditions for mordenite and beginning silica–chalcedony crystallization did not differ significantly. In stage IIc, the chalcedony/quartz assemblage clearly separates the formation of calcite I and calcite II and occurs during the stage of burial. The last phase of stage II after the deposition of calcite II and before calcite III is characterized by the precipitation of zeolite II. The formation temperature of zeolite II is not known and should be estimated according to the zeolite temperature scale [22,31]. Based on the maximum temperature of approximately 150 °C in burial, zeolite II crystallization could have taken place at 150 °C at most during burial. If precipitation took place significantly later after ending of volcanism and erosion started, the forming temperatures for zeolite II should be less than 150 °C. Since the mineralization sequence provided no indication that zeolites II crystallized on calcite III, the formation of zeolite II resulted from conditions of late burial metamorphism.

Stage III—Includes all late mineralization, which occurred at temperatures that cannot be explained by low-grade metamorphism resulting from burial. Considering lowered burial depth resulting from erosion and decreasing geothermal gradient after the end of volcanism (compare Figure 21), estimated temperatures resulting from burial during the time of stage III were significantly lower than the analyzed homogenization temperatures of stage III minerals. It was assumed that stage III lasted over a long time span, beginning with the precipitation of calcite III and ending with apophyllite millions of years after eruption of the lava. Precipitation of calcite III (stage IIIb) at homogenization temperatures of up to 254 °C for fluid inclusions indicated a late hydrothermal formation. It had to be considered that calcite III precipitation occurred at a time of postvolcanic activity when the thermal gradient significantly decreased and erosion had already started reducing the thickness of the overlaying basalt. The formation of powellite (stage IIIb) requires elevated concentrations of Mo generated by leaching of large masses by circulating fluids, which cannot be explained by low grade burial metamorphism. Circulating fluids with elevated temperatures can also explain the dissolution of vanadium, which is present in apophyllite in concentrations of up to 3000 ppm. The results of age dating indicated that apophyllite (stage IIIc) was formed millions of years after the preceding mineral assemblages by late hydrothermal fluids.

In summary, stage I happened in a rather short time after eruption near to the surface before burial started. Stage II reflects a metamorphism and mineralization sequence by burial. Stage III is

characterized by high homogenization temperatures and results from late hydrothermal processes, predominantly for calcite III, powellite, and apophyllite, several million years later.

*5.8. General Relevance of the Multistage Mineralization Model for the Deccan Volcanic Province (DVP)*

It must be considered that the above described multistage mineralization model has its specific validity for the Savda/Jalgaon area and is limited to the secondary minerals in the large cavities in the flow core line. An investigation on the secondary minerals from Savda in all three zones of the flow, including the whole rock, would allow a better understanding of complete alteration and mineralization processes. The conditions for mineralization of the secondary minerals could widely differ in other areas of the DVP for several reasons. For example, it can be speculated that the difference in thickness between the paleo- and the recent surface near Nashik is much higher (2000–2500 m) than in Savda. Therefore, the temperatures during burial might have reached more than 200 °C in the Nashik area. Temperatures of >200 °C are necessary for the formation of laumontite, which is common in the Nashik region. In contrast, in the Mumbai area, outcrops like in Mumbai Malad are built up by basalts–spilites of the Salsette subgroup, representing the latest eruption phase of the DVP [42]. Most likely these parts were never buried significantly after their deposition, and they experienced no burial metamorphism. The scattering age dates and temperatures of the fluid inclusions indicate different activities of hydrothermal processes at different times and places, even at small distances.

To develop a general mineralization model for the main part of the DVP, further extensive investigation needs to be conducted starting with systematic sampling over several stratigraphic formations and different altitudes. In contrast to the former reported mineralization models and recommended zones by Walker [8] and Sukheswala [14], when a predominantly visual determination of the sampled minerals takes place, numerous analyses of fluid inclusions, stable isotopes, rock composition, as well as mineral identification by spectrometry, XRD, and SEM/EDX are necessary.

*5.9. Comparison with Mineralization Models of Other Volcanic Provinces*

The multistage mineralization model established for Savda, based on the secondary minerals from the large cavities of a flow core zone, allows only a limited comparison with models that consider other basalts. Neuhoff [9] developed a model for basaltic lavas at Teigarhorn, Eastern Iceland, based on burial metamorphism and spatial and temporal pore filling. The precipitation of celadonite was noted in an early stage near to the surface, and precipitation of smectite and heulandite-stilbite were noted by burial in stage II. The later mineral assemblage in stage III was explained by associated dykes. Calcite, for the determination of homogenization temperatures, was not available. The presence of celadonite/smectite together with albite and zeolites was confirmed in the altered matrix but not as secondary minerals in the vesicles or cavities. Weisenberger and Selbekk [11] described a multistage zeolite facies mineralization in the Hvalfjödur area, Iceland. They noted as results of different alteration stages with increasing temperature: (a) celadonite at near-surface alteration; (b) mafic phyllosilicates with increasing burial; (c) zeolite assemblage during burial; and (d) zeolite assemblages by later hydrothermal activities resulting from tectonic events. Stage I was characterized by primary pore lining by celadonite and silica minerals and later by mafic phyllosilicates. They divided the phyllosilicate precipitation into stage Ia including green celadonite and Ib consisting of celadonite–smectite mixed layers. Zeolite mineralization took place by burial in stage II and could be grouped in three different zones according to the depth below the surface. It was concluded that the paleo-temperature field governed zeolite distribution. A paleo-geothermal gradient of 133 ± 10 °C/km was estimated. Late mineralization like in Savda was not reported for the Hvalfjödur area. A comparison of the mineralization conditions of zeolites in Savda and Iceland showed no fundamental differences in mineralization in Stages I and II.

A thorough mineralogical and geochemical study by Gilg et al. [111,112] using fluid inclusion studies in combination with hydrogen, oxygen, and sulfur isotope data revealed four mineralization processes during the genesis of giant amethyst-bearing geodes in Early Cretaceous Paraná continental flood basalts at Amestita do Sul, Brazil. Different silica minerals, calcite, and sulfates formed

by ascending hydrothermal fluids of up to 80–90 °C infiltrating a basaltic host rock of less than 45 °C. A comparison with the present data was difficult because of the significantly lower temperature conditions.

Kousehlar et al. [25] described a low-temperature mineral formation in basaltic to andesitic volcanic rocks of Kahrizak, Iran. They proposed formation of smectite–chlorite and early stage I zeolites by burial metamorphism and a second stage II zeolite assembly by new hydrothermal activity. They noted a secondary albitization of plagioclase resulting from whole rock alteration but, in contrast to Savda, not as subhedral crystals in vesicles or cavities. Stage I alteration and mineralization is caused by interaction with heated groundwater and burial. It is suggested that increasing temperature and pressure, and changing fluid composition by a decrease in Mg and Fe relative to Ca, results in a change from mafic phyllosilicates to zeolites. The mineralization stages of the Kahrizak basalts demonstrate, similar to Savda, an early precipitation of clay minerals near to the surface and a following sequence of burial. A comparison of the secondary mineralization between Savda and the former cited volcanic provinces in Iceland and Iran showed remarkable similarities. Clay minerals precipitated in an early stage near to the surface. Zeolites formed resulting from burial according to the temperature scale for zeolites in depth- and temperature-controlled zones.

The age of apophyllite precipitation from Savda gives strong evidence for late hydrothermal fluids with high temperatures, which are not noted for Iceland or Iran. Age dating of apophyllite from the Kirkpatrick Basalts, Antarctica [70], resulted in apophyllite ages between 95–144 Ma, depending on the locality and using K-Ar and Rb-Sr methods, while ages of approximately 175–185 Ma for the basalts were calculated. The timing of apophyllite formation in Savda as well as in the Kirkpatrick basalts proves late mineralization a long time after the basalt eruption by fluids with elevated temperatures without an identification of the cause of the responsible event.

The formation temperature of the zeolites in the basalts of Iceland and Iran was determined by applying the formation temperature scale for zeolites [22,33]. Calcite of several generations for the determination of the homogenization temperatures was not available. In Savda, on the other hand, there was insufficient data available for zeolites of different altitudes to calculate a formation temperature for zeolites. The determined homogenization temperatures of calcite from Savda, however, allowed classification of the zeolites according to the mineralization sequence in temperature zones that matched those of the zeolite temperature scale. Previously cited articles on secondary minerals from Iceland and Iran were mainly based on minerals that occurred in vesicles and amygdales as well as in the whole rock. This paper gives no detailed information about the alteration and mineralization in the whole rock or the vesicles and amygdales in the upper zone, but it allows an insight into the mineralization in the large cavities over a long period.

The mineralization sequence in Savda is typical for terrestrial lava flows without any influence of sediments or marine environment. Different conditions prevailed in other parts of the large lava flows (e.g., in the coastal part of Mumbai), where the lava spilled into water and caused the formation of pillow lavas, and also different mineralization, which is similar to the Watchung Mountains, NJ, USA [113–115]. The large cavities in Paterson (Watchung Mts.) are explained as diapirs [116], whereas the origin of the large cavities in Savda can be explained by the commonly accepted process of vesiculation of compound flows [117]. The formation of the large sizes in the center of the flow in Savda is due to the fact that bubbles have the longest time to rise from the lower parts and coalesce with each other.

## 6. Conclusions

Investigations on secondary minerals from large cavities in basalts from the Deccan Volcanic Province allow the following conclusions:

Exceptional mineralization of subsurface filamentous fabrics (SFFs), calcite, and other secondary minerals (such as zeolites or apophyllite) in Savda provides a unique insight into the mineralization conditions of the DVP.

Secondary mineral formation shows a characteristic succession and took place in three main stages: (1) Phyllosilicates/clay minerals including the innermost SFF formed during cooling of the lava flows near the Earth's surface. (2) The calcite generations I + II, zeolites I + II, and chalcedony developed during burial. Occasionally, zeolites (e.g., mordenite) are enclosed in silica minerals. (3) Calcite III, powellite, and apophyllite crystallized from late hydrothermal fluids, which showed no significant change in their chemical composition over a long period of time.

Morphology and texture of the SFF cannot be explained by formation of stalactites or biomorphs and provide strong indication of a biogenetic origin. It is assumed that initial microbial activities resulted in the formation of threads with diameters up to 50 μm and lengths of more than 100 cm consisting of a larger number of single accumulated strands with a characteristic morphology. This filamentous core probably represents the seed for the following crystallization sequence starting with sheet silicates.

Occurrence of specific minerals and their chemical characteristics in the different crystallization stages provides indications concerning the physico-chemical conditions during the formation of secondary minerals in the basalts of the DVP. Chemical elements necessary for the formation of secondary minerals derive from the alteration of volcanic glass and minerals (olivine, clinopyroxene, and plagioclase) of the volcanic host rocks.

$\delta^{18}O$ and $\delta^{13}C$ isotope data support the influence of less exchanged meteoric water during the early stages of mineralization in the cavities and more exchanged, magmatic fluids during the later stages of mineralization. An early formation of Fe/Mg-rich phyllosilicates was followed by a decrease of Mg and Fe relative to Ca fluid activities in the crystallization sequence resulting in the precipitation of Ca-rich zeolites. In addition, homogenization temperatures of the fluid inclusions in certain minerals provide essential data for the mineralization sequence by burial and hydrothermal processes, respectively.

Rb-Sr and K-Ar age data obtained from apophyllite indicated crystallization over a long time period, from the Paleogene to the early Miocene. Crystallization ages cluster at 44–48 Ma and 25–28 Ma for samples from the Savda quarry complex, whereas those from the Nashik area show 55–58 Ma and 21–23 Ma, respectively. This indicates formation of apophyllite at different times even within individual lava flows at certain localities.

**Author Contributions:** B.O. and R.S. collected the studied samples. All authors conducted different analytical measurements and evaluated the mineralogical and geochemical data. B.O, J.G., and R.S. wrote the manuscript.

**Funding:** This research received no external funding.

**Acknowledgments:** We like to thank Christian Rewitzer (Furth im Wald) for his support with SEM/EDX analyses, Reinhard Kleeberg (TU Bergakademie Freiberg) for assistance with XRD data acquisition, and Jessica Gärtner (TU Bergakademie Freiberg) for polarizing and scanning electron microscopy. Monika Horschinegg is acknowledged for her help performing the Rb-Sr measurements at the University of Vienna. K/Ar analyses were supported by the János Bolyai Scholarship of the Hungarian Academy of Sciences to Zsolt Benkó. The authors also like to thank Sam F. Sethna, Mumbai and three anonymous reviewers for the critical and insightful reading and comments.

**Conflicts of Interest:** The authors declare no conflict of interest.

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
