# Peer review of "Exceptional Multi Stage Mineralization of Secondary Minerals in Cavities of Flood Basalts from the Deccan Volcanic Province, India"

_minerals, doi:10.3390/min9060351_

Round 1
Reviewer 1 Report
I find this is a very interesting study, only few similar detailed investigations of secondary mineralizations in basalts are existing.
General points that the authors should address:
- The use of the term „filament“ is unusual, and is not really clear what the authors mean with the term. A filament is a one-dimensional structure of undefined origin. The authors use the term for rod-like mineral aggregates, but also for the core structures of these aggregates. IN the beginning of the paper, the use implies the presence of filaments in the core, but this needs to be demonstrated first. -> I suggest that a different term should be used for the mineral aggregates (rod, stalk...), and then the authors should show that they formed by mineral precipitation on a filamentous substrate..
- Some terms used are related to the „quality“ of minerals as seen from a mineral market perspective, such terms should be avoided in a scientific paper (noted in special remarks)
- The term „hydrothermal“ seems to be used to indicate the influence of hot, extrernally derived aqueous fluids, while minerals formed at 100°C during normal burial from formation waters are not referred to as „hydrothermal“. This term should be used free of genetic implications.
- The authors infer a late-stage hydrothermal mineralization that cannot be explained by burial alone. The fluids are oxidizing (transport of molybdate, vanadate). There appear to be times of preferred apophyllite mineralization in the Miocene-Oligocene and in the Eocene. Can the authors tell someting about possible causes, e.g. collision history with Eurasia?
- Some references relevant are missing, or old ones are used. I suggest that the authors should compare their results with the study by Gilg et al. (2014):
Gilg, H.A., Krüger, Y., Taubald, H., Kerkhof, A.M.v.d., Frenz, M., Morteani, G., 2014. Mineralisation of amethyst-bearing geodes in Ametista do Sul (Brazil) from low-temperature sedimentary brines: evidence from monophase liquid inclusions and stable isotopes. Mineralium Deposita 49, 861-877.
For the very common occurrence of filamentous fabrics, the the paper by Hofmann et al. (2008) is much more comprehensive than the one from 2000:
Hofmann, B.A., Farmer, J.D., von Blanckenburg, F., Fallick, A.E., 2008. Subsurface filamentous fabrics: An evaluation of possible modes of origins based on morphological and geochemical criteria, with implications for exopalaeontology. Astrobiology 8, 87-117.
Individual points in the text (with line number):
24: „quality“ see comment above
30, 32: second calcite II and thrid calcite III is a bit strange (is there a first calcite II?)
30: plagioclase: can this be specified, albite?
54: use of the term „hydrothermal“, the minerals surely are hydrothermal, but the origin of the fluids is under discussion
55: „Fine developed“, better „well developed“, is without „market classification“
65: „... of large filaments“ is an example of the use of filaments noted above. I suggest to use a description such as „large rod-like mineral aggregates with filament cores“ or „...encrusting a filamentous substrate“ or similar.
158: „Polarizing microscopy“ -> polarized light microscopy, specify whether in transmitted, reflected light or both
159: „transmitted and polarized light“ do you mean „transmitted polarized light microscopy“?
181 X-ray diffractormetry, methods: I have used this method a lot. Based on the description in the manuscript I must assume that 0.03° steps were used, each with a measuring time of 5s...? Then a complete measurement of 75° 2Theta would have taken more than 3 hours??
207 „analyses were made“ better: performed, done
273 „form complex filaments“ try to find another wording to make clear that the filaments older than the clay minerals
273, 321 „celadonite, glauconite“ glauconite is not an official IMA mineral name, composition is covered by the four celadonite minerals celadonite, aluminoceladonite, farroaluminoceladonite and ferroceladonite and their solid solutions.
See:
400 „was formed from“ replace by „incorporated“ (there are other elements in calcite)
402 „compared to the delta 18O values“ this is unclear, how can you compare C isotope values with oxygen isotope values?? Do you mean „while the delta 18O values are much lower“ ?
433: „....as first solid phase“. This is not based on your work here, I assume, so there should be a reference.
443: plagioclase: A plagioclase, in the sense of a Ca-bearing Na-Ca-feldspar, seems a bit strange as hydrothermal formation. Have you checked whether it is really Ca-bearing? Typically such feldspars are albite.
480 This table lacks a header
465 small „o“ i nC“aMoO4“
536 X-ray spectrometry -> diffractometry?
539-540 Strictly speaking the whole rock is not cogenetic with the secondary minerals and should not be plotted on an isochron. I agree the deviation will be small.
Figute 13: It would be nice to include the absolute ages here somehow.
Figure 15: It is unclear what „flint“ (=SiO2) is meaning on a diagram of delta13C versus delta 18O ?
734: „a hydrothermal origin“ see comment at beginning.
818 „glauconite“ see comment before
Author Response
Review report 1 Author answer (blue) to the general and single points
Open Review (x) I would like to sign my review report
English language and style (x) English language and style are fine/minor spell check required
Yes Can be improved Must be improved Not applicable
Does the introduction provide sufficient background and include all relevant references? ( ) ( ) (x) ( )
Is the research design appropriate? (x) ( ) ( ) ( )
Are the methods adequately described? (x) ( ) ( ) ( )
Are the results clearly presented? ( ) (x) ( ) ( )
Are the conclusions supported by the results? ( ) (x) ( ) ( )
Comments and Suggestions for Authors
I find this is a very interesting study, only few similar detailed investigations of secondary mineralizations in basalts are existing.
General points that the authors should address:
- The use of the term „filament“ is unusual, and is not really clear what the authors mean with the term. A filament is a one-dimensional structure of undefined origin. The authors use the term for rod-like mineral aggregates, but also for the core structures of these aggregates. IN the beginning of the paper, the use implies the presence of filaments in the core, but this needs to be demonstrated first. -> I suggest that a different term should be used for the mineral aggregates (rod, stalk...), and then the authors should show that they formed by mineral precipitation on a filamentous substrate..
The term “filament” is used for similar chalcedony fabrics by Hofmann-Farmer, [55] and Feucht_Chr_2006. The term subsurface filamentous fabrics is used by Hofmann-Farmer, [56]. Other use terms like filamentous fossil microbiotas or biofabrics, filamentous fossil bacteria, jasper filaments or filamentous iron-silica deposits. Morteani et al., [70] use the term stalactite.
“Subsurface filamentous fabrics (SFF) are here defined as microscopic to macroscopic mineral fabrics that result from the precipitation of minerals on a substrate of filamentous (thread-like) geometric units in subterraneous environments”. (Hofmann et al., [56])
The fabrics described in the paper are complex subsurface formations of first developed filaments with probably bacteria origin overgrown by minerals like clay minerals, zeolites and silica. In respect of these conditions, the term filament will be changed generally for the complex fabrics into “subsurface filamentous fabrics (SFF)”.
The definition will be explained in the chapter 4.2.2. (Filaments) Subsurface filamentous fabrics
- Some terms used are related to the „quality“ of minerals as seen from a mineral market perspective, such terms should be avoided in a scientific paper (noted in special remarks)
are corrected
- The term „hydrothermal“ seems to be used to indicate the influence of hot, externally derived aqueous fluids, while minerals formed at 100°C during normal burial from formation waters are not referred to as „hydrothermal“. This term should be used free of genetic implications.
The formation of zeolites by low (temperature) grade metamorphism during burial is a result of a distinct hydrothermal process. In other papers relating to zeolite formations the authors (M. KOUSEHLAR et al., [21]) use the term “low temperature metamorphism by burial” or “burial metamorphism” to characterize the distinct kind of hydrothermal formation. They use the term hydrothermal commonly in contrast to the burial for late hydrothermal events. We will use burial for the distinct process and origin by late hydrothermal fluids.
- The authors infer a late-stage hydrothermal mineralization that cannot be explained by burial alone. The fluids are oxidizing (transport of molybdate, vanadate). There appear to be times of preferred apophyllite mineralization in the Miocene-Oligocene and in the Eocene. Can the authors tell someting about possible causes, e.g. collision history with Eurasia?
Any explanations for the late hydrothermal mineralization at higher temperatures in the observed area in the DVP are difficult and base on speculations. The mineralization took place during a long time span. During the Deccan volcanism (65 -62 Ma) new erupting flows through dykes could influence the fluid temperature in the former, underlaying flows. Such influence should have be finished after the end of volcanism. Tectonic events after ending of the volcanism in the DVP were not reported and should not be considered for the mineralization at much younger times. Probably, upwelling heat flows can be the cause. Investigations on heat flows show (Shanker, 1988 ) elevated values in the Sone -Narmada Tapti (SONATA) zone. Jalgaon / Savda are situated close to this zone.
- Some references relevant are missing, or old ones are used. I suggest that the authors should compare their results with the study by Gilg et al. (2014):
Gilg, H.A., Krüger, Y., Taubald, H., Kerkhof, A.M.v.d., Frenz, M., Morteani, G., 2014. Mineralisation of amethyst-bearing geodes in Ametista do Sul (Brazil) from low-temperature sedimentary brines: evidence from monophase liquid inclusions and stable isotopes. Mineralium Deposita 49, 861-877.
For the very common occurrence of filamentous fabrics, the paper by Hofmann et al. (2008) is much more comprehensive than the one from 2000:
Hofmann, B.A., Farmer, J.D., von Blanckenburg, F., Fallick, A.E., 2008. Subsurface filamentous fabrics: An evaluation of possible modes of origins based on morphological and geochemical criteria, with implications for exopalaeontology. Astrobiology 8, 87-117.
We will check it and add relevant references
Individual points in the text (with line number):
24: „quality“ see comment above
changed in: partly well developed
30, 32: second calcite II and third calcite III is a bit strange (is there a first calcite II?)
rewritten
30: plagioclase: can this be specified, albite?
explained in results as plagioclase with Norm. At. % 2.3 Ca, 0.2 Na und 0.2 K
54: use of the term „hydrothermal“, the minerals surely are hydrothermal, but the origin of the fluids is under discussion
changed in: late hydrothermal mineralization
55: „Fine developed“, better „well developed“, is without „market classification“
rewritten
65: „... of large filaments“ is an example of the use of filaments noted above. I suggest to use a description such as „large rod-like mineral aggregates with filament cores“ or „...encrusting a filamentous substrate“ or similar.
changed in: filamentous subsurface fabrics
158: „Polarizing microscopy“ -> polarized light microscopy, specify whether in transmitted, reflected light or both
see below
159: „transmitted and polarized light“ do you mean „transmitted polarized light microscopy“?
rewritten:
181 X-ray diffractormetry, methods: I have used this method a lot. Based on the description in the manuscript I must assume that 0.03° steps were used, each with a measuring time of 5s...? Then a complete measurement of 75° 2Theta would have taken more than 3 hours??
This is correct; we used these analytical conditions for a thorough phase characterization.
207 „analyses were made“ better: performed, done
changed
273 „form complex filaments“ try to find another wording to make clear that the filaments older than the clay minerals
change in: filamentous subsurface fabrics
273, 321 „celadonite, glauconite“ glauconite is not an official IMA mineral name, composition is covered by the four celadonite minerals celadonite, aluminoceladonite, farroaluminoceladonite and ferroceladonite and their solid solutions.
changed in: Celadonite, glauconite (composition is covered by the isostructural minerals aluminoceladonite, celadonite, chromceladonite, ferroaluminoceladonite, ferroceladonite) and smectites
in 321 not to change, because explained before
402 „was formed from“ replace by „incorporated“ (there are other elements in calcite)
the sentence was rewritten
405 „compared to the delta 18O values“ this is unclear, how can you compare C isotope values with oxygen isotope values?? Do you mean „while the delta 18O values are much lower“ ?
the sentence was rewritten:
436: „....as first solid phase“. This is not based on your work here, I assume, so there should be a reference.
reference is given
443: plagioclase: A plagioclase, in the sense of a Ca-bearing Na-Ca-feldspar, seems a bit strange as hydrothermal formation. Have you checked whether it is really Ca-bearing? Typically such feldspars are albite.
the feldspar detected in sample JAM 10 by EDX has a composition with norm. At. % 2.3 Ca, 0.2 Na und 0.2 K indicating plagioclase.
480 This table lacks a header
see bottom of the page above
465 small „o“ i nC“aMoO4“
corrected
536 X-ray spectrometry -> diffractometry?
corrected “X-ray diffraction” instead of “X-ray spectrometry”
539-540 Strictly speaking the whole rock is not cogenetic with the secondary minerals and should not be plotted on an isochron. I agree the deviation will be small.
Yes, we agree. To get an idea about the impact of the error on the age value caused by a different initial 87Sr/86Sr value we calculated ages with virtual initial ratios. The calculated initial ratios for the three apophyllite-WR ages are in the range of 0.707190+/-0.000005 (see Table 7). Using initial ratios reflecting the lowest and highest values measured on the minerals of 0.7084 (celadonite) and 0.7065 (calcite) the deviation on the age is about ±0.5 Ma, which is more or less in the range of the given error. The sentence “The initial 87Sr/86Sr ratios calculated with the individual ages are 0.7071-07078 in the Savda quarries and 0.7088-0.7164 in the Nashik area.” is a statement, but it might be misleading because the “initial values” correspond to different ages. We skipped this sentence.
We changed the text
Figute 13: It would be nice to include the absolute ages here somehow.
will be added in the figure 17
Figure 15: It is unclear what „flint“ (=SiO2) is meaning on a diagram of delta13C versus delta 18O ?
Fig. 15 was redrawn (including revised Figure caption)
734: „a hydrothermal origin“ see comment at beginning.
see our comment at the beginning
818 „glauconite“ see comment before
must not be explained again
Reviewer 2 Report
The manuscript describes a series of phases of secondary mineralisation in the Deccan basalts at Savda. The descriptions of the samples are good and a considerable range of analytical techniques have been applied to document the sequence of mineralisation. They produce a model with a sequence of mineralisation that is plausible in general terms but is not fully justified in the information and data in the manuscript.
The analytical work give a range of results that do not hang together particularly well and a range of reasons are given to try to account for this. The explanations are not always clear and the end conclusion is that there is not a coherent widespread series of mineralisation events, rather that different processes happen at different times in different places in various degrees of intensity. One cannot help thinking that perhaps the data are not giving reliable results and this causes much of the variation. The authors need to improve the explanations to eliminate that possibility.
The manuscript is produced by authors who do not have English as a first language and in general it is fairly good. However in several places it needs to be improved to clarify the meaning and make it easier to read. These section are marked by a highlight on the commented manuscript.
Line 81 it would be useful to have a general picture of the quarry similar to Figure 11 here to give the reader a general impression of the setting
Line 97 what is the basis for geothermal gradient
Line 107 Place names in text and figure have different spellings
Line 117 explain diastrophism a term rarely used these days
Line 125 Figure 2 b needs some arrows to indicate the different minerals
Line 136 You use the phrase core line of lobes. Not clear what you mean. Lobes are projecting parts. Do you mean lobes projecting from the original lava flows or embayments within the main quarry wall? Clarify
Line 195 can cooling the sample first cause problems with later measurments?
Line 249 Why do you use lobe rather than just flow as this is just a flat quarry face?
Line 280 subsurface in respect to what, the flow as whole the individual cavity are there also surface filaments?
Line 285 Figure 4 caption. Is the shading meant to indicate that the lower part was liquid and the upper part vapour. By gravity controlled do you mean microstalactites?
Line 315 The formation of the filaments with the empty tube needs explanation. Do you think there was originally something there if so say so and suggest what it might have been and why it has gone. Alternatively do you think chalcedony nucleated in free space surrounding the filament.
Line 362 some melting points are above zero and greater than the stated error. This should be mentioned and explained. Also applies to Apophylite.
Line 411 Figure 9. a/b no arrows.
Line 475 do you mean XRD?
Line 478 wt % oxide. What is the basis for the H2O calculation?
Line 486 are there reports of vanadium bearing minerals and their observed alteration. Magnetite?
Line 493 the data from NAS10 are very different should be mentioned here
Line 503 the numbers of figures and tables are transposed
Line 510 figure 12. The data points should show error bars. The pale grey data points are difficult to see, use different colours.
Line 515 move the tables to the relevant subsections to
Line 543 This whole paragraph needs to be redone to improve clarity
Line 566 Why is there only leaching of Fe from the basalt in the first phase when all the later fluids also go through basalt?
Line 570 Figure 4 suggests that the cavities are not completely filled. If they are completely filled why are only some filaments gravity controlled?
Line571 Were the filaments monomineralic at the time of initial formation and polymineralic after overgrowths? Could each phase of mineral growth be monomineralic?
Line 612 explain how the ferrihydrite is transformed to oxide/hydroxide and also how it is reduced to Fe2+ that is present in the silicate minerals.
Line 635 when conditions change from what to what, information needed.
Line 636 This last sentence does not make sense both several steps and smooth?
Line 639 Why are you taking examples from very pure silica rich rocks such as agate and flint rather than carbonate rocks
Line 644 what other deposits?
Line 645 the temperatures suggested here do not fit with the FI temperatures
Line 648 Figure 15. State what type of rocks and environment the data in the references come from
Line 660 Figure 16. Where do you get the temperatures for the types of calcite, they do not match the FI temperatures. Needs explanation. Is there also data from JAL-33 here?
Line 664 The temperatures from the FI data would put the data in the marine water field?
Line 672 this section needs a better explanation rather than multiple speculations
Line678 can any of this variation be caused by the method used? Again the explanation of the variation is weak.
Line 691 what is common zoning
Line 714 give the geological environment of the molybdenum deposits
Line 719 what is the significance of oxidising conditions? Were any of the phases reducing?
Line 734 Why does the age indicate hydrothermal origin, explain
Line 757 is there data on the variation in initial Sr ratio in Deccan basalts, surely this is relevant.
Line 775 Are the differences anything to do with closure temperatures of the different minerals?
Line 776 ok to say they are not related to magmatic age but give justification for hydrothermal process, ? referring back to previous section
Line 781 so do the ages really mean anything? Do your data?
Line 788 What events is the Ar loss caused by?
Line 792 Have you done multiple samples from apophylite close together in the same cavity to demonstrate that they give the same age. This would justify saying that different ages from closely spaced sights really are showing true ages otherwise it might be that the data are not very good?
Line 817 You must give a plausible explanation of the process to form the clay minerals. Is there direct evidence of this transformation, partial transformation?
Line 820 Why did these parameters change?
Line 823 Why meteoric water, stable isotopes?
Line829 how do you know this was burial?
Line 832 Explain why they cannot be due to burial metamorphism
Line 841 Sate the relationship of powellite to apophylite to justify this conclusion
Line 852 Do you mean the thickness of overlying basalt?
Line 858 do you burial metamorphism. Why use a different term here?
Line 872 Need a clearer explanation. Do you mean the minerals show no difference in composition despite being formed in several different episodes over a long period of time and in different places.
Line 886 be more specific to make your conclusions clearer
Author Response
Review report 2 Author answer (blue) to the general and single points
Open Review (x) I would not like to sign my review report
( ) I would like to sign my review report
English language and style ( ) Extensive editing of English language and style required
(x) Moderate English changes required
( ) English language and style are fine/minor spell check required
( ) I don't feel qualified to judge about the English language and style
Yes Can be improved Must be improved Not applicable
Does the introduction provide sufficient background and include all relevant references? (x) ( ) ( ) ( )
Is the research design appropriate? (x) ( ) ( ) ( )
Are the methods adequately described? (x) ( ) ( ) ( )
Are the results clearly presented? ( ) ( ) (x) ( )
Are the conclusions supported by the results? ( ) ( ) (x) ( )
Comments and Suggestions for Authors
The manuscript describes a series of phases of secondary mineralisation in the Deccan basalts at Savda. The descriptions of the samples are good and a considerable range of analytical techniques have been applied to document the sequence of mineralisation. They produce a model with a sequence of mineralisation that is plausible in general terms but is not fully justified in the information and data in the manuscript.
The analytical work give a range of results that do not hang together particularly well and a range of reasons are given to try to account for this. The explanations are not always clear and the end conclusion is that there is not a coherent widespread series of mineralisation events, rather that different processes happen at different times in different places in various degrees of intensity. One cannot help thinking that perhaps the data are not giving reliable results and this causes much of the variation. The authors need to improve the explanations to eliminate that possibility.
The manuscript is produced by authors who do not have English as a first language and in general it is fairly good. However in several places it needs to be improved to clarify the meaning and make it easier to read. These section are marked by a highlight on the commented manuscript.
Line 81 it would be useful to have a general picture of the quarry similar to Figure 11 here to give the reader a general impression of the setting
Such a general picture is not available. There is no elevated point you can take one photo representing several or all quarries. The quarry complex covers an area of approx. 3,000 x 500 m.
Line 97 what is the basis for geothermal gradient
A particular paleo-geothermal gradient for a particular regional or local position in the central DVP at a particular point in time between eruption and mineralization cannot be calculated properly due to lack of nessecary data. The only possibility for an approximate value was the comparison with similar volcanic areas with known or estimated values for paleo geothermal gradients. The volcanism, basalts and secondary minerals of Faroe Iceland are intensively researched, which makes sense for a comparison with the DVP. Jørgensen (2006) [18] plotted the distribution of sampled zeolites against the stratigraphic location and calculated the paleo geothermal gradient by using a scale for forming temperatures of zeolites. Jørgensen notes “Once the position and temperatures are known of the boundaries of the zeolite zones, the geothermal gradient and the altitude of the palaeosurface of the basalt plateau can be estimated using least squares regression, assuming a linear palaeotemperature gradient. However, in order to make a reliable estimate, the regression must be based on three or more zone boundaries.”
The paleo geothermal gradient in this paper is estimated for the time of max. burial by overlaying basalt at the end of volcanism in the studied area.
Line 107 Place names in text and figure have different spellings
The place names are given according to Sukheswala [11] and had been used in the past. In the map we used the recent ones. We add the recent ones in line 107 in brackets. Bombay (Mumbai), Poona (Pune) and Nasik (Nashik)
Line 117 explain diastrophism a term rarely used these days
Sabale and Vishwakarma [5] used the term diastrophism without an interpretation. We understood they believe that diastrophism covers tectonic processes like the movement of solid crust material, as opposed to movement of molten material, which is covered by volcanism.
Line 125 Figure 2 b needs some arrows to indicate the different minerals
will be added in the photos
Line 136 You use the phrase core line of lobes. Not clear what you mean. Lobes are projecting parts. Do you mean lobes projecting from the original lava flows or embayments within the main quarry wall? Clarify
To understand the morphology of the lava flows in the Savda quarry complex, it was necessary to study the quarry walls in several quarries. It was possible to recognize a morphology indicating individual extensive sheet lobes according to Bondre et al. 2004 [33]. In figure 3 you can see a curved structure of the lobe. We will change to flow.
Line 195 can cooling the sample first cause problems with later measurements?
After specific cooling, FIs were subsequently heated for phase identification and to determine the temperatures of phase transitions by using optical examinations.
Line 249 Why do you use lobe rather than just flow as this is just a flat quarry face?
See answer for line 136
Line 280 subsurface in respect to what, the flow as whole the individual cavity are there also surface filaments?
The filamentous fabrics were developed under the paleo-surface of the erupted basalts according to burial at a continuous increasing depth and not on the surface.
Line 285 Figure 4 caption. Is the shading meant to indicate that the lower part was liquid and the upper part vapour. By gravity controlled do you mean microstalactites?
The shading part represents a zone on the bottom of the cavity with helical filaments. The surface of this zone is irregularly shaped. Under results we do not explain possible causes for the macro morphology of the SFF. For excluding misinterpretation the top line of the shaded zone will be changed into irregular.
During formation of the minerals the cavities were completely filled with fluid and probably bacteria filaments were developed in the fluid without any direction. After transition of bacteria filaments to filamentous fabrics consisting of “heavy” iron-bearing clay minerals a part of the filamentous fabrics was connected with the cavity walls and hanging down vertically gravity controlled. Others built stable curved or helical formations or were sinking down on the cavity bottom. The vertical subsurface filamentous fabrics are not microstalactites.
Line 315 The formation of the filaments with the empty tube needs explanation. Do you think there was originally something there if so say so and suggest what it might have been and why it has gone. Alternatively do you think chalcedony nucleated in free space surrounding the filament.
We did not explain before the possible causes for the empty tube in results. This point is still under investigation for morphology and texture of the subsurface filaments. We believe that the empty tube is the result of a contracting of the the core filament. Such a process is still under investigation.
Line 362 some melting points are above zero and greater than the stated error. This should be mentioned and explained. Also applies to Apophylite.
Sorry, there were some minus signs missing in the numbers!!! We solved the problem.
Line 411 Figure 9. a/b no arrows.
The text passage was twice and belongs to Fig. 9 d; the text in the Figure caption was corrected.
Line 475 do you mean XRD?
is changed
Line 478 wt % oxide. What is the basis for the H2O calculation?
Theoretical H2O content was calculated based on stoechiometry of the formula with 8 H2O. The F-bearing position is completely filled by F for most analysis, only some contain minor amounts of OH, which was neglected.
Line 486 are there reports of vanadium bearing minerals and their observed alteration. Magnetite?
According to our knowledge the source for the vanadium in apophyllite (also for cavansite/pentagonite in the Pune/Wagholi area) from vanadium bearing minerals is not detected. The basalts from the DVP contain a vanadium content of approx. 200 – 400 ppm. Earlier comparisons by Ottens of 100 rock analyses of the basalt (in different publications of several authors) from the DVP have shown a significant correlation between V and Ti, indicating occurrence in the same mineral. Further investigations are necessary. Goldschmidt (1954) stated that the bulk of the vanadium is concentrated in magnetite and titano-magnetite of the igneous rocks. Pyroxenes are also believed to be a possible source.
Line 493 the data from NAS10 are very different should be mentioned here
Sample NAS-10 is a little bit problematic from the analytical point of few. The FIs were very thin and difficult to reproduce. As we have only one sample from this locality and we cannot give a reasonable explanation we do not use this data any more.
Line 503 the numbers of figures and tables are transposed
we changed it accordingly
Line 510 figure 12. The data points should show error bars. The pale grey data points are difficult to see, use different colours.
We changed the colors. In the Figure the error ellipses are shown. However, they are smaller than the plotted data points. Those of the lower intercept points are not visible.
Line 515 move the tables to the relevant subsections to
changed
Line 543 This whole paragraph needs to be redone to improve clarity
Rewritten
Line 566 Why is there only leaching of Fe from the basalt in the first phase when all the later fluids also go through basalt?
All elements including Fe and Mg necessary for forming of the stage I minerals had been leached in the first stage. It is remarkable that the later formed minerals in stage II and III do not contain Fe or Mg. We do not know and we did not state, whether there is also Fe and Mg leaching in stage II. But it is clear, that the minerals precipitated in stage II do not contain these elements. The reason for this different composition and missing elements Fe and Mg was not subject of the investigation. Iron bearing secondary minerals such as pyrite, ilvaite, babingtonite, pumpellyite or julgoldite have been detected in the spilite of the Malad quarry in Mumbai. But the genesis of these minerals is quite different to the secondary minerals in the main part of the DVP and is not subject of this paper.
Line 570 Figure 4 suggests that the cavities are not completely filled. If they are completely filled why are only some filaments gravity controlled?
Is explained before
Line571 Were the filaments monomineralic at the time of initial formation and polymineralic after overgrowths? Could each phase of mineral growth be monomineralic?
This is a very important point and cannot be answered completely. The diameter of the initial formed filaments seems to be less than 5 µm. It can be assumed that after transition from ferrihydroxide to an iron-bearing clay mineral the composition was monomineralic. First observations of the texture of filaments with diameters up to 30-50 µm indicate that the overgrowths do not reproduce a consistent sequence. This is still under investigation.
Line 612 explain how the ferrihydrite is transformed to oxide/hydroxide and also how it is reduced to Fe2+ that is present in the silicate minerals.
A detailed explanation of the process for transforming ferrihydrite to oxide/hydroxide and layer silicates was not recommended for this paper. As references we gave Konhauser [76], Tazaki [77], Hofmann [55,56], Posth [78] and Fortin [79].
Tazaki (1997) [71] explains “The microbial mats described here demonstrate that bacteria are capable of promoting both the absorption of elements and the crystallization of AI and Si in close proximity, so that layer silicates can be biogeochemically formed and immobilized simultaneously. Several studies of the synthesis and natural occurrence of layer silicates and siliceous ferrihydrites have provided considerable information about the nature of chemical reactions as compared to biomineralized reactions.”
Posth et al. [78] developed a conceptual model of biogenic Fe(III) production, Fe(III) mineral precipitation. The clay minerals investigated in this paper contain Fe3+ as well Fe2+.( Ferrosaponite Ca0.3(Fe2+,Mg,Fe3+)3((Si,Al)4O10)(OH)2 · 4H2O), (Celadonite K(Mg,Fe2+)Fe3+(Si4O10)(OH)2).
Line 635 when conditions change from what to what, information needed.
Mineral crystals can develop different habits in dependence on Eh/pH, p/T and chemical composition of the mineralizing fluids due to the preferred growth of crystallographic planes under specific conditions. In addition, these conditions influence the absolute rate (velocity) of crystal growth. Accordingly, changes in the growth conditions can result in dramatic variations of crystal habit such as in the case of the observed scepter growth of calcite. The calcite I scepters precipitated during ongoing burial, which includes variations in temperature, pressure, oversaturation and other parameters, which should not be explained in detail here. A thorough investigation of all parameters would require extensive laboratory experiments.
Line 636 This last sentence does not make sense both several steps and smooth?
Dublans81 [80] describes the development of scepters as a result of a drastic change of the environmental conditions. It could observed that the scepters in Savda were developed in several small steps indicating the fluid parameters did not change drastically, but probably in a more continuous manner.
Line 639 Why are you taking examples from very pure silica rich rocks such as agate and flint rather than carbonate rocks
Most carbonate rocks are formed either by precipitation of carbonate minerals in sedimentary environment (e.g. limestone) or from a carbonate-rich melt (carbonatite). However, in the case of the calcite mineralization within the secondary mineral sequence we have a formation mechanism based on alteration processes of volcanic host rocks. The used examples for comparison originate from the same type of formation mechanism. Schmitt-Riegraf [81], Götze et al. [82] and Blankenburg [83] also investigated calcite crystals from secondary mineralization sequences in volcanic host rocks. Together with calcite, silica minerals such as chalcedony or agate (banded chalcedony) can also be precipitated in another mineralization stage (similar to Savda). The example from marine environment (calcite associated with flint) was only included to show the complete different isotope signature for this environment.
To avoid confusion, the sentence was rewritten
Line 644 what other deposits?
The other deposits are those cited from literature.
Line 645 the temperatures suggested here do not fit with the FI temperatures
The FI temperatures and oxygen isotope data of calcite I and II cannot be explained under the assumption of the same mineral-forming fluid In contrast. The homogenization temperatures of Stage II calcite (Th 94-174 °C) associated with relatively high δ18O values require considerably higher δ18O values for the mineralizing fluid. The plot in Figure 17 shows that the calcite II generation seems to be crystallized from a dominantly magmatic fluid, whereas calcite I was probably influenced by a meteoric component.
à The whole paragraph was rewritten
Line 648 Figure 15. State what type of rocks and environment the data in the references come from
Figure 15 presents a comparison of oxygen and carbon isotope data of the investigated calcite samples with other calcites from secondary mineralization in volcanic rocks; data of carbonatite calcite and those of sedimentary origin (calcite associated with flint) were added to show the differences. The Figure caption was revised:
Line 660 Figure 16. Where do you get the temperatures for the types of calcite, they do not match the FI temperatures. Needs explanation. Is there also data from JAL-33 here?
There was a mistake in the preparation of Fig. 16. The Figure was corrected and redrawn. FI and isotope data of calcite II from sample JAL-33 are included in the data set.
Line 664 The temperatures from the FI data would put the data in the marine water field?
The FI and isotope data of calcite II would really result in a plot into the marine water field. However, it is also possible by mixing of meteoric and magmatic water.
Line 672 this section needs a better explanation rather than multiple speculations
The change in the isotopic composition from calcite I generation to calcite II required a change in the isotopic composition of the mineralizing fluid.
à The whole paragraph was rewritten
Line678 can any of this variation be caused by the method used? Again the explanation of the variation is weak.
We do not have indications for a methodical reason and at present there is also no better explanation for the variation.
Line 691 what is common zoning
Common growth zoning represents growth zones (growing crystallographic crystal planes), which are visible by weak differences in CL intensity due to slight changes in the crystallization conditions (e.g. slight variations of temperature or growth velocity during cooling). Such slight variations are common during a more or less continuous crystallization in contrast to drastic changes of the crystallization conditions.
Line 714 give the geological environment of the molybdenum deposits
The most important primary Mo-mineral in Mo-deposits is molybdenite in pegmatitic-pneumatolytic deposits associated with granites (typical Sn-W-Mo paragenesis, e.g. Ehrenfriedersdorf/Altenberg, Germany) or in high-temperature hydrothermal molybdenite-quartz veins (e.g. Climax, USA). Powellite is a secondary Mo-mineral formed mostly in the oxidation zone of primary Mo- deposits (e.g. Azegour, Morocco or Minussinsk, Russia).
Line 719 what is the significance of oxidizing conditions? Were any of the phases reducing?
Assuming formation of (at least some of) the minerals during burial, reducing conditions would be not surprising. The occurrence of late precipitated powellite is evidence for oxidizing conditions.
Line 731. What about the 62-65 oC of Nas-10. (not very hydrothermal).
We do not have an explicit interpretation for the lower temperatures for sample NAS-10. However, fluid inclusion data on apophyllite are scarce in the literature and therefore we think it might be useful to give also this aberrant sample to allow further discussion. NAS 10 values are cancelled
Line 734 Why does the age indicate hydrothermal origin, explain
The high homogenization temperatures of apophyllite, formed millions of years after eruption, cannot be explained by burial metamorphism. The late formation (age) together with the high temperatures indicate a late hydrothermal origin by temperature elevated circulating fluids. If the deposition of apophyllite would have happened by burial, it could be speculated that the high homogenization temperatures are resulting from high burial temperatures (thickness of overlaying basalt and paleo thermal gradient). Without the age you could also speculate the high homogenization temperature results from nearby dykes (Neuhoff et al. [7]). However, the influence of new dykes is limited to the time of volcanism. Therefore, the age is an important indicator.
736 change fine
changed to well developed
Line 757 is there data on the variation in initial Sr ratio in Deccan basalts, surely this is relevant.
There are some data on the initial Sr-isotopic ratios in the literature. They show some regional and stratigraphic variations. However we analyzed just one basalt sample and we are not able to contribute to this topic.
Line 775 Are the differences anything to do with closure temperatures of the different minerals?
There is nothing known about the closure temperature in apophyllite, stilbite or calcite. In any case the minerals with low Rb/Sr ratios are not critical. Several reasons argue against an influence of a closure temperature: The K-Ar and Ar-Ar ages are correlated. Therefore the closure temperature would have to be the same. In the quarry’s of Savda different ages appear in a distance of a few hundred meters without any tectonic activity or tilting in the area. Therefor the temperature was nearly similar in all sample localities at a certain time. This would imply a very distinct closure temperature of a few degrees to affect the samples in the western part much stronger than in the eastern part. A fluid activity along fractures at different times is much more likely. All investigated apophyllites are idiomorphic, water clear and show no signs of dissolution. For this reason alteration processes can be ruled out.
Line 776 ok to say they are not related to magmatic age but give justification for hydrothermal process? referring back to previous section
see explanation for line 734
Line 781 so do the ages really mean anything? Do your data?
Yes, we are sure the ages are valid (see discussion)!
We have experience in both methods for a long time. We repeated the analytics for sample JAL-16 apophyllite and the results are the same in error. Most important, there is a correlation of Rb-Sr and K-Ar ages (see also Line 775 closure temperature). There are just a few studies in similar minerals from basalts and the observations are very similar. In all this studies the authors discuss the data but at the end they argue for a geological significance.
See explanation for line 734. Without the ages for the apophyllite precipitation you would not be able to explain the high homogenization temperatures as result of late hydrothermal processes. The ages of apophyllite demonstrate that up to date unknown causes must be responsible for the elevated temperature of the fluids.
Line 788 What events is the Ar loss caused by?
This is an important question! Maybe this is due to the crystal structure. We hope our data initiate research activity on this problem! Apophyllite is also used to date low temperature mineralization in granitic rock (atomic waste repositories in Finland, Gotthard tunnel in Switzerland). The investigated apophyllites are much smaller and not that good developed. However, the results are interpreted to be geologically meaningful. In any case our data can contribute to further discussions.
Line 792 Have you done multiple samples from apophylite close together in the same cavity to demonstrate that they give the same age. This would justify saying that different ages from closely spaced sights really are showing true ages otherwise it might be that the data are not very good?
From one sample we analyzed core and rim and they are the same in error for Rb-Sr and K-Ar, even when the K-Ar data are younger. This data indicate that there is no zoning (Jal-13C). From another sample we made two analyses of different crystals (Jal-16B) and they are also the same in error (see text).
Line 817 You must give a plausible explanation of the process to form the clay minerals. Is there direct evidence of this transformation, partial transformation?
The alteration/transformation process of basaltic rocks to clay minerals (sheet silicates) is well established. Several investigations of natural volcanic rocks and laboratory experiments showed that clay minerals (sheet silicates) are quantitatively the most significant alteration minerals ( e.g., Kristmannsdottir [63], Seyfried and Bishoff [64]; Peretyazhko et al. [61], Pradeep et al. [66]). The fluids percolating the volcanic rocks contain all released elements of the host rocks, such as Si, Mg, Al and Fe2+ which are necessary for clay mineral formation. Iron-bearing saponite (magnesioferrosaponite) is a typical mineral representative of low-temperature alteration processes in basaltic rocks and was experimentally formed from volcanic glass and diabase at 150 °C in the laboratory.
Chapter 5.1. was rewritten and expanded with explanations of this process and references.
Line 820 Why did these parameters change?
Conditions of temperature, pressure and other parameters at surface after eruption und cooling change during burial. Temperature and pressure increase, chemical composition and pH are changing as a result of forming different minerals in the remaining fluids. The influence of burial to such parameters is widely accepted and known and should not be explained in the paper again.
Line 823 Why meteoric water, stable isotopes?
The crystallization started near the surface, so that a participation of meteoric water can be assumed. The isotope data indicate a mixing of meteoric and magmatic fluids.
Line829 how do you know this was burial?
The investigated lava flow had been buried after eruption by an estimated thickness of younger flows. Such a burial must have an influence on the mineralization conditions. The reported temperatures confirm the characteristic temperatures scale for zeolites by burial metamorphism under consideration of the burial depth (e.g. Jørgensen) [18]. According to their precipitation sequence they could not be formed during cooling of the basalt at the surface.
Line 832 Explain why they cannot be due to burial metamorphism
Burial metamorphism in Savda has to consider the roughly calculated max. burial depth and the estimated thermal temperature gradient. Both values must have changed and decreased over the timespan after eruption and should be accepted with a tolerance. The high homogenization temperatures of 200°C and more at up to 40 Ma years after eruption and erosion are not explainable by burial.
Rewritten
Line 841 Sate the relationship of powellite to apophyllite to justify this conclusion
rewritten
Line 852 Do you mean the thickness of overlying basalt?
The thickness (as explained for Nashik) of the former overlaying basalt can be one point. Other influence for the mineralization conditions could be for example the regional/local rock composition, like breccia as in Wagholi or high porous rocks with high permeability. The pillow lava in Mumbai was partly spilitized, resulting in a significant different mineralization. Long term observation of several outcrops in different parts of the DVP by the author Ottens led to the conclusion that characteristic mineralization occur in certain localities.
Line 858 do you burial metamorphism. Why use a different term here?
the term thermal overprint will be changed in metamorphism
Line 872 Need a clearer explanation. Do you mean the minerals show no difference in composition despite being formed in several different episodes over a long period of time and in different places.
Apophyllite was precipitated over a long time span, but doesn’t show significant differences in chemical composition. Apophyllite from the main part of the DVP is basically apophyllite (F) with minor (OH).
Line 886 be more specific to make your conclusions clearer
The conclusions are partly rewritten
Reviewer 3 Report
The manuscript by Ottens et al. is based on impressive sets of data including fluid inclusion, isotopic, and geochronology and must have involved considerable effort. I would like to recommend that the manuscript be published but not until important revisions are made. The major weaknesses are the serious lack of reference to important studies pertaining to secondary mineralization outside of the Deccan; a confused multi-stage interpretation that includes unlikely events but excludes some probable events; a very confusing interpretation of some calcite data (particularly Fig 16); and an absence of any quantitative data pertaining to mineral abundance in “cavities” of unspecified origin. Most of my suggestions are made on a line by line basis.
Line 2. Vesicles is a better choice than "cavities".
Line 23. Nowhere in the manuscript are there any clear descriptions of the “cavities”. A huge amount of literature is ignored pertaining to basalt cavities that have important diagnostic implications. How many of the sampled cavities are lava tubes, half-moon vesicles, pipe vesicles, molds, breccia cavities, microvesicles, vesicle cylinders, solution cavities, etc; and how big are they? Also “They are filled by..” should read “They are partially filled… ‘ otherwise they would be amygdules.
Lines 29-33. Nowhere in the manuscript is there any quantitative or semi-quantitative estimate or the abundance of any of these minerals. At minimum, a short table indicating which of these minerals is abundant, scarce, or rare is in order.
Line 67. Delete “to get”.
Line 95. A stratigraphic diagram would be a good idea. Are both sample locations (Savda and Nashik) buried under 1500 m of basalt and are both located about half way through the 3500 m total thickness? Was any sediment deposited on top of the Deccan and did any post Deccan igneous activity occur in the area? About how much of the Deccan was effected by secondary mineralization similar to your sample locations? About how widespread is secondary mineralization? Since you did not mention some of the minerals that the Deccan province is famous for (like prehnite and some unlisted zeolites) should we assume they are absent from your locations?
Line 252. What is the evidence for your interpretation that the layer above your flow consists of weathered basalt material? Chemical evidence, plant roots, soil texture? That may be important.
Line 274. What is the upper stability limit for some of your weathering products? How did they survive subsequent burial metamorphism and hydrothermal activity? or are they late weathering products?
Line 443. You almost totally ignore feldspar. Do you have secondary feldspar or not? I would expect some albite. Even in thin-section you can estimate plag composition.
Line 468. Delete “are”
Line 568. None of your vesicles are “caves” unless you sampled some large lava tubes. Delete the whole sentence.
Line 608. Where did you report evidence of carbon in mica and saponite? I didn’t see it.
Lines 643 – 692. This section is a total mess and should be carefully re-written.
Line 645. Fig 16 indicates that the higher O isotope values of Cal 11 compared to Cal 1 support higher temps not lower temps.
Line 658. The dominance of meteoric water is NOT supported by the isotopic data of Fig 16. None of the calcites plot in the field of meteoric water. Cal 1 plots within the field of marine water and Cal 11 plots off scale beyond the field of magmatic water.
Line 661. You give the false impression that meteoric water is represented by -10 isotopic units, etc; but that is only the case if T = 0 oC. At 100 oC, -10 units plots within the magmatic water field of Figure 16.
Line 665, 666. Again, the isotopic data indicates marine water for calcite 1, NOT meteoric. And you should refer to Table 4 and Figure 16 whenever you use them.
Line 668. After “…its hydrothermal origin” add “although calcite 11 plots at high temperatures beyond the magmatic water stability field on Figure 16.
Line 693. Again, nothing is said about primary or secondary feldspar in section 5.4. Didn’t you look at any of your thin sections? I would also be interested in the condition of the pyroxene. Is it still unaltered? Any chlorite? Did the meteoric, marine, and magmatic alteration have any effect on the basalt or was all your secondary mineralization totally confined to vesicles? You might not care but your readers would be interested.
Line 729. How do you know it was boiling? What was the pressure?
Line 731. What about the 62-65 oC of Nas-10. (not very hydrothermal).
Line 799. What is zeo 1, zeo 11, and pow plotted on the figure? Where do you find apophyllite temps in the 150 to 275 oC range? I can only find 62-65 and 141-222 oC in Table 4. Why can’t you increase the temp of the basalt to at least the 222 oC of the apophyllite and the 250 oC of the Calcite 11 samples of Table 4. You have 1500 m of hot igneous activity overlying your flow lobe. Don’t you think some hot hydrothermal activity would be likely during all that time? It is very likely that the Deccan province was a hot geothermal field at depth very much like other active geothermal fields and particularly like other major continental flood basalts such as the Newark Basin portion of CAMP described by Puffer et al (2018), Bull. of Volcanology; Puffer and Laskowich (2012), J. Volcan. or the Hartford Basin portion by Greenberger et al. (2015), Geochim Cosmochim; or Iceland described by Franzson et al. (2008) J Volcan. Several additional studies of secondary mineralization have been recently published that should be considered. Clearly the Deccan must have existed as a deep heated geothermal system throughout its long active duration and during subsequent burial.
Line 806. If you use “Stage O” you will need to change everything to 4 stages.
Line 855. Laumontite is very diagnostic. Why isn’t it on your Fig 13 if it is common in the region?
Line 860. How can you have all three stages at the top of
the volcanic pile if deep burial is required for stage 2? Re-write the
sentence.
Author Response
Review report 3 Author answer (blue) to the general and single points
(x) I would not like to sign my review report
( ) I would like to sign my review report
English language and style ( ) Extensive editing of English language and style required
( ) Moderate English changes required
(x) English language and style are fine/minor spell check required
( ) I don't feel qualified to judge about the English language and style
Yes Can be improved Must be improved Not applicable
Does the introduction provide sufficient background and include all relevant references? ( ) ( ) (x) ( )
Is the research design appropriate? ( ) (x) ( ) ( )
Are the methods adequately described? (x) ( ) ( ) ( )
Are the results clearly presented? ( ) (x) ( ) ( )
Are the conclusions supported by the results? ( ) (x) ( ) ( )
Comments and Suggestions for Authors
The manuscript by Ottens et al. is based on impressive sets of data including fluid inclusion, isotopic, and geochronology and must have involved considerable effort. I would like to recommend that the manuscript be published but not until important revisions are made. The major weaknesses are the serious lack of reference to important studies pertaining to secondary mineralization outside of the Deccan; a confused multi-stage interpretation that includes unlikely events but excludes some probable events; a very confusing interpretation of some calcite data (particularly Fig 16); and an absence of any quantitative data pertaining to mineral abundance in “cavities” of unspecified origin. Most of my suggestions are made on a line by line basis.
We will give additional reference to secondary mineralization in other basalt provinces in discussion.
Line 2. Vesicles is a better choice than "cavities".
No uniform usage of the terms vesicle, amygdules, geodes, or cavities was noted before (Rakovan, amygdule). Specifically, the dividing line between what we would call an amygdule and a mineralized pocket, and the degree of vesicle infilling necessary to be called an amygdule, is not precisely defined—although, as already mentioned, vesicles and hence amygdules are usually less than 2 inches in size.
As one definition was given: A small cavity in a volcanic rock that was formed by the expansion of a bubble ….. When vesicles are filled with a secondary alteration mineral, they are called amygdules. The term vesicle is predominantly used for small, spherical cavities in volcanic rock, produced by bubbles of air or gas in the molten rock. Our investigation is based on the mineralization in big cavities.
We will explain in chapter 2.2 Distribution and mineralization …that we use the term cavity in contrast to small vesicles (as in line 252/258) for large ones, partly with secondary filling.
Line 23. Nowhere in the manuscript are there any clear descriptions of the “cavities”. A huge amount of literature is ignored pertaining to basalt cavities that have important diagnostic implications. How many of the sampled cavities are lava tubes, half-moon vesicles, pipe vesicles, molds, breccia cavities, microvesicles, vesicle cylinders, solution cavities, etc; and how big are they? Also “They are filled by..” should read “They are partially filled… ‘ otherwise they would be amygdules.
will be explained together with answer to Line 2 in chapter 2.2.
Lines 29-33. Nowhere in the manuscript is there any quantitative or semi-quantitative estimate or the abundance of any of these minerals. At minimum, a short table indicating which of these minerals is abundant, scarce, or rare is in order.
we added a new table
Line 67. Delete “to get”.
deleted
Line 95. A stratigraphic diagram would be a good idea. Are both sample locations (Savda and Nashik) buried under 1500 m of basalt and are both located about half way through the 3500 m total thickness? Was any sediment deposited on top of the Deccan and did any post Deccan igneous activity occur in the area? About how much of the Deccan was effected by secondary mineralization similar to your sample locations? About how widespread is secondary mineralization? Since you did not mention some of the minerals that the Deccan province is famous for (like prehnite and some unlisted zeolites) should we assume they are absent from your locations?
A stratigraphic diagram representing a section Nashik – Jalgaon/Savda is not available. The Nashik sample location is close to the area with the highest and youngest formations and steep hills which did not occur in the Savda region. It is assumed that the erosion of steep hilly formations at Nashik over a long time span had been higher as in Savda. The examined apophyllite sample was collected from a quarry (610 m asl, 19° 54,426'N 73° 56,602'E), where laumontite occurs too. In other quarries around Nashik in an altitude between 650 to 720 m asl, laumontite was not observed. This leads to the estimation, that the lower Nashik formations had been buried more, resulting in burial metamorphism necessary for laumontite precipitation. In Savda the max. burial depth had been less.
Alluvial sediments of weathered basalt occur more or less on the top of the basalt in the valleys and plains, recently no thick alluvial layers and no other sediments are in the areas of the quarries at Savda or Nashik. Neither any post Deccan igneous activity did occur in the central part of the DVP and nor intertrappeans are noted.
The secondary mineralization in the main part of the DVP is widely present in the vesicular compound flows. The distribution of certain species differs significantly between different outcrops. Subsurface filamentous fabrics were observed in several parts of the Central DVP. 1 to 3 calcite generations are observed in numerous localities. Calcite I twins like in Savda occur very limited in an area Jalgaon-Jamner. Apophyllite is very common throughout the DVP/compound flows. A general mineralization sequence by burial was excluded by Jeffery [12]. The most frequent occurring minerals such as stilbite heulandite or scolecite occur in all stratigraphic units in all altitudes. Some minerals like prehnite or yugawaralite are restricted to the spilite in the Mumbai area. Mesolite is restricted mainly to Pune-Lonavala area, where chalcedony is absent. Powellite is observed in different localities in different formations. It is believed that a wide variation of porosity and permeability plays an important role for different alteration and mineralization.
Line 252. What is the evidence for your interpretation that the layer above your flow consists of weathered basalt material? Chemical evidence, plant roots, soil texture? That may be important.
By personal observations of the authors. Between the layers of massive basalt in some places layers composed of slightly weathered basalt components in a fine-grained matrix (diamict) occur. In some places, a gradation with respect to the grain size is visible. The quarry complex is situated on a slightly sloping terrain, approx. 10 m from 230 m NE to SW 220 m asl.
Line 274. What is the upper stability limit for some of your weathering products? How did they survive subsequent burial metamorphism and hydrothermal activity? or are they late weathering products?
Assuming weathering as a surface process, the secondary minerals are not real weathering products, but more products of hydrothermal alteration processes. Investigations of recent alteration processes in geothermal systems as well as laboratory experiments showed that smectites (Fe-rich saponite) and zeolites form at temperatures below 200 °C (e.g. Kristmannsdottir [63], Seyfried & Bischoff [64]). Formation temperatures for heulandite, stilbite and mordenite of 70 to 200 °C are reported by several authors (e.g. Jørgensen [18]). Above 200 °C, most zeolites disappear and smectites may transform into mixed-layer clay minerals and swelling chlorite. The zeolites formed during burial could easily survive later hydrothermal activities up to 250°C.
Line 443. You almost totally ignore feldspar. Do you have secondary feldspar or not? I would expect some albite. Even in thin-section you can estimate plag composition.
We reported feldspar as plagioclase in chapters 4.2.5 and 5.4. The detected feldspar had been the only one observed as secondary formation and occurred in a filamentous fabric under the chalcedony crust. Plagioclase occurs commonly as phenocrysts, as a major component in the unaltered whole rock in the basalt from Savda and was confirmed by former investigations (thin section) by the authors. This paper is focused on secondary mineralization and doesn’t include such rock analyses.
Studying other papers on the secondary minerals in vesicles and cavities of the DVP, you cannot find any hint for secondary feldspar in vesicles or cavities.
Line 468. Delete “are”
deleted
Line 568. None of your vesicles are “caves” unless you sampled some large lava tubes. Delete the whole sentence.
The sentence shall point out that minerals and filamentous fabrics occurring in cavities as investigated are not from caves and therefore, they are not speleothems.
Line 608. Where did you report evidence of carbon in mica and saponite? I didn’t see it.
EDX analyses of the secondary minerals revealed that sheet silicates associated with the filament sometimes contain considerable amounts of carbon (up to several wt%). Because of the carbon coating of the samples and thin sections, the amount of carbon could not be quantified. Considering the average amount of carbon related to the carbon coating, the excess amount could reach several wt%.
We have added an explanation in section 4.2.2.
Lines 643 – 692. This section is a total mess and should be carefully re-written.
The whole section is completely rewritten.
Line 645. Fig 16 indicates that the higher O isotope values of Cal 11 compared to Cal 1 support higher temps not lower temps.
Fig. 16 was redrawn because of a mistake in the earlier version. In general the Figure shows overlapping temperatures for the two calcite generations with a tendency of a broader temperature range for calcite II. A third calcite generation shows slightly higher homogenization temperatures for fluid inclusions. Because of the lack of isotope data, this calcite III is not included in Fig. 16.
Line 658. The dominance of meteoric water is NOT supported by the isotopic data of Fig 16. None of the calcites plot in the field of meteoric water. Cal 1 plots within the field of marine water and Cal 11 plots off scale beyond the field of magmatic water.
The new Fig. 16 shows that calcite II plots close to the magmatic water line, whereas calcite I plots close to marine water. Because of the geological background there is no indication for the participation of marine water in the formation process. Therefore, we assume that we have a result of mixing of meteoric and magmatic fluids.
Line 661. You give the false impression that meteoric water is represented by -10 isotopic units, etc; but that is only the case if T = 0 °C. At 100 oC, -10 units plots within the magmatic water field of Figure 16.
This is corrected in the rewritten paragraph.
Line 665, 666. Again, the isotopic data indicates marine water for calcite 1, NOT meteoric. And you should refer to Table 4 and Figure 16 whenever you use them.
The sentence was removed and the references to Table 4 and Fig. 16 included in the revised version of the paragraph.
Line 668. After “…its hydrothermal origin” add “although calcite 11 plots at high temperatures beyond the magmatic water stability field on Figure 16.
The sentence was rewritten.
Line 693. Again, nothing is said about primary or secondary feldspar in section 5.4. Didn’t you look at any of your thin sections? I would also be interested in the condition of the pyroxene. Is it still unaltered? Any chlorite? Did the meteoric, marine, and magmatic alteration have any effect on the basalt or was all your secondary mineralization totally confined to vesicles? You might not care but your readers would be interested.
Our investigations had been focused and limited on the secondary minerals in the cavities. We present a paper which gives for the first time a mineralization model for a certain locality in the DVP on basis of comprehensive analytical work with modern analytical equipment. We know there are still many questions and points, which need further work. We hope that our paper will initiate intensive research for the secondary mineralization of the DVP.
In the only paper about the composition of the basalts in the Central DVP Melluso [35, p588-591] states: “The lava sequence of the central-western Deccan Traps (from Jalgaon towards Mumbai) is formed by basalts and basaltic andesites having a significant variation in TiO2 (from 1.2 to 3.3 wt%), Zr (from 84 to 253 ppm), Nb (from 5 to 16 ppm) and Ba (from 63 to 407 ppm), at MgO ranging from 10 to 4.2 wt%. Most of these basalts follow a liquid line of descent dominated by low pressure fractionation of clinopyroxene, plagioclase and olivine ……. “
A different mineralization of the different flow zones “upper, core line and lower part” was noted by field observations - see description in figure 3 (4 new text)). The upper intensely altered zone is rich in mm to serval cm sized vesicles and probably the main source for the immigrating fluids and mineralization in the big cavities in the core line. The cavities in the core line are surrounded by rather dens unaltered rock, without clear visible aureoles to 1 cm. The cavities are connected to the upper zone by partly mineralized cracks. Several XRD analyses did not gave evidence for chlorite. Meteoric and magmatic alteration (marine not present) seems to have not affected the dense rock in the core line in Savda.
Line 729. How do you know it was boiling? What was the pressure?
Srikantappa and Mookherjee [13] [91] indicate a genesis at boiling conditions “Presence of pure water inclusions with small vapor phase indicates trapping of hot water in cavity minerals in Deccan basalts. Occurrence of vapor-rich inclusions (homogenization to gas phase), empty inclusions and water inclusions (Th to liquid phase) in some of the secondary minerals suggest “boiling“ conditions during fluid entrapment. Different types of fluids recorded in many cavity minerals in basalts points to volatile-rich nature of Deccan basalts (DVP). The fluids were boiling during the formation of cavity minerals which crystallized at temperatures ranging from 250 to 280°C.”
Line 731. What about the 62-65 °C of Nas-10. (not very hydrothermal).
Sample NAS-10 is a little bit problematic from the analytical point of few. The FIs were very thin and difficult to reproduce. As we have only one sample from this locality and we can not give a reasonable explanation do not use this data any moreLine 799. What is zeo 1, zeo 11, and pow plotted on the figure?
Zeo I are the zeolites stilbite heulandite and mordenite of stage II. Will be explained.
Where do you find apophyllite temps in the 150 to 275 oC range? I can only find 62-65 and 141-222 oC in Table 4.
The high homogenization temperature for apophyllite from Savda has been reported by Srikantappa and Mookherjee [13] as noted in chapter 5.5
Why can’t you increase the temp of the basalt to at least the 222 °C of the apophyllite and the 250 °C of the Calcite 11 samples of Table 4. You have 1500 m of hot igneous activity overlying your flow lobe. Don’t you think some hot hydrothermal activity would be likely during all that time.
After the max. burial of 1500 m and ending of the volcanic activity the temperature in the investigated flow reached approx. 150°C. The decreasing basalt temperature represents the influence by decreasing burial and decreasing geothermal gradient after the end of volcanic activity. It is very unlikely that at a deposition of apophyllite after 20 to 40 Ma later the basalt temperature was 200 to 250°C. The high temperature millions of years after the end of volcanism are best explained by hot circulating fluids. Such hot fluids could occur all the time, but locally different and not permanently.
It is very likely that the Deccan province was a hot geothermal field at depth very much like other active geothermal fields and particularly like other major continental flood basalts such as the Newark Basin portion of CAMP described by Puffer et al (2018), Bull. of Volcanology; Puffer and Laskowich (2012), J. Volcan. or the Hartford Basin portion by Greenberger et al. (2015), Geochim Cosmochim; or Iceland described by Franzson et al. (2008) J Volcan. Several additional studies of secondary mineralization have been recently published that should be considered. Clearly the Deccan must have existed as a deep heated geothermal system throughout its long active duration and during subsequent burial.
We agree that the DVP should have been a deep seated geothermal field during the volcanic activity. It is accepted that the eruption had been through dykes, predominantly by 3 dyke swarms (e.g. Sheth, Building a continental flood basalt province: Key significance of the Deccan Trap dyke swarms)
It can be estimated that the thermal activity was decreasing after the end of volcanism while rifting of the Indian plate northwards and the paleo-geothermal gradient was decreasing too. Under consideration of the late precipitation ages of apophyllite at 40 Ma after eruption at significant lower geothermal gradients and by erosion reduced thickness of overlaying basalt, a formation of apophyllite at temperatures to approx. 250°C by burial metamorphism should be excluded. For answering the question for the heat source of the fluids with temperatures to approx. 250°C upwelling heat flows can give probably the answer. Such an explanation needs further investigation and is not part of this paper.
Line 806. If you use “Stage O” you will need to change everything to 4 stages.
deleted stage 0
Line 855. Laumontite is very diagnostic. Why isn’t it on your Fig 13 if it is common in the region?
Laumontite is common in the Nashik and Mumbai area, but not at most other localities as Pune, Aurangabad and Savda. Mumbai mineralization is different to the main part of the DVP and occurred in a spilite, which has not been influenced by burial metamorphism (no significant overlaying volcanic rock). Mineralization in Mumbai is influenced by spilitization of pillow lava.
It can be assumed that laumontite from the outcrops in the lower Nashik area, where the mineral is common, results from higher burial. Most laumontite in Nashik is precipitated before stilbite and heulandite, but rather rare on apophyllite as a second generation. The thickness of overlaying basalt in Savda was roughly calculated, which is not enough for laumontite formation by burial at temp. more than approx. 200°C.
Line 860. How can you have all three stages at the top of the volcanic pile if deep burial is required for stage 2? Re-write the sentence.
sentence is deleted
Round 2
Reviewer 2 Report
The manuscript has been revised and many improvements made. However several importabnt scientific points have still not been addressed and this must be done to make it suitable for publication.
Line 117 explain diastrophism a term rarely used these days
You need to explain what you mean.
Line 280 subsurface in respect to what, the flow as whole the individual cavity are there also surface filaments?
The authors introduce the term subsurface filamentous fabrics. I do not think this is either correct or appropriate for the features described in the paper. A fabric is the arrangement of the components in a multi component body not the individual components. These are filamentous structures and in particular the paper draws attention to single filaments not aggregates. They quote papers by Hoffmann et al to justify the use of the term but Hoffmann described micron sized mats of filaments, those in this paper are orders of magnitude larger. Are these really the same type of thing? Calling them subsurface is also misleading as although the cavities within the basalt are below the ground surface the filaments are actually growing on the inner surface of the cavities not below the surface of the place where they are growing.
Line 285 Figure 4 caption. Is the shading meant to indicate that the lower part was liquid and the upper part vapour. By gravity controlled do you mean microstalactites?
A better explanation of the formation of the various types is needed within the paper. There is no need for a shaded section at the bottom of the cavity it is misleading.
Line 315 The formation of the filaments with the empty tube needs explanation. Do you think there was originally something there if so say so and suggest what it might have been and why it has gone. Alternatively do you think chalcedony nucleated in free space surrounding the filament.
There must be some discussion in the paper even if it cannot be adequately explained. You cannot just brush it under the carpet.
Line 475 do you mean XRD?
Line 478 wt % oxide. What is the basis for the H2O calculation?
The method used should be included in the paper
Line 486 are there reports of vanadium bearing minerals and their observed alteration. Magnetite?
Add a sentence of explanation for readers not familiar with the trace element geochemistry of basalts
Line 543 This whole paragraph needs to be redone to improve clarity
Line 566 Why is there only leaching of Fe from the basalt in the first phase when all the later fluids also go through basalt?
Again you need some discussion you cannot just ignore this
Line 570 Figure 4 suggests that the cavities are not completely filled. If they are completely filled why are only some filaments gravity controlled?
Line 612 explain how the ferrihydrite is transformed to oxide/hydroxide and also how it is reduced to Fe2+ that is present in the silicate minerals.
Some discussion is needed in the paper
Line 635 when conditions change from what to what, information needed.
State the types of parameter changes involved
Line 636 This last sentence does not make sense both several steps and smooth?
Improve wording
Line 639 Why are you taking examples from very pure silica rich rocks such as agate and flint rather than carbonate rocks
Much improved, you could still include the plot of marine carbonate to highlight the difference
Line 691 what is common zoning
Line 714 give the geological environment of the molybdenum deposits
You need to clarify that Mo is associated with felsic intrusions significant amounts are not present in basalt, not basalt. This is why it needs further investigation.
Line 734 Why does the age indicate hydrothermal origin, explain
State that the age indicates formation long after erruption
Line 757 is there data on the variation in initial Sr ratio in Deccan basalts, surely this is relevant.
You should not speculate that the variation in Sr ratio is due to variation in the host basalt if there is no data about the host basalt.
Line 788 What events is the Ar loss caused by?
You should add a sentence of discussion or explanation even if there is no clear answer.
Line 817 You must give a plausible explanation of the process to form the clay minerals. Is there direct evidence of this transformation, partial transformation?
Are clay minerals present in the host basalt, if so state this, if not add explanation
Line 823 Why meteoric water, stable isotopes?
Add your explanation to the text
I made several minor comments directly on the previous manuscript as well as indicating places where the English needed improving. Many of these have not been addressed.
Author Response
Review 2 Report Reviewer 2
Comments and Suggestions for Authors
The manuscript has been revised and many improvements made. However several importabnt scientific points have still not been addressed and this must be done to make it suitable for publication.
Considering your first revision report several times it was not clear, whether you like to get just more information for yourself or you wanted additional explanations or corrections within the manuscript. We believe that in your second review report are again a few comments or questions which are already explained in the paper or are not focused on the main topic and not necessary to explain. Some explanations are asked, which can be given only by speculations or further very specific research activities. Especially considering the sub-surface filamentous fabrics (SFF), which are not described before from the DVP, we do not like to speculate. We prefer to describe the discovered facts and to give hints for further research.
The second revised version of the paper contains several additions and explanations, which give answers to the following questions. For a more easy understanding we marked the main changes of the first update in grey and the second ones in yellow.
Line 117 explain diastrophism a term rarely used these days
You need to explain what you mean.
The term diastrophism is used by the cited author Sabale and concerns to a part of geotectonics (just as folding and faulting). We added the term geotectonics.
Line 280 subsurface in respect to what, the flow as whole the individual cavity are there also surface filaments?
The authors introduce the term subsurface filamentous fabrics. I do not think this is either correct or appropriate for the features described in the paper. A fabric is the arrangement of the components in a multi component body not the individual components. These are filamentous structures and in particular the paper draws attention to single filaments not aggregates. They quote papers by Hoffmann et al to justify the use of the term but Hoffmann described micron sized mats of filaments, those in this paper are orders of magnitude larger. Are these really the same type of thing? Calling them subsurface is also misleading as although the cavities within the basalt are below the ground surface the filaments are actually growing on the inner surface of the cavities not below the surface of the place where they are growing.
Subsurface is already explained in chapter 4.2.2. of the last revised version as subterraneous environments. In Figure 6a and 7a (last version) photographs of both, single SFF and SFF mats were shown. The SFF as described are arrangements of different components (clay minerals, zeolites, and chalcedony) and do not consist of one component only. We explained also “The silica zone commonly encloses several filaments and develops a more or less round cross-section with proceeding precipitation”, which are aggregates. You can see in Figure 14 (last version) the diameter of the core filament measures approx. 30 µm, and this is probably not the innermost and has to be investigated in future. The SFF described in our paper are basically the same as published by Hofmann. Hofmann described 3 phases for filament formation. A) Initial precipitation forming filaments of approximately 1 µm (B) Deposition of later minerals on filaments, typically increasing the diameter to 20-40 µm, (C) Final cementation, leading to stalk-like forms up to several cm in diameter and several tens of cm long. The size of the SFF explained in our paper and by Hofmann is the same of type B and C. Type A is not investigated. One exceptional feature of the SFF from the DVP seems to be the length and the thick variable mineral assemblage (clay minerals, mordenite and chalcedony) around the innermost filament. We added in the first update the explanation by Hofmann [63,64]: Hofmann defines the term subsurface filamentous fabrics (SFF) as microscopic to macroscopic mineral fabrics that result from the precipitation of minerals on a substrate of filamentous (thread-like) geometric units in subterraneous environments. The investigated SFF in our paper fulfill the same parameters.
Line 285 Figure 4 caption. Is the shading meant to indicate that the lower part was liquid and the upper part vapour. By gravity controlled do you mean microstalactites?
A better explanation of the formation of the various types is needed within the paper. There is no need for a shaded section at the bottom of the cavity it is misleading.
Shading means a zone in the lower part of the cavity filled completely with solution (explained in 5.2), where predominantly intergrowths of helical filaments occur and are grown together, forming a moss agate like structure. The caption will be completed. SFF are not stalactites or micro-stalactites as explained in discussion 5.2. SFF formed in the solution morphologies with a variable orientation as explained in 4.2.2. It was observed, that in many cavities you could find SFF around the cavity wall without gravity controlled orientation. Gravity controlled growth of SFF is only one variety. We believe that the explanation within the “Results” chapter (4.2.2.) is quite clear. The formation of stalactites, self-organized assembling and biomorphs versus bio mineralization is discussed in chapter 5.2. SFF from the DVP with a probably biogenetic history are reported for the first time in our paper. It cannot expected to give all answers for their genesis in this paper. Because of the complex processes leading to the formation of SFF, our results are first results and first explanations, but, of course, need more studies in future.
Line 315 The formation of the filaments with the empty tube needs explanation. Do you think there was originally something there if so say so and suggest what it might have been and why it has gone. Alternatively do you think chalcedony nucleated in free space surrounding the filament.
There must be some discussion in the paper even if it cannot be adequately explained. You cannot just brush it under the carpet.
We have already added an explanation in chapter 5.2 (old version) to point out the problem. However, the empty tube doesn’t have an influence for the whole mineralization model.
Line 475 do you mean XRD?
Yes, we changed it now also in the text.
Line 478 wt % oxide. What is the basis for the H2O calculation?
The method used should be included in the paper
We added the text from the letter in the Table caption.
Line 486 are there reports of vanadium bearing minerals and their observed alteration. Magnetite?
We didn’t notice any reports about vanadium bearing minerals in the basalt of the Central DVP.
Add a sentence of explanation for readers not familiar with the trace element geochemistry of basalts We added a sentence for explanation:
“The V content of basalts in the DVP is in general in the range of 200 – 400 ppm (24).”
Line 543 This whole paragraph needs to be redone to improve clarity
The paragraph is already rewritten in the last updated paper.
Line 566 Why is there only leaching of Fe from the basalt in the first phase when all the later fluids also go through basalt?
Again you need some discussion you cannot just ignore this
In line 566 it was not stated that leaching of Fe and Mg is limited to the stage of the precipitation of clay and other secondary minerals. It is explained that the observed minerals, precipitated after the clay minerals, do not contain important concentrations of the two elements.
In the updated version we added a new chapter 5.1. about alteration, which will explain the leaching of Fe and Mg from glass and olivine in the first phase of alteration.
Line 570 Figure 4 suggests that the cavities are not completely filled. If they are completely filled why are only some filaments gravity controlled?
It is explained in former updated paper chapter 4.1 “The visual observations of the cavities in the quarry indicate that the big cavities were completely filled by fluids during the time of precipitation, because all minerals crystallized all over the cavity walls without any specific crystallization direction.”
A discussion about the fact, why some filaments show a gravity controlled morphology and others not, is already given roughly in the revised version on chapter 5.2 lines 700-717.
Line 612 explain how the ferrihydrite is transformed to oxide/hydroxide and also how it is reduced to Fe2+ that is present in the silicate minerals.
Some discussion is needed in the paper
The transformation of ferrihydrite to oxide/hydroxide and its incorporation into Fe-rich minerals such as clay minerals are complex processes, which cannot be described in detail within this paper. The occurrence of these transformation processes is in general accepted and references are given by Hofmann [64] and Fortin [89] as well as Baldermann [61] and Treiman [75] Tazaki [87], where these processes are discussed in detail.
Line 635 when conditions change from what to what, information needed.
State the types of parameter changes involved
A new sentence is incorporated for explanation:
Development of scepters seems to be a common process while oversaturation takes place and environmental conditions (e.g. pH, T, chemical composition) change significantly [82].
Line 636 This last sentence does not make sense both several steps and smooth?
Improve wording
The calcite scepters from the Yucca mountains had been formed according to authors by a drastic change in the anisotropy of the growth rate, which is required to produce a change in the growth pattern from predominantly bladed to blocky scepters. In contrast, the scepters from Savda do not show a single significant step for the scepter formation. Several small steps, sometimes difficult to distinguish, indicate a rather continuous change of the parameters, which gives an impression of a smooth change. We change smooth to slowly.
Line 639 Why are you taking examples from very pure silica rich rocks such as agate and flint rather than carbonate rocks
Much improved, you could still include the plot of marine carbonate to highlight the difference
The presented data show a comparison of oxygen and carbon isotope data of the investigated calcite I and calcite II generations from Savda with calcites from comparable secondary mineralization in volcanic rocks of other locations as well as hydrothermal vein mineralization. These calcites are not hosted in silica-rich rocks, but are also secondary alteration/mineralization products in basic volcanic rocks.
Line 691 what is common zoning
Common growth zoning represents growth zones (growing crystallographic crystal planes), which are visible by weak differences in CL intensity due to slight changes in the crystallization conditions (e.g. slight variations of temperature or growth velocity during cooling). Such slight variations are common during a more or less continuous crystallization in contrast to drastic changes of the crystallization conditions.
Line 714 give the geological environment of the molybdenum deposits
In general molybdenum deposits (molybdenite) are related to veins and impregnations in pegmatitic-pneumatolytic environment (often related to granites) or high-temperature hydrothermal conditions. It is not assumed that molybdenum deposits are present in basalts. Within the secondary mineralization in the DVP we have only small crystals of powellite (that is normally in the oxidation zone of primary molybdenum deposits), but not a Mo deposit.
You need to clarify that Mo is associated with felsic intrusions significant amounts are not present in basalt, not basalt. This is why it needs further investigation.
See explanation above.
Because of the common association of Mo with more felsic rocks, the occurrence of powellite in basalts is surprising.
We explained that in the new manuscript versionchapter 5.5 in lines 1019-1031:
“A conspicuous feature of the secondary mineralization in several localities of the central DVP is the occurrence of powellite [68,69,103]. It is rarely present as a few small crystals in the cavities, very rarely up to 5 cm in size and grown on stilbite-Ca and overgrown by apophyllite. Although specific data concerning the Mo content for the DVP are not available, results of Liang et al. [104] show that Mo is present only at a low ppm level (< 5 ppm) in basalts. Therefore, the formation of big powellite crystals in the cavities of Mo-poor basalts is surprising. In general, powellite is only common in the oxidation zone of molybdenum deposits related to more felsic rocks. In the investigated localities, the occurrence of powellite could probably be related to the release of Mo from the basalts during alteration of large basalt masses by fluids and its supply and accumulation by the hydrothermal fluids in the cavities. The powellite precipitation is noteworthy, since the molybdenum must have been present in the same fluids as the elements necessary for the formation of stilbite. This point needs further investigation. The presence of powellite at least points to oxidizing conditions during crystallization.”
Line 734 Why does the age indicate hydrothermal origin, explain
As explained in chapter 5.7 stage III, burial metamorphism (by decreased geothermal gradients and eroded overlaying basalts as explained) at the high homogenization temperatures during the late age, millions of years after eruption is not possible. The only possibility for formation is by late hydrothermal processes.
State that the age indicates formation long after eruption
In the chapter introduction was explained that the flood basalts of the DVP erupted through fissures between about 67.5 to 60.5 Ma ago. Formation ages for apophyllite with a range of 20 to 45 Ma, 21 to 59 Ma respectively were noted. Such ages emphasize a long time after eruption.
Line 757 is there data on the variation in initial Sr ratio in Deccan basalts, surely this is relevant.
You should not speculate that the variation in Sr ratio is due to variation in the host basalt if there is no data about the host basalt.
According to your suggestion we removed the discussion on the initial ratios of the age regression lines.
Line 788 What events is the Ar loss caused by?
You should add a sentence of discussion or explanation even if there is no clear answer.
Here again the answer from the letter to the first review: This is an important question! Maybe this is due to the crystal structure. We hope our data initiate research activity on this problem! Apophyllite is also used to date low temperature mineralization in granitic rock (atomic waste repositories in Finland, Gotthard tunnel in Switzerland). The investigated apophyllites are much smaller and not that good developed. However, the results are interpreted to be geologically meaningful. In any case our data can contribute to further discussions.
We added a sentence.
Line 817 You must give a plausible explanation of the process to form the clay minerals. Is there direct evidence of this transformation, partial transformation?
Are clay minerals present in the host basalt, if so state this, if not add explanation
The alteration processes resulting also in the formation of the clay minerals are discussed in the new chapter 5.1. Alteration processes.
Line 823 Why meteoric water, stable isotopes?
Add your explanation to the text
Using the homogenization temperatures of fluid inclusions (Table 5) and the corresponding δ18O data (Table 6), the field of calcite I plots close to the fractionation curve of oxygen from marine water. However, there is no indication from the geological background of the DVP concerning marine influence during formation of the secondary mineralization. Therefore, it is most likely that the fluid for the formation of calcite I was a mixture of meteoric and magmatic components.
The paragraph was added to the new text version (lines 943-948).
I made several minor comments directly on the previous manuscript as well as indicating places where the English needed improving. Many of these have not been addressed
Reviewer 3 Report
The revised draft of the manuscript is a minor improvement over the earlier draft. The only significant improvement is the table indicating the relative amount of the various secondary minerals. Important omissions remain including an absence of any information about the alteration of the primary igneous minerals. The secondary minerals are described as if they were pulled out of cavities of unknown origin without any reference to the structures or textures involved, the position in the flow, or the vesicular origin. Description of some secondary minerals are detailed while others are ignored. Secondary plagioclase is mentioned but totally undescribed. Secondary plagioclase can supply considerable useful information. In addition most classic work on secondary mineralization was not cited while some very obscure papers of marginal significance were. Most of my suggestions were also ignored and I am quite disappointed.
Author Response
Dear reviewer,
we submit a new, thoroughly revised version of our manuscript….
The new version contains a new chapter about the host-rocks and related alteration processes providing elements necessary for the formation of the secondary minerals. The secondary minerals themselves are discussed in more detail. New Figures and chemical data are given.
In addition, there is some additional discussion about the comparison of the investigated area with former studies of secondary mineralization in other basalt provinces.
In conclusion, we rewrote several parts of the manuscript, added new data and think that we fulfill the requirements of the reviewers (in particular reviewer 3). You will find all changes indicated in the new version of the paper.
Answer to reviewer 3
The revised draft of the manuscript is a minor improvement over the earlier draft. The only significant improvement is the table indicating the relative amount of the various secondary minerals.
Important omissions remain including an absence of any information about the alteration of the primary igneous minerals.
The lack of any information about the alteration of primary igneous minerals is complained. As stated in the paper, the investigations were focused on the secondary minerals in the big cavities of the core line of the lava flow in Savda. In the host rocks surrounding the investigated cavities plagioclase and pyroxene show minor alteration whereas glass and olivine are mostly replaced. We did not study them in more detail. Nevertheless, we will document general information about alteration of basalt and specific ones in the Savda basalt in the revised version. Alteration processes in other zones of the lava flow or other flows need an extensive additional research and were not part of the present investigations
The secondary minerals are described as if they were pulled out of cavities of unknown origin without any reference to the structures or textures involved, the position in the flow, or the vesicular origin.
The position of the big cavities is described in chapter 4.1 Observations in the outcrops. The authors have sampled personally in the quarry and reported the coordinates for the samples of age dating. It was generally stated in 3.1 Sample material that “ The sample material includes mainly secondary minerals from the big cavities within the core line of the lobes in the quarry complex of Savda near Jalgaon (Maharashtra, India)”. We believe that big cavities in the core line are a quite precise described, because due to ongoing excavation cavities get destroyed but continuously new ones get accessible.
Description of some secondary minerals are detailed while others are ignored.
In the coming update we will also add additional data for some other secondary minerals.
Secondary plagioclase is mentioned but totally undescribed. Secondary plagioclase can supply considerable useful information.
The detection of secondary plagioclase in only one sample was reported (we will give EDX analysis), but does not provide enough arguments for a general discussion. Authors believe the detection of plagioclase plays a minor role and has no influence for the multi-stage mineralization model.
In addition most classic work on secondary mineralization was not cited while some very obscure papers of marginal significance were.
The criticized lack of references of other secondary mineralization sites is surprising, since in our first update we added 20 citations about secondary mineralization including explanations in several chapters. We personally believe that this is probably enough, but will add another few additional.
Round 3
Reviewer 2 Report
The authors have made many improvements in this revised version.
All the comments I made were for additions in explanation to be included in the paper not just for my personal satisfaction.
There are still a few points where I think the level of discussion is inadequate but the authors suggest this is outside the scope of this paper. The editors can decide if they think it is adequate. If so then the paper is suitable for publication after some tidying up of the English.
Author Response
Answer
The present paper is based on many investigations and analyses, which have not been made before in such a comprehensive matter for the secondary minerals of the DVP. For instance, the paper includes for the first time age dating of secondary cavity minerals from the DVP, gives first detailed explanations for the subsurface filamentous fabrics and develops for the first time a multi-stage mineralization model for a specific area in the DVP, based on numerous data obtained by advanced analytical methods.
Nevertheless, we can understand the interest for more information to specific points. However, the number of analyses could not be extended endless and the volume of the paper had to be limited to a suitable size. Therefore, the study could provide a fundamental basis for further studies of the DVP.
For instance, the alteration processes of comparable basalts have been described in other papers and the formation of the clay minerals is generally accepted. We should not enlarge the paper by discussing commonly accepted processes and included appropriate references.
One specific point for further studies might be the occurrence and probably biogenetic origin of the subsurface filamentous fabrics (SFF). In the past such formations have wrongly been interpreted as chalcedony stalactites only based on visual observations. In our paper such fabrics are for the first time investigated in detail, providing strong indications for their biogenetic origin. We believe that the explanations in this point are very usable to bring the focus on the SFF, which occur in the DVP in an unusual size, variety and quantity.
Other points like the origin of vanadium need extensive new, mainly petrographic investigations and new field work.
We hope you can agree with this statement.
Reviewer 3 Report
There is still a very serious problem with the Manuscript by Ottens et al. Important improvements have been made but other problems have been made much worse. The revised line 92 is not true. "Comparable investigations of other basalt provinces do not exist" is false and totally ignores paper that I have recommended pertaining to the zeolite and prehnite province of Paterson, New Jersey. The claim is made on lines 90-96 that the "big cavities in the lava flow core zone in Savda … " and the occurrence of apophyllite together with different generations of calcite are unique to Siliva. That is absolutely not the case. I have been to both the Deccan zeolite province and to the Paterson zeolite province. There is substantial overlap. Although the origin and size of the "cavities" examined by Ottens el al are not clearly specified, Puffer et al (Bull of Volcanology, 2018) describe in great detail "Mega-vesicles up to 1 m across are common within diapirs, as are collapsed vesicle breccia lenses up to 4 m across." Large, euhedrial, apophyllite is commonly found together with several other zeolites throughout the Paterson province and is described by several authors. Still larger mineralized lava tubes are also described. And what exactly is the origin of the Savda "cavities"?
Author Response
Comparison with zeolite mineralization in the Paterson province (USA):
You recommend the zeolite and prehnite province of Paterson (USA) for comparison with the Savda mineralization. According to our knowledge and data from literature (e.g.: Faust, A Review and Interpretation of the Geologic Setting of the Watchung Basalt Flows, New Jersey, 1975; Mason, The trap rock minerals of New Jersey, 1960; Kent and Butkoswski, Minerals of the Millington quarry, 2000 and Laskowich and Puffer, 2016) the basalt formation of the Watchung Mts. including Paterson and Millington is very different to the Central DVP. Considering the stratigraphy of the Watchung basalts large sediment masses are under-, inter- and overlaying the three flows. In contrast, in the investigated area of the Central DVP the stratigraphy consists of basalts only, without sediments between or above.
The secondary mineralization in the Watchung Mts. is widely different to the Central DVP. In the Watchung Mts., the mineral paragenesis shows several periods, starting with saline minerals and pillow lavas, indicating that the early lava spilled into water (pillows you find in the coastal part of the DVP e.g. in Mumbai, but not in the main part). The most important minerals described by Laskowich and Puffer (2016) occur in the second flow “extruded into a shallow brackish lake water”. This is similar to the mineralization of the Mumbai Salsette group, but very different to the main part of the DVP. The later mineralization sequences in the Watchung Mts. are quite different to the Central DVP. Prehnite, natrolite, babingtonite, ilvaite, pyrite, galena, sphalerite are found in the DVP, but only in the spilite of the costal part, and not in the main part of the DVP (Ottens, 2003, 2011). Pectolite, datolite, casts after anhydride and glauberite do not occur in the DVP. The formation of secondary minerals in the Watchung Mts. indicates similarities to the conditions in the coastal part of the DVP, but is widely different to the Central DVP.
The mineralization conditions in Paterson and in the Mumbai/Salsette subgroup seem to be very similar. Both occurrences contain prehnite and laumontite. Both are pillow basalts, whereas Mumbai has been one of the latest Deccan volcanic events and is not overlain by any significant later flows, excluding burial metamorphism. Formation temperatures for prehnite could not have been acchieved by burial, late hydrothermal activity is most likely.
Therefore the zeolite and prehnite province of the Watchung Mts. is not usable for comparison with the Central DVP and not considered. Nevertheless, we included a short comment with a reference in the text at the end of the discussion (lines 1260-1264) to emphasize the different possible mineralization conditions.
Lines 92/93:
To avoid misunderstandings, the sentence in line 92 is deleted. In line 93 we replaced the word “exclusive” by “specific”.
Origin of big cavities:
The origin of the cavities has not been part of our investigations. It is known that in the main part of the DVP several different kinds of open spaces exist, necessary for the formation of the secondary fine developed crystals. Lava tunnels or breccia’s are observed at several places, e.g. Pune. Most common for the formation of the secondary minerals are the compound flows, which show a typical structure with a high vesiculated top, a central zone with a less number of vesicles/cavities and the bottom with upwards orientated pipe- or Y-shaped vesicles (e.g. Self et al., 1998).
The basalt in the Savda quarry is formed by compound flows and presents the characteristic structure as explained under chapter “Results” with an unusual high number of large cavities. It is believed that large cavities occur predominantly in thick flows. There is no evidence for breccia, lava tunnels or similar features. The flows in the approx. 3000 m long and 200 – 500 m wide quarry complex do not indicate different conditions. Due to long term observations over 20 years 4 m2 of horizontal projected basalt per 1 big cavity can be calculated. The area of effective extracted basalt is roughly calculated with 300,000 m2 indicating a total number of 75,000 excavated big cavities. The even distribution over a large area doesn’t indicate the origin of the cavities by breccia, tunnels, collapsed material or volcanic diapirs. Big cavities of the type discussed by Laskowich and Puffer (2016) do not occur in Savda. However, in the Mumbai spilite you can observe similar pockets as in Paterson, probably not as big.
The origin of the big cavities in Savda is believed to be explained by the accepted process of vesiculation of compound flows.
Sahagian et al. (2002) describe: ”Vesicular zones develop in lava flows because of buoyant bubble rise and concurrent crystallization of the lava from top and bottom (Sahagian 1985). As the bubbles rise from the base, they are quickly caught by a rapidly rising crystallization front. Like the cooling of the ocean lithosphere, crystallization fronts progress at a speed proportional to the square root of time. They start quickly, then slow down as they are more insulated by the surrounding lava. Thus virtually all bubbles are “frozen in” at the base of the flow, while only the smallest bubbles are caught by the crystallization front higher up (the larger ones can escape by their more rapid rise through the lava). The situation at the top of the flow is the same, except that bubbles rise headlong into the dropping crystallization front. Thus the very top includes all bubbles as they were originally emplaced (just as at the very bottom). The vesicularity of the flow interior is the result of bubble rise and coalescence, with a specific and readily identifiable size distribution as a function of stratigraphic position in the flow (fig. 1; Sahagian et al. 1989). In the center of the flow, bubbles have the longest time to rise from the lower parts and coalesce with each other.”
We assume that the vesiculation in Savda took place under the same conditions. Therefore, this kind of vesiculation is generally accepted and it was not investigated for this paper and we do not discuss this topic in detail. We only added one sentence with a reference (lines 1264-1268) to point to earlier investigations.